# Characteristics of consecutive tsunamis and resulting tsunami behaviors in southern Taiwan induced by the Hengchun earthquake doublet on 26 December 2006

An-Chi Cheng[1,2], Anawat Suppasri[2,3], Kwanchai Pakoksung[3], Fumihiko Imamura[2,3]

[1]Civil and Environmental Engineering, Graduate School of Engineering, Tohoku University, 6-6-06 Aoba, Aramaki-Aza, Aoba, Sendai 980-0845, Japan

[2]WISE Program for sustainability in the Dynamic Earth, Tohoku University, 6-3 Aoba, Aramaki Aza, Aoba, Sendai 980-8578, Japan

[3]International Research Institute of Disaster Science, Tohoku University, 468-1 Aoba, Aramaki-Aza, Aoba, Sendai 980-0845, Japan

*Correspondence to:* An-Chi Cheng (cheng.anchi.r6@dc.tohoku.ac.jp)

**Abstract.** Consecutive $M_L 7.0$ submarine earthquakes occurred offshore the Hengchun Peninsula, Taiwan, on 26 December 2006. A small tsunami was generated and recorded at tide gauge stations. This important event attracted public interest, as it was generated by an earthquake doublet and produced a tsunami risk for Taiwan. This study analyzed tide gauge tsunami waveforms and conducted numerical simulations to understand the source characteristics and resulting behaviors of tsunamis. The maximum wave heights at the three stations were 0.08 m (Kaohsiung), 0.12 m (Dongkung), and 0.3 m (Houbihu), and only Houbihu recorded the first wave crest as the largest. The tsunami duration was 3.9 h at Dongkung and over 6 h at Kaohsiung and Houbihu. Spectral analyses detected dominant periodic components of spectral peaks on the tsunami waveforms. The period band from 13.6-23.1 min was identified as the tsunami source spectrum, and the approximate fault area for the consecutive tsunamis was estimated to be 800 $km^2$, with central fault depths of 20 km (first earthquake) and 33 km (second earthquake). The focal mechanisms of the first earthquake, with a strike of 319°, dip of 69°, and rake of -102°, and the second earthquake, with a strike of 151°, dip of 48°, and rake of 0°, could successfully reproduce the observed tsunami waveforms. Numerical simulations suggested that the tsunami waves were coastally trapped on the south coast of Taiwan during the tsunami's passage. The trapped waves propagated along the coast as edge waves, which repeatedly reflected and refracted among the shelves, interfered with incoming incident waves, and resonated with the fundamental modes of the shelves, amplifying and continuing the tsunami wave oscillation. These results elucidated the generation and consequential behaviors of the 2006 tsunami in southern Taiwan, contributing essential information for tsunami warning and coastal emergency response in Taiwan to reduce disaster risk.

## 1. Introduction

Taiwan is located at the southeast margin of the Eurasian plate and the Philippine Sea plate. The abrupt
movement of plates results in active seismic activity at the boundary area, such as in the Manila Trench and
Ryukyu Trench. The Manila Trench and Ryukyu Trench are located in offshore Taiwan and have been
identified as hazardous tsunamigenic regions, as both have the potential to generate megathrust earthquakes
and cause severe tsunami impacts on coastal plains (Liu et al., 2009; Megawati et al., 2009; Wu and Huang,
2009; Li et al., 2016; Sun et al., 2018; Qiu et al., 2019). In addition to potential megathrust earthquakes,
historical earthquake tsunamis in Taiwan are well recorded in ancient and written documents. Examples
include the 1781/1782 Jiateng Harbor flooding and tsunami event (Okal et al., 2011; Li et al., 2015) and
the 1867 northern Taiwan earthquake (Cheng et al., 2016; Sugawara et al., 2019).
Two large earthquakes occurred off the coast of Hengchun Peninsula, Taiwan, on 26 December 2006.
The first earthquake occurred at 12:26:21 UTC (i.e., 20:26:21 Taiwan Standard Time) and was followed by
a second earthquake 8 min later at 12:34:15 UTC (i.e., 20:34:15 Taiwan Standard Time). The Central
Weather Bureau (CWB) catalog (R.O.C.) located the epicenter of the first shock at 21.69° N and 120.56° E
and that of the second shock at 21.97° N and 120.42° E. The locations of the Hengchun Peninsula and the
epicenters of the successive earthquakes are shown in Figure 1.
The respective magnitudes of these two earthquakes were suggested to be $M_L = 7.0$ ($M_w = 7.0$ in the
Global Centroid Moment Tensor (CMT) catalog) for the former and $M_L = 7.0$ ($M_w = 6.9$ in the Global CMT
catalog) for the latter. From a seismological perspective, pairs of large earthquakes with equivalent fault
sizes that occur in similar spatial and temporal proximities are referred to as doublets (Lay and Kanamori,
1980; Kagan and Jackson, 1999). As they shared comparable earthquake magnitudes and very close
epicenters and occurrence times, the successive earthquakes on 26 December 2006 are considered an
earthquake doublet event (Ma and Liang, 2008; Wu et al., 2009). These 2006 earthquakes in southern
Taiwan were considered the largest event in the past hundred years. Several casualties and some structural
damages were reported in southern Taiwan during this seismic event (National Disaster prevention and
Protection Commission, R.O.C., 2007). The tectonic settings of the 2006 earthquake doublet are shown in
Figure 2.
A small tsunami was generated after the successive strong motions of these earthquakes. The tsunami
propagated toward and reached the western coast of southern Taiwan immediately after the earthquakes.
Although no coastal run-up or inundation was reported, tsunami signals were instrumentally recorded at
CWB tide gauge stations in southern Taiwan. The December 2006 tsunami was an important event that
attracted public interest, as it was unique not only because it was generated by earthquakes in short
succession but also because it was a new occurrence for ordinary citizens in Taiwan. This recent tsunami
not only corroborates the tsunami risk in Taiwan but also increases awareness of disaster risk management,
such as preparedness and mitigation countermeasures for future tsunamis.
The tsunami observations that were reported following the 26 December 2006 tsunami also raised some
questions. First, the first tsunami wave crest was not shown to be the largest at some stations. This amplified
tsunami wave is considered an important issue for tsunami warnings, as a higher later wave could suddenly
upgrade the threat level of the tsunami (Suppasri et al., 2017; Suppasri et al., 2021). Second, the tsunami
oscillation recorded at some stations lasted for more than 6 h following the earthquakes. This indicated that
the high-energy waves persisted along the coast without decay during the 2006 tsunami and were considered
one of the cascading risks of tsunamis, as they could further intensify the damaging impacts of the tsunamis
on the coastal region.
The other issue was to identify which source models could better explain the successive tsunamis
compared to the recorded observations in southern Taiwan. Wu et al. (2008) simulated the tsunami from
this event using single fault models. They numerically computed the tsunami propagation on a nested grid
system with fine-grid of 0.125 arc min resolution bathymetry data and compared their results with
observational data from tide gauge stations. Although the source models for this tsunami event have been
specified and modeled in previous studies, the uncertainty and variability aspects of these models and the
bathymetry have not been thoroughly investigated. These uncertainties in earthquake fault parameters and
significant differences among open-source bathymetries can exaggerate the modeled results compared to
the predictions of previous studies to the 2006 tsunami. Therefore, it is critical to discuss these model
performances from a sensibility perspective because it is desirable to obtain a tsunami source model and to
understand the reliability of bathymetry data that is utilized for numerical simulation to reasonably estimate
the tsunami wave activities of the 2006 tsunami.
Based on the above background, the primary intent of this article is to address all aforementioned issues
related to the 2006 tsunami that have not been previously discussed and to provide some results. The content
of this article is organized as follows. First, the observed tsunami waveforms are analyzed to determine the
physical characteristics of the tsunami and employed as inputs for root mean square (RMS) analyses to
detect the tsunami duration. Second, spectral analyses are performed to detect the periodic components of
the tsunami waves based on the identification of the tsunami source spectrum and resonance modes. Then,
a sensitivity analysis of the source models and open-source bathymetries is conducted based on the
simulated waveforms from forward tsunami simulations. The mechanism of tsunami wave trapping around
southern Taiwan is examined based on the comparison of modeled results from numerical experiments
using actual and manipulated bathymetry. The December 2006 earthquake tsunami represents a unique and
recent incident in Taiwan; therefore, these findings could not only help further clarify tsunami generation
and the important behaviors responsible for tsunami hazards facing the island of Taiwan but also have
implications for tsunami warning and disaster risk management.

**2.   Data and methods**
**2.1 Tide gauge tsunami data**
Time history data of sea levels that are recorded at coastal sites provide one source of information that
we can use to study tsunami patterns. To investigate the characteristics of the 2006 tsunami, sea level
records from tide gauge stations were employed for analysis in the present study. For this purpose, the
recorded data from three tide gauge stations (Kaohsiung, Dongkung, and Houbihu) located in southern
Taiwan were obtained. These tide gauge stations are operated and maintained by the CWB, R.O.C. All
stations recorded sea levels at a sampling rate of 6 min. In this doublet event, the first and second
earthquakes occurred at 20:26:21 and 20:34:15 (Taiwan Standard Time), respectively. Hence, 28 h of tide
gauge records (from 8:00 on 26 December 2006 to 12:00 on 27 December 2006, Taiwan Standard Time)
were adopted for analysis. To approximate the wave components of the tsunami and to remove the low-
frequency noise that was attributed to the tidal effect, the sea level records at the tide gauge stations were
de-tided by removing the long-period (> 2 h) tidal constituents. The original data recorded at the tide gauge
stations in southern Taiwan are shown in Figure 2a, and the de-tided data are presented in Figure 2b. The
locations of the tide gauge stations are shown in Figure 3.
The tsunami durations represent the observation time of high-energy tsunami waves persisting at a
coastal site. The tsunami durations at all the stations were identified based on a calculation of root mean
square (RMS) sea levels, indicating the elapsed time of the wave amplitude above the normal oscillation
level before the tsunami wave arrived (Heidarzadeh, 2021). The RMS analysis calculated the moving
average of the recorded sea level along a moving time window of 24 min. The calculation for RMS sea
level is presented in equation (1):

$$S(t) = \sqrt{\frac{1}{w} \int_{t-\frac{w}{2}}^{t+\frac{w}{2}} h(x)^2 \, dx} \tag{1}$$


In this equation, S (t) represents the RMS sea level at time step t, h (t) denotes the recorded sea level at time
t, and w stands for the moving time window. In the present study, the length of the tsunami data employed
for RMS analysis is 12 h, which includes 120 data points, ranging from 17:00 on 26 December 2006 to
5:00 on 27 December 2006 (Taiwan Standard Time). A similar method has been applied in the research by
Hayashi et al. (2012).

**2.2 Spectral analyses**
To apply spectral analyses to the tsunami data, two types of analyses were included and processed in this
study: the Fourier analysis and the wavelet (time-frequency) analysis. The Fourier analysis is based on the
fast Fourier transform (FFT) algorithm and applied based on the updated open-source library, Numpy, in
the Python package (Harris et al., 2020). The Fourier analysis was performed to estimate the spectral
components of the time history data of the tsunami waveform. The entire dataset of the tsunami waveform
inputted for Fourier analysis covered 600 min, which included 100 data points ranging from 5 h before to
5 h after the tsunami, as the sampling rate of the data was 6 min. The Fourier analysis was separately applied
to the de-tided background (i.e., 5 h data before the tsunami arrival) and the tsunami signals (i.e., 5 h data
after tsunami arrival) to identify significant changes in the spectral energy associated with the tsunami.
Additionally, the spectral ratio was computed for the tsunami spectra to exclude the local modes of coastal
sites from the periodic components. Wavelet analysis was computed based on the Morlet mother function,
as suggested by Torrence and Compo (1988). Wavelet analysis detects the periodic change in spectral peaks
over time. The length of the tsunami data input in the wavelet analysis was 15 h (15:00 on 26 December
2006 to 06:00 on 27 December 2006, Taiwan Standard Time). A similar method has been widely applied to
solve time-frequency problems for many recent tsunami events, such as the 2018 Sulawesi tsunami in
Indonesia and the 2020 Aegean Sea earthquake tsunami (Heidarzadeh, 2019; Heidarzadeh, 2021).
**2.3 Numerical tsunami simulation**
Numerical simulation is a computer-based method that describes equations for the motion of tsunami
wave propagation. Tsunami wave propagation can be numerically modeled based on various theories,
including shallow water and dispersive wave theories. Among those theories, the shallow water equations
are some of the most commonly used methods to model tsunami propagation from the source to nearshore
areas. Various computational models have been developed to solve shallow water equations, and the
TUNAMI (Tohoku University Numerical Analysis Model for Investigation of tsunamis) code is one of the
widely used models to numerically simulate both far-field and near-field tsunamis (Suppasri et al., 2010;
Suppasri et al., 2014). The second version of the TUNAMI code (TUNAMI-N2) was mainly developed to
deal with near-field tsunamis by applying the nonlinear theory of shallow water equations, which is solved
using a leap-frog scheme (Imamura, 1995). Since the 2006 tsunami presented as a near-field tsunami in
Taiwan, the TUNAMI-N2 model was used in this study to simulate the 2006 tsunami with nonlinear shallow
water equations. The nonlinear shallow water equations on the Cartesian coordinate system are presented
in equations (2)-(4), and the nonlinear equations are solved by applying the finite difference method:

$$\frac{\partial \eta}{\partial t} + \frac{\partial M}{\partial x} + \frac{\partial N}{\partial y} = 0 \tag{2}$$

$$\frac{\partial M}{\partial t} + \frac{\partial}{\partial \chi}\left(\frac{M^2}{D}\right) + \frac{\partial}{\partial y}\left(\frac{MN}{D}\right) + gD\frac{\partial \eta}{\partial \chi} + \frac{gn^2}{D^{\frac{7}{3}}}M\sqrt{M^2 + N^2} = 0 \tag{3}$$

$$\frac{\partial N}{\partial t} + \frac{\partial}{\partial \chi}\left(\frac{MN}{D}\right) + \frac{\partial}{\partial y}\left(\frac{N^2}{D}\right) + gD\frac{\partial \eta}{\partial y} + \frac{gn^2}{D^{\frac{7}{3}}}N\sqrt{M^2 + N^2} = 0 \tag{4}$$

In these equations, $\eta$ is the water level, $M$ and $N$ are the discharge fluxes in the $x$ and $y$ directions,
respectively, $D$ is the total water depth, $g$ is the gravitational acceleration, and $n$ is Manning's roughness
coefficient. The bottom friction term was represented by the Manning roughness coefficient, which was set
as 0.025 s m$^{-1/3}$, assuming that the seafloor in the model domain is in perfect condition. The numerical
tsunami simulations were conducted with a time interval of 0.1 s and grid intervals of 450 m. The entire
model domain covered the source region and southern Taiwan, which comprised mesh numbers of 538 and
631 in the x and y directions, respectively. The time interval and grid intervals were set up to satisfy the
Courant–Friedrichs–Lewy (CFL) condition to ensure the stability of the simulation. The CFL condition is
presented in equation (5):

$$\Delta t \leq \frac{\Delta x}{\sqrt{2gh_{max}}} \tag{5}$$

where $\Delta t$ is the time interval, $\Delta x$ is the grid spacing, and $h_{max}$ is the maximum water depth in the model
domain. As the initial condition inputted for numerical tsunami simulation, the initial water level
distribution was calculated from the earthquake fault parameters using the theory of Okada (1985). In
addition, the horizontal deformation contribution to tsunami generation on steep bathymetric slopes
(Tanioka and Satake, 1996) was included. The calculation conditions for the numerical tsunami simulation
are summarized in Table 1.

**2.4 Sensitivity analyses of source models**
**2.4.1 Single fault models**

Multiple forward tsunami simulations were conducted using single fault models with different fault

depths and fault orientations. The main goal of the multiple forward tsunami simulations was to find a
single fault model that could produce tsunami waveforms that were highly consistent with the tide gauge
station observations in southern Taiwan.

There were two moment tensor solutions available from the Global Centroid Moment Tensor (GCMT)

Project and United States Geological Survey (USGS) for the successive earthquakes on 26 December 2006
(Figure 2.). Each solution suggested two possible fault planes for those earthquakes. The focal mechanisms
for the two earthquakes estimated by the GCMT and USGS are summarized in Table 2. Through the analysis
of the tsunami waveforms simulated by the multiple forward tsunami simulations, one of those fault planes
could be chosen as the appropriate fault plane for the respective earthquakes of the 2006 earthquake doublet.
A similar approach has been applied in a previous study to obtain the optimum fault plane for the 2016
Fukushima normal faulting earthquake (Gusman et al., 2017).

Wu et al. (2008) computed synthetic tsunami waveforms based on single fault models using different

fault planes of the GCMT solutions. They found that the nodal plane (NP) of NP2 of the first earthquake,
with a strike of 329°, dip of 61°, and rake of -98°, and the fault plane of NP1 for the second earthquake,
with a strike of 151°, dip of 48°, and rake of 0°, produced tsunami waveforms that better fit the observed
data.

Based on the study conducted by Wu et al. (2008), the focal mechanisms of NP2 to the first earthquake

and NP1 to the second earthquake from the GCMT solution were used for a sensitivity analysis of fault
depths. An approximated fault area with a 40 km length and a 20 km width (fault size = 800 km$^2$) was
estimated for the successive earthquakes based on the empirical formula with tsunami source periods. The
methods by which the fault area of the two earthquakes was obtained are discussed in section 4.1. For the
given moment magnitude ($M_w$) values of the 7.0 and 6.9 earthquakes, the amount of average slip can be
estimated to be 1.66 m for the first earthquake (i.e., $M_w$ 7.0) and 1.17 m for the second earthquake ($M_w$ 6.9),
assuming a rigidity of 30 Gpa. The centroid depths of the GCMT (20 km) and USGS (25 km) solutions for
the first earthquake are significantly different, while a similar depth of 33 km was estimated from both
solutions for the second earthquake. Therefore, for the sensitivity analysis of central fault depth, the central
fault depths of 15, 20, 25, and 35 km of the first earthquake were evaluated.
After determining the best central fault depth for the single fault models of the two earthquakes, multiple
tsunami forward simulations were applied to all possible fault planes from the moment tensor solutions
estimated by GCMT and USGS using a single fault. The misfit of observed and simulated tsunami
waveforms from the multiple tsunami forward simulations was calculated and compared to examine the
focal mechanisms that better explain the observed tsunami data. The misfit of the observed and simulated
tsunami waveforms can be calculated using equation (6):

$$\varepsilon = \frac{1}{N} \sqrt{\sum_{i=1}^{N} \frac{(Obs_i - Sim_i)^2}{(Obs_i)^2}} \qquad (6)$$

where $\varepsilon$ is the misfit of the observed and synthetic tsunami waveforms, $N$ is the total number of data
points, $Obs_i$ is the observed data at time step $i$, and $Sim_i$ is the simulated data at time step $i$. Equation
(8) calculates $\varepsilon$ for one station. For cases with several stations, the overall misfit is obtained from the mean
of the $\varepsilon$ values computed from all the stations.

**2.4.2 Multiple fault models**
After determining the best central fault depths and fault orientations of a single fault, the area of each
single fault was subdivided into 8 subfaults with areas of 10 km $\times$ 10 km, with 4 and 2 subfaults along
the strike and dip axes, respectively. The locations of each subfault in the fault model of the two earthquakes
are shown in Figure 4. The top depths for the two earthquakes are 15.3 km and 29.1 km, which correspond
to subfaults 1-4 in each fault model (Figure 4a, b). The rest of the depths from the shallowest to the deepest
portion along the dip axis are derived using fault parameters of width dimensions and dip angles. The
respective fault parameters of each subfault in the fault models of the two earthquakes are summarized in
Table 3.
The tsunami sensitivity to the non-uniform slip distribution of the fault model was evaluated. For that
purpose, two slip levels for each subfault were established, namely, the large (asperity) slip and the
background slip region of the entire fault. The large slip and background slip region should satisfy the $M_w$
to avoid overestimation. The slip amount in each region was obtained using the following procedures. First,
the amount of average slip ($D_a$) was calculated using the $M_w$, the entire fault area (S), and a rigidity ($\mu$) of
30 GPa, per the equations introduced by Kanamori (1977):

$$M_w = \frac{\log M_0 - 9.1}{1.5} \qquad (7)$$

$$D_a = \frac{M_0}{\mu S} \tag{8}$$

Next, the amount of large slip ($2D_a$) was assumed to be twice that of the average slip based on a 2017
tsunami receipt report. The total area of the large slip area (S') was set to be 25% of the entire fault area,
and the seismic moment of the large slip area ($M_0$') can be obtained using equation (8). Then, the slip
amount of the background area ($D_b$) can be estimated using the area of the background region ($S_b$) following
equations (8)-(9):

$$S_b = S - S' \tag{8}$$

$$D_b = \frac{M_0 - M_0'}{\mu S_b} \tag{9}$$

The details of the slip amount in each region for the two earthquakes are summarized in Table 4a.
After determining the slip amount of the asperity and background regions, the tsunami sensitivity to the
non-uniform fault slip distribution was studied. The asperity area with the large slip was located in the
shallow portion of the entire fault area based on information from the 2011 Tohoku-Oki earthquake (Satake
et al., 2013; Fukutani et al., 2021), focusing on the north (subfaults 3-4), central (subfaults 2-3), and south
(subfaults 1-2) parts of each earthquake fault model. Assuming different asperity locations for the two
earthquakes, a total of 9 scenarios were simulated. The multiple fault models and the generated tsunamis
of each earthquake are shown in Figures 5 And 6. The asperity locations of multiple fault models for the
two earthquakes in each scenario are summarized in Table 4b.

**2.5 Tsunami simulation using open-source bathymetry data**
In addition to the fault parameters of the source models, bathymetry data are needed for simulating
tsunami wave propagation. Simulated tsunami propagation results are known to be sensitive to the accuracy
and resolution of bathymetry data. Although it can be expected that bathymetry data with a higher accuracy
and resolution can produce simulated results that better fit the actual values, such data are not always
available and freely accessible. Due to this limitation, open-source datasets have often been utilized for
modeling tsunamis in many previous studies (Koshimura et al., 2008; Suppasri et al., 2012; Li et al., 2016;
Otake et al., 2020).
Unfortunately, open-source datasets are sometimes problematic and insufficient for the accurate
simulation of tsunami waves because they lack accurate, quality data (Griffin et al., 2015). A similar issue
has been reported by Zengaffinen et al. (2021) and Heidarzadeh et al. (2019) in simulating the 2018 Anak
Krakatoa tsunami and the 2018 Sulawesi tsunami. Significant differences in various sources of datasets can
also result in modeled results that in contrast to values from previous studies. Therefore, , it is important to
assess and note different available open-source bathymetries in relation to model performances for purpose
of tsunami hazard assessment..
For this purpose, a tsunami simulation was separately applied to two different sources of bathymetry data,
namely, General Bathymetric Chart of the Oceans (GEBCO) data and ETOPO1 data, and the misfit between
the modeled results was evaluated. The GEBCO data contain bathymetry data with grid intervals of 15 arc
seconds, while ETOPO1 data have sea depth data with a resolution of 1 arc minute. To fairly investigate
the model performances from different datasets, bathymetry data from the two datasets were converted to
450 m grids and used as the input for the numerical tsunami simulations. Figure 7 Shows the bathymetry
data of the modeled domain obtained from GEBCO and ETOPO1 data. As the initial condition, the
simulated initial water distribution of the tsunami generated by the proposed multiple fault model (LS2)
was used for these simulations, in which the asperity locations of the two earthquakes were assumed to be
at the center of the entire fault area.

**2.6 Evaluation of the bathymetry effect on tsunami wave trapping**
To examine any significant change in tsunami wave transmission that could be attributed to the
bathymetry effect during the passage of the 2006 tsunami, numerical experiments (MS, EXP1, EXP2) for
tsunami propagation were conducted using actual and manipulated bathymetry data. For the main
simulation (MS) numerical experiment, actual GEBCO bathymetry data with a resolution of 450 m derived
from sea depth data with grid intervals of 15 arc seconds were used. For the manipulated bathymetry data
that were used for numerical experiment EXP1, sea depths greater than 500 m were replaced with 500 m
depths. For numerical experiment EXP2, the bathymetry data were manipulated by removing sea depth
data with a flattened sea bottom at a depth of 500 m. The 500 m depth was specified because the bathymetric
slopes are very gentle at sea depths shallower than 500 m near southern Taiwan, and the area is therefore
considered a shelf region. Figure 8 Shows the map-manipulated bathymetry of the model domain for
numerical experiments EXP1 and EXP2. The details of the bathymetry data used for numerical experiments
MS, EXP1, and EXP2 are summarized in Table 5.
The results of the numerical experiments were compared to examine how tsunami wave directivity could
change due to the bathymetric effect and to evaluate how much tsunami wave energy could be coastally
trapped in different bathymetric conditions during the passage of the tsunami.

**3.    Analyses of tsunami waveforms and durations**
**3.1 Physical characteristics of tsunami waveforms**
The December 2006 earthquake tsunami was observed at several tide gauges located along the
southwestern coast of Taiwan. The tsunami observations are plotted in Figure 9a. The initial wave arrived
at all three tide stations in southern Taiwan with an amplitude sign of a trough wave. The travel times of
the initial wave to all the stations were recorded: 16 min to Houbihu, 28 min to Dongkung, and 52 min to
Kaohsiung. The initial wave was recorded as -0.12 m in Houbihu, -0.09 m in Dongkung, and 0.06 m in
Kaohsiung. Following the trough sign of the initial wave, the first wave crest record at Houbihu was 0.3 m,
which was approximately 3 times greater than that at Dongkung and 4 times larger than that at Kaohsiung.
This was natural because Houbihu was the station closest to the epicentral region and therefore had an
earlier arrival time and was relatively sensitive to the surface elevation change in sea level that was induced
by the tsunami. The maximum wave heights were recorded as 0.08 m (Kaohsiung), 0.12 m (Dongkung),
and 0.3 m (Houbihu). In Kaohsiung and Dongkung, the maximum height was not recorded for the initial
wave. The maximum wave height appeared 36 min after the initial wave arrived at Kaohsiung and after 24
min at Dongkung, indicating a pattern of wave amplification at these stations. These results suggest that
different propagation effects existed at these coastal sites during the passage of the 2006 tsunami. In
addition to significant differences in wave amplitude and arrival time, the tsunami records at each station
also varied in terms of visible wave periods. The visible period of the tsunami wave at Kaohsiung was
recorded from 30-48 min based on the tsunami waveform, which was approximately two times longer than
those observed at Dongkung and Houbihu (from 18-24 min). This indicated that wave components with
shorter periods were not well recorded in Kaohsiung. The locations and details of the tide gauge
observations are summarized in Table 6a for wave amplitude and Table 6b for arrival time and visible period.

**3.2 Tsunami durations**
Another issue was to determine the tsunami duration at each station because it can help to identify the
length of wave oscillations at a coastal site due to the tsunami. Typically, the tsunami duration describes the
elapsed time during which a high-energy wave at a tide gauge station exceeds the mean sea level of a normal
oscillation. The normal oscillation was defined as the site-specific oscillation at each station before the
tsunami arrived. RMS analysis was applied to the recorded sea level data at each station. The results of the
RMS analysis are plotted in diagrams shown in Figure 9b.
The RMS Sea level diagram illustrates how long the high-energy wave persisted at each station.
Accordingly, the tsunami duration was determined through a comparison of the RMS sea level and the basic
oscillation in sea level at each station. The maximum RMS sea level derived at the Houbihu station was
estimated to be 2-3 times higher than those at the Dongkung and Kaohsiung stations. The calculated tsunami
duration at Dongkung was as much as 3.9 h, while the tsunami continued for more than 6 h in Kaohsiung
and Houbihu.
Generally, several oscillation modes are expected to be induced during a tsunami event in association
with the tsunami source, propagation path, and topographic effects (Rabinovich, 1997; Rabinovich et al.,
2013). An island setting such as Taiwan, where insular shelves and gentle slopes exist, commonly traps
waves over the shelf during the passage of tsunamis (Roeber et al., 2009). The trapped waves propagate
along the coastline and normally trigger various oscillation modes in the coastal water due to the
interference of wave reflection at the edge of the continental shelves (Yamazaki et al., 2011). The wave
resonance of these oscillation modes with the fundamental modes of the continental shelf can enhance
coastal hazards with amplified amplitudes and long tsunami durations (Wang et al., 2020). The triggered

oscillation modes are expected to be mixed with the tsunami source spectrum in the observation records from the coastal sites. To identify these modes from the tsunami source spectrum, spectral analyses were performed on the observation records at all three tide gauge stations in southern Taiwan, as detailed in the next section.

## 4. Spectral analysis

### 4.1 Tsunami source spectra

To examine the spectral characteristics of the tsunami waves, Fourier analysis was applied to 10 h of de-tided observed data (i.e., 5 h before and after the tsunami's arrival) that was recorded at all the tide gauge stations in southern Taiwan. The background spectra were calculated in addition to the spectra of the observed tsunami waveform to identify the tsunami effect. The background spectra were the spectral components calculated from observed data 5 h before the tsunami's arrival, and the spectral components of the observed tsunami waveform were computed using the 5 h of data recorded at the tide gauge after the tsunami wave arrived. Figure 10 shows the respective spectra of the observed tsunami waveform and background signals at each tide gauge station.

At all the stations, the spectral peaks of the observed tsunami spectra were estimated to be different from those of the background spectra. A visible gap also appeared in the spectral energy between the observed tsunami and the background spectra, revealing the energy generated by the arrival of tsunami waves. To examine the spectral components induced by the arrival of the tsunami waves, the spectral ratio of the observed tsunami and background spectra was derived using equation (10):

$$S_{tsunami}(\omega) = \frac{S_{obs}(\omega)}{S_{bg}(\omega)} \tag{10}$$

In this equation, $S_{obs}(\omega)$ is the spectral component of the observed tsunami waveform, $S_{bg}(\omega)$ is the background spectrum, and $S_{tsunami}(\omega)$ is the spectral component induced by the arrival of the tsunami waves. Figure 11 shows the spectral ratios for the tsunami spectra at all the stations. Equation (6) assumes equivalent background spectra before and after the tsunami's arrival, indicating that there was no large change in the coastal topography during the tsunami event. Although many earlier studies have reported that coastal topography might be largely changed during a massive tsunami event (e.g., Sugawara, 2018; Masaya et al., 2020), this was not the case for the 2006 tsunami because the tsunami wave was small. Therefore, the dominant peaks of the spectral ratio were connected to either the tsunami source or perhaps the wave oscillation induced by the nonsource phenomenon.

Tsunami source periods are periodic components that primarily appear in coastal observations close to the tsunami source region (Toguchi, 2018; Rabinovich, 1977). Accordingly, the tsunami source periods can be estimated from the mean of the spectral ratios calculated from all three stations (i.e., the solid black line shown in Figure 11). From the analysis result of the spectral ratio, the periods of 13.6 min, 16.7 min, and

23.1 min are distinct in comparison to other periodic components. The periods within this band most likely
presented the source periods of the 2006 tsunami since the periodic components within this band were
mostly visible at all stations.

In general, a larger earthquake can ordinarily generate a larger tsunami wave with a longer period. For

instance, the major periods of the 2011 Tohoku-Oki earthquake tsunami were reported to be 37-67 min in
association with that magnitude $M_w$ 9.0 earthquake (Heidarzadeh and Satake, 2013), while shorter dominant
periods of 10-22 min were found for the 2013 Santa Cruz tsunami, a $M_w$ 8.0 earthquake (Heidarzadeh,
2016). According to the theory introduced by Rabinovich (1997), the approximate dimensions of fault
rupture can be estimated from the source periods using the empirical formula defined in equation (11):

$$L = \frac{T}{2}\sqrt{gh} \tag{11}$$


where $g$ stands for the gravitational acceleration and is set to a constant value of 9.81 m s$^{-2}$, $h$ represents the
seafloor depth around the tsunami source region, $L$ denotes the fault rupture dimensions of length or width,
and $T$ is the source period. The approximate source region could be illustrated based on the aftershock
distribution one day after the first earthquake occurred. Assuming that the sea depths around the tsunami
source region range from 0-600 m and the source periods are 13.6 min, 16.7 min, 20.0 min, and 23.1 min,
the relationship between the fault rupture dimensions and sea depths can be derived from equation (3). The
correlation derived from equation (11) is plotted in Figure 12.

Assuming that the mean sea depth around the tsunami source region is 300 m, the fault rupture

dimensions for the two earthquakes can be estimated to be 20-40 km. The approximate fault size of these
two earthquakes was estimated to be 800 km$^2$, where a longer dimension of 40 km was considered the fault
length and 20 km was considered the fault width. The estimation of fault size was fairly consistent with the
results derived from the empirical scaling relations of Papazachos et al. (2004), with findings of 794 km$^2$
in association with the $M_w$ 7.0 normal fault earthquake (first earthquake) and 738 km$^2$ in association with
the $M_w$ 6.9 strike-slip fault (second earthquake).

**4.2  Resonance modes induced by tsunami trapping waves**

In addition to the Fourier analyses, wavelet (time-frequency) analyses were also applied to 15 h of de-

tided observed data (i.e., from 15:00 on 26 December 2006 to 6:00 on 27 December 2006, Taiwan Standard
Time) at all the stations. Wavelet analyses are commonly employed as a method to examine periodic
variations over time series through the distribution of tsunami spectral energy. Figure 13 shows the tsunami
wavelets derived from the tsunami records observed at each station. According to the wavelet plots at all
the stations, period bands of 13.6-23.1 min were clearly recorded after the first wave arrived at all the
stations. This also confirmed that the period bands of 13.6-23.1 min were associated with the source periods.
At Kaohsiung, the tsunami energy became apparent with periods of 16 min and 36 min approximately 3 h
after the arrival of the first wave. In the period channel of 16 min, the oscillation was preserved for
approximately 5 h, while the 36 min channel was occupied by a high-energy wave for more than 9 h. At
Houbihu, more energy was channeled than at other stations in the period bands of 13.6-23.1 min soon after
the first earthquake. This was reasonable because Houbihu was the closest station to the epicentral region
and was therefore considered to be more sensitive to the tsunami source than were the other stations.
Following the arrival of the first wave, the persistent oscillation (i.e., lasting more than 4 h) was visible
approximately 2 h after the first earthquake in the period channels of 16 min, 16.4 min, 20 min, 22.5 min,
25.7 min, 30 min, 36 min, and 60 min. These periodic components were considered as possible modes of
trapped tsunami waves resonating within the shelf since the wave resonance commonly requires some time
to be formed (Heidarzadeh et al., 2021). Among these periods, the 16 min and 36 min periods most likely
presented the resonance mode since that mode is visible at only the Kaohsiung and Houbihu stations, where
tsunami durations of more than 6 h were recorded (Figure 9b.). From the wavelet analysis of the observed
data recorded at the tide gauge stations, the persisting wave oscillations at the Kaohsiung and Houbihu
stations might be attributed to tsunami resonance.

**5.   Sensitivity analyses of source models and bathymetry data**
**5.1  Single fault models**
**5.1.1   Tsunami sensitivity to fault depths**
The sensitivity of simulated tsunami waveforms to fault depth was evaluated by varying the central fault
depths of the first earthquake. Fault dimensions of 40 km × 20 km were applied to the two earthquakes.
The single fault model of the two earthquakes was constructed using the GCMT solution of nodal plane
NP2 for the first earthquake and NP1 for the second earthquake. The tide gauge stations of Dongkung and
Houbihu were chosen for this sensitivity analysis because they the closest stations to the source region and
were therefore more sensitive to the tsunami source. The single fault models of the two earthquakes and the
locations of the near-field tide gauge stations that were used for the sensitivity analysis of fault depths are
shown in Figure 14a.
Figure 14b shows the observed and simulated tsunami waveforms at the Dongkung and Houbihu stations
using different fault depths of the first earthquake. At the Dongkung station, the first circle of simulated
tsunami waveforms matched the observed data well regardless of the fault depths. At the Houbihu station,
the first wave crest of the simulated waveform from a fault depth of 35 km was half the size of the observed
value. Simulated tsunami waveforms with shallower depths of 15 km and 20 km produced significantly
higher amplitudes during the arrival of the first crest wave. These results revealed that coastal sites with a
shorter distance to the source are more sensitive to earthquake fault depths. The simulated waveforms from
a central fault depth of 20 km fit the observed data better than other simulations did, and therefore, this was
considered the best fault depth for simulation.

### 5.1.2 Comparison of eight models


Single fault models with fault dimensions of 40 km $\times$ 20 km and central depths of 20 km for the

first earthquake and 33 km for the second earthquake were used in tsunami simulations using eight different
sets of focal mechanisms for the two earthquakes estimated from GCMT and USGS data. The single fault
models of the two earthquakes with different focal mechanisms are plotted in Figures 15 and 16. The details
of the eight different sets of earthquake focal mechanisms are listed in Table 7.

In general, the simulated tsunami waveforms from all eight sets of earthquake focal mechanisms

matched the observed data well. Figure 17 shows the observed and simulated tsunami waveforms at the
Dongkung and Houbihu stations using the eight different sets of earthquake focal mechanisms. The
simulated tsunami waveform from the earthquake focal mechanisms of S3 (misfit = 0.530), S5 (misfit =
0.529), and S7 (misfit = 0.493) showed a better fit to the observations than did the other simulations (Table
7). Among them, the earthquake focal mechanisms of S7 were found to be the best fitting scenario with the
smallest misfit from the observations. Scenario S7 contained the fault orientations of NP2 for the first
earthquake and NP1 for the second earthquake from USGS's moment tensor solution (Figures 15d, 16c).

While the single fault models can produce simulated tsunami waveforms that are consistent with the

observations, the poorly sampled (i.e., 6 min interval) signals recorded at the coastal stations also raised
some questions, as one would expect some potential high tsunami waves behind the observed signals. To
that sense, overestimation of the modeled results was expected, but the simulated tsunami waveforms using
single fault models presented the opposite results. This indicates that the single fault models (i.e., with
uniform fault slip) may not be sufficient and that the asperity area (i.e., with a large fault slip) on the fault
should be evaluated. The tsunami sensitivity to asperity locations of multiple fault models are discussed in
the next section.

### 5.2 Tsunami sensitivity to uniform and non-uniform fault slip models

The sensitivity of simulated tsunami waveforms to non-uniform fault slip distribution (i.e., multiple fault
model) was evaluated based on the best fitting fault geometry of S7. The fault model with uniform fault
slip (i.e., single fault model) was also modeled to identify the significant differences in the modeled results
from the uniform and non-uniform slip fault models.
Figure 18 shows the observed and simulated tsunami waveforms at the Dongkung and Houbihu stations
using non-uniform slip models (LS1-LS9) and a uniform slip model (S7). At the Dongkung station, the
simulated tsunami waveforms from non-uniform slip models were not much different from those of the
uniform slip models. Both models could produce tsunami waveforms in good agreement with the observed
values recorded at this station. At the Houbihu station, the non-uniform slip models produced a significantly
higher first wave crest than the observations. The simulated wave peaks from the non-uniform slip models
produced wave heights approximately twice those simulated using the uniform slip. These results indicated
that the near-field station of Houbihu was rather sensitive to the effect of the fault slip distribution, and
some high tsunami waves might have been missing from the recorded signals at the Houbihu station during
the 2006 tsunami.

**5.3 Tsunami simulation using open-source bathymetric data**
To analyze the tsunami sensitivity on different sources of open-source, accessible bathymetry data,
numerical simulations were applied using GEBCO and ETOPO1 data. The differences between the modeled
results using these different bathymetry data were evaluated to compare the modeled wave peaks and
waveforms in the 2006 tsunami.
Figures 19a and 19b show the spatial distribution of the maximum wave heights simulated using two
bathymetric grids, the GEBCO data and ETOPO1 data. To evaluate the differences between the modeled
wave peaks, the variation and percent change in the variation were calculated, which can be defined in
equations (12) and (13):

$$Var_{peak} = Peak_{GEBCO} - Peak_{ETOPO1} \qquad (12)$$

$$\% \, Var_{peak} = \frac{Peak_{GEBCO} - Peak_{ETOPO1}}{Peak_{GEBCO}} \times 100 \qquad (13)$$

where $Var_{peak}$ is the variation in the modeled wave peaks calculated at each computational grid with
GEBCO and ETOPO1 data and $Peak_{GEBCO}$ and $Peak_{ETOPO1}$ are defined as the calculated wave peaks
of progressive waves in a unit area of the free surface. Figures 19c and 19d illustrate the spatial distribution
of the variation and percent change in the variation of the modeled wave peaks in the model domain,
indicating the differences in the modeled results using the two bathymetries. The results suggested that the
variation in the modeled wave peaks using the two bathymetries was greater than 0.05 m and the percent
change was greater than 50% between the modeled results for areas with sea depths of less than 500 m.
Figure 20 shows the modeled tsunami waveforms at the three coastal stations (i.e., black circles in Figure
19) using the two bathymetric grids. At Kaohsiung, the modeled waveforms from the two bathymetries
matched each other well; however, the modeled wave peak from the ETOPO1 data was significantly smaller
than that from the GEBCO data. The bathymetries from the GEBCO and ETOPO1 data could produce
tsunami waveforms at Dongkung and Houbihu that were similar in both wave periods and peaks. Table 8
summarizes the details of the coastal stations and the peak variation percentage of the modeled tsunami
waveforms from the two bathymetries.

**6. The mechanism of tsunami wave trapping**
**6.1 Bathymetry effect on tsunami wave directivity**
It is commonly understood that tsunami velocities are mainly governed by seafloor depths. A tsunami
propagates at a slower speed when the tsunami wave enters shallow water from deeper water. The
significant change in propagation speed allows the tsunami to change its wave direction. To assess the
bathymetry effect on tsunami wave directivity during propagation, simulations were applied using actual
(MS) and manipulated bathymetry experiments (EXP1 and EXP2).

Simulated snapshots of tsunami wave propagation using actual (MS) bathymetry data are shown in

Figure 21. The continental shelves in front of Hengchun Peninsula have shallow depths compared to the
open ocean. Figures 21a and b present how tsunami waves repeatedly changed their directions among the
shelves and then refracted into the west coast embayment. The tsunami waves were reflected from the coast
after arrival and tended to radiate offshore. However, they did not fully radiate offshore; instead, they were
reflected again at the boundary of the shelf and refracted north toward Kaohsiung and Dongkung (Figure
21c, d). The high-energy waves repeatedly reflected and refracted among the shelves. Only rare tsunamis
were transmitted back to the open ocean or to the east coast. These results indicated that the tsunami waves
were trapped over the shelves during their passage in the 2006 tsunami event. Due to this fluctuation, the
high-energy tsunami wave remained along the western coast for a long time, which could be clearly seen
at 75 min and 90 min after the occurrence of the first earthquake (Figure 21e, f).

Figure 22 shows snapshots of the simulated tsunami wave propagation using manipulated (EXP1)

bathymetry. In this situation, the transmission of tsunami waves in the shallow area was similar to those
simulated using the actual (MS) bathymetry, in which the tsunami waves were persistent and repeatedly
reflected and refracted among the shelves, but more reflected waves from the coast radiated to the open sea
(Figure 22b-f). This is because the tsunami source was located in an area with sea depths over 500 m, and
bathymetry data with sea depths over 500 m were replaced with a 500 m depth in this hypothetical situation.

Aside from the numerical experiment EXP1, a rather hypothetical situation (EXP2) was conducted to

simulate tsunami wave propagation on a bathymetry with a flat sea bottom and a sea depth of 500 m. Figure
23 shows snapshots of simulated tsunami wave propagation using the manipulated (EXP2) bathymetry. An
inspection of the tsunami wave transmission in the shallow area indicated that the reflected tsunami waves
from the coast radiated homogeneously offshore, and the wave reflection and refraction could not be clearly
seen. In addition, the tsunami waves propagated at a rather fast speed (i.e., in comparison to MS and EXP1)
and mostly radiated out of the model domain at 75 min and 90 min after the occurrence of the first
earthquake (Figure 23 d, e).

**6.2 Tsunami wave energy trapped on the shelf**

While the past section specified that tsunami waves are trapped over shelves due to the wave directivity

change associated with the configuration of coastal bathymetry, the question remains of how much wave
energy can be trapped over the shelves in front of southern Taiwan during the passage of tsunamis. To
quantitatively evaluate the wave energy trapped over the shelves, the trapped ratio was used to indicate the
tsunami energy trapped in bathymetric situations, as calculated in equation (14):

$$R_T = \frac{E_{Shelf}}{E_{Total}} \times 100 \qquad\qquad\qquad (14)$$

where $R_T$ is the ratio of tsunami energy trapped, $E_{Shelf}$ is the calculated tsunami potential energy on the
shelves (i.e., shallow areas with sea depths under 500 m), and $E_{Total}$ is the calculated total tsunami
potential energy of the model domain at each time step. The tsunami potential energy was determined
assuming that the energy flux of the tsunami wave progressed in a unit region of the free sea surface (Nosov
et al., 2014) and was determined using equation (15):

$$E_p = \oiint \frac{1}{2} \rho g \eta^2 \, dx \, dy \tag{15}$$

where $E_p$ is the tsunami potential energy, $\rho$ is the water density of the ocean, $g$ is the gravitational
acceleration (set as 9.81 m s$^{-2}$), and $\eta$ represents the surface integral of the ocean surface disturbance at
each time step. The ratio of trapped tsunami energy was calculated from the snapshots of tsunami
simulations using actual (MS) and manipulated (EXP1 and EXP2) bathymetry. Figure 24 shows the
calculated trapped ratio from simulated tsunami propagation snapshots every 15 min using actual (MS) and
manipulated (EXP1 and EXP2) bathymetry. Note that for calculating the trapped ratio from simulations
using manipulated bathymetry (EXP1 and EXP2), the shelf region corresponding to the actual bathymetry
(MS) was used (i.e., the shallow area illustrated by the solid and dashed black lines shown in Figures 22
and 23). According to equations (14) and (15), the simulations yielded a ratio of trapped tsunami energy of
more than 50% when using actual bathymetry (MS) and manipulated bathymetry (EXP1) but a smaller
trapped ratio of 20% when using manipulated bathymetry (EXP2). These results quantitatively provided
confirmation that the coastally trapped tsunami wave energy was related to the shape of the bathymetry.

**6.3 Comparison of simulated tsunami waveforms**
To understand any significant change in tsunami waveforms that can be recognized with and without
wave trapping, tsunami waveforms simulated from actual (MS) and manipulated bathymetry (EXP1 and
EXP2) were compared. Figure 25 shows the simulated tsunami waveforms at the three coastal stations in
southern Taiwan using actual and manipulated bathymetry.
Using the manipulated bathymetry (EXP1), the first few circles of simulated tsunami waveforms at all
the stations were consistent with those simulated using actual bathymetry (MS) but produced slightly
smaller later phase amplitudes. An inspection of the simulated waveforms using the manipulated
bathymetry (EXP2) indicated an earlier arrival time of the first wave and smaller amplitudes of the later
phase than those of the simulation results using actual (MS) bathymetry. These results indicated that the
persistent high-energy waves along the south coast of Taiwan were associated with the mechanism of
tsunami wave trapping.

**6.4 Amplified and persistent high-energy waves along the coast**
As described in the previous sections, the tsunami wave was trapped over the shelves and transmitted
along the coast as edge waves during the 2006 tsunami. This section describes how tsunami waves behave
as edge waves and to what extent such wave fluctuations influence the amplified and persisting high-energy
waves along the south coast of Taiwan. Figure 26 shows the shelves in front of south Taiwan and the
simulated tsunami heights of the 2006 tsunami from the main simulation (MS).
To study the behaviors of edge waves along the south coast during the 2006 tsunami, a time-distance
diagram of tsunami waves is shown. Figure 27a shows the time-distance diagram of the tsunami wave along
the contour of the 20 m sea depth (i.e., dashed black line in Figure 26a). Based on the phase shift of the
tsunami wave, the propagation path and the travel time curve of edge waves were illustrated (i.e., green
arrow in Figure 27a). According to the travel time curve, the edge waves propagated along the coast at a
speed of 50 m s$^{-1}$. The edge waves propagated along the coast and were iteratively reflected at the shelf
edge. The coupling of the edge waves and the later-arriving incident waves amplified the tsunami waves
and maintained the wave oscillation in the later phase. These were visible from simulated tsunami
waveforms at numerical wave gauges C and E, as shown in Figure 27c.
To understand the persisting high-energy waves along the south coast of Taiwan during the 2006 tsunami,
the decreasing tendency of the tsunami wave energy along the 20 m sea depth contour was analyzed. The
temporal tsunami wave energy was first determined using equation (11) and then normalized according to
the maximum temporal tsunami energy in the time series. Figure 27b shows the time-distance diagram of
the normalized tsunami energy along the 20 m sea depth contour (i.e., dashed black line in Figure 26a).
Figure 27d shows the normalized tsunami energy at numerical wave gauges C and E. At the numerical wave
gauge C, the normalized tsunami energy achieved its greatest value at approximately 40 min after the first
earthquake occurred. However, this high-energy channel did not decrease with time after the first wave
arrived; instead, a persisting channel of strong energy was visible. This energy channel lasted for more than
60 min, and the wave energy repeatedly reached the maximum value in this channel. Beyond this channel,
the energy commenced to decrease with a rate of energy loss of 50% at 110 min and 20% at 270 min after
the occurrence time of the first earthquake. At the numerical wave gauge E, the normalized tsunami energy
achieved its greatest value approximately 30 min and 120 min after the first wave arrived. Beyond this
channel, the energy commenced to decrease at a rather fast rate of energy loss of 80% at 150 min and 70%
at 215 min after the occurrence time of the first earthquake. Accordingly, the tsunami decay process in this
region was expected to last for more than 300 min. These results indicated that the wave amplification and
persistent high-energy waves along the coast during the 2006 tsunami were connected to tsunami wave
trapping and the influence of edge waves. According to these behaviors, southern Taiwan could be affected
by intensified coastal hazards and severe impacts from tsunamis.

**7. Conclusions**
**7.1 Main findings**
In this article, the characteristics of the consecutive tsunamis on 26 December 2006 and the resulting
tsunami behaviors in southern Taiwan were investigated and clarified. The methodology comprised

analyses of tide gauge tsunami waveforms, spectral analyses and numerical tsunami simulations. The main findings are summarized as follows:

(1) The physical characteristics of the tsunami waveforms at all three tide gauge stations in southern Taiwan during the December 2006 tsunami were analyzed. The initial tsunami wave arrived at Kaohsiung, Dongkung, and Houbihu at 21:18, 20:54, and 20:42 Taiwan Standard Time, respectively, with a trough sign of tsunami amplitude. Following the initial wave trough, the initial wave crests were 0.07 m (Kaohsiung), 0.09 m (Dongkung), and 0.3 m (Houbihu). The maximum tsunami wave heights at the three tide gauge stations at Kaohsiung, Dongkung, and Houbihu were 0.08 m, 0.12 m, and 0.3 m, respectively, and the maximum tsunami wave heights at Kaohsiung and Dongkung were not recorded with the first arrivals. The approximate tsunami duration in Dongkung was 3.9 h, while the tsunami lasted for more than 6 h in Kaohsiung and Houbihu.

(2) Based on the spectral analyses of tsunami waveforms, a period band of 13.6-23.1 min was attributed to the tsunami source spectrum. The periods of 16 min and 36 min were considered as the modes of trapped tsunami waves resonating with the fundamental modes of the shelves.

(3) A tsunami source model for the 2006 earthquake doublet tsunami was proposed. The fault size of the successive earthquakes was estimated to be 800 km$^2$, comprising a length of 40 km and a width of 20 km. Uniform slips of 1.66 m (first earthquake, $M_w$ 7.0) and 1.17 m (second earthquake, $M_w$ 6.9) were estimated. The respective central fault depths of the two earthquakes were 20 km and 33 km. The focal mechanisms of the first earthquake, with a strike of 319°, dip of 69°, and rake of -102°, and the second earthquake, with a strike of 151°, dip of 48°, and rake of 0°, could successfully produce the observed tsunami waveforms. Moreover, the tsunami sensitivity of the non-uniform fault slip distribution indicated that some tsunami signals might have been missing from the record signals due to the poor sampling rate (6 min intervals), and the wave peaks at Houbihu station might have reached twice the values of those observed during the 2006 tsunami.

(4) A comparison of tsunami propagation simulations using actual (MS) and manipulated bathymetry (EXP1 and EXP2) revealed that the tsunami waves were coastally trapped during the passage of the 2006 tsunami. The trapped tsunami waves iteratively reflected and refracted among the shelves. The trapped waves interfered with incident waves and resonated with the fundamental modes of the shelves, resulting in an amplified and persistent oscillation of tsunami waves. This explained why the observed tsunami waves recorded at some stations in southern Taiwan were amplified and had a tsunami duration of more than 6 h during the 2006 tsunami.

(5) Tsunamis are one of the most dangerous coastal hazards and can cause destructive damage and loss of life in coastal regions. Taiwan is at risk of tsunamis and is exposed to potential near-field tsunamis generated from the Manila Trench on the South China Sea (SCS) side and the Ryukyu Trench on the Pacific Ocean side. The results of the present study based on the 2006 tsunami revealed that the tsunami wave was trapped over the insular shelves around southern Taiwan during the passage of the tsunami.

Wave couplings and resonant features might result in unexpected amplification of tsunami heights and
persistent wave oscillation in southern Taiwan. In other words, even if the initial wave heights are small,
the tsunami waves that arrive later are expected to be higher and more persistent along the coast of
southern Taiwan. Therefore, decision makers and people in southern Taiwan should be aware of this
possibility and stay clear of coastal regions for a long time as an emergency response to future tsunamis,
even if the wave height of an initially arriving tsunami wave is small. These findings are important and
valuable for improving the existing system of tsunami warnings and coastal planning for disaster risk
management.

**7.2 Limitations and future improvements**
In this study, the characteristics of the December 2006 tsunami and resulting tsunami behaviors in
southern Taiwan were explored using available data from tide gauge tsunami waveforms and numerical
tsunami simulations. Nevertheless, the analyses in this article had some limitations. The first limitation was
related to the tsunami data recorded at the tide gauge stations, which were employed as input data for the
spectral analyses (i.e., Fourier analyses and wavelet analyses) and compared with the numerical results.
The sampling interval of the tide gauge data recorded at all CWB tide gauge stations was 6 min, indicating
that tsunami wave components with shorter periods might not be well recorded in the tide gauge data. Due
to this existing limitation, spectral analyses might cause discrepancies in detecting periodic components of
tsunami spectra. This limitation could be improved by including tsunami data with more frequent sampling
rates.
Another limitation was related to the simulation grid size (i.e., 450 m) for the tsunami propagation
simulation. Although the simulated tsunami waveforms were reasonably consistent with the observed
values recorded at the tide gauge stations in terms of wave amplitude and arrival time, the reproducibility
of the numerical results for the 2006 tsunami could be further improved by constructing a finer grid of
bathymetric data.
All the limitations mentioned above suggest further improvements to research to provide a more detailed
investigation of long-lived edge wave and shelf resonance issues, especially in the region of southern
Taiwan. In addition, more fundamental studies on the complex wave mechanisms of tsunami reflection and
refraction, shoaling effects, and wave trapping by insular shelves are planned for future work.

*Code availability.* The second version of TUNAMI code (TUNAMI-N2) conducted in this research is
currently not open-source model but is available from the corresponding author upon reasonable request.

*Data availability.* The record sea level data at tide gauge stations were obtained from Central Weather
Bureau, R.O.C. through reasonable request. The seismic information is available in publicly accessible
catalogs of the GCMT Project and USGS, as mentioned in the body of the article. The GEBCO and
ETOPO1 bathymetric data used for the numerical tsunami simulations are publicly assessable at
https://www.gebco.net (General Bathymetric Chart of the Ocean, 2021) and
https://www.ngdc.noaa.gov/mgg/global (National Oceanic and Atmospheric Administration, 2009).

*Author contributions.* All authors read, reviewed, and approved the manuscript. C.A.C wrote the manuscript,
performed numerical simulation, and analyzed the results. F.I. and A.S. supervised the research and
collected the funding. K.P. provide constructive suggestions to the study and numerical simulation.

*Competing interests.* All authors declare no competing interest.

*Acknowledgments.* The authors thank Ms. Shyh-Fang Liu and Ms. Cheng-Lin Huang for their great support
in collecting the observation data used in this work. The authors would like to express their sincere
appreciation to Dr. Yo Fukutani from Kanto Gakuin University for his valuable suggestions on conducting
the sensitivity analysis of the fault models. We appreciated Andy Brandt, PhD, from group of American
Journal Expert (https://www.aje.com) for editing the draft of the manuscript. This work was supported (in
part) by the MEXT WISE Program for Sustainability in the Dynamic Earth.

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

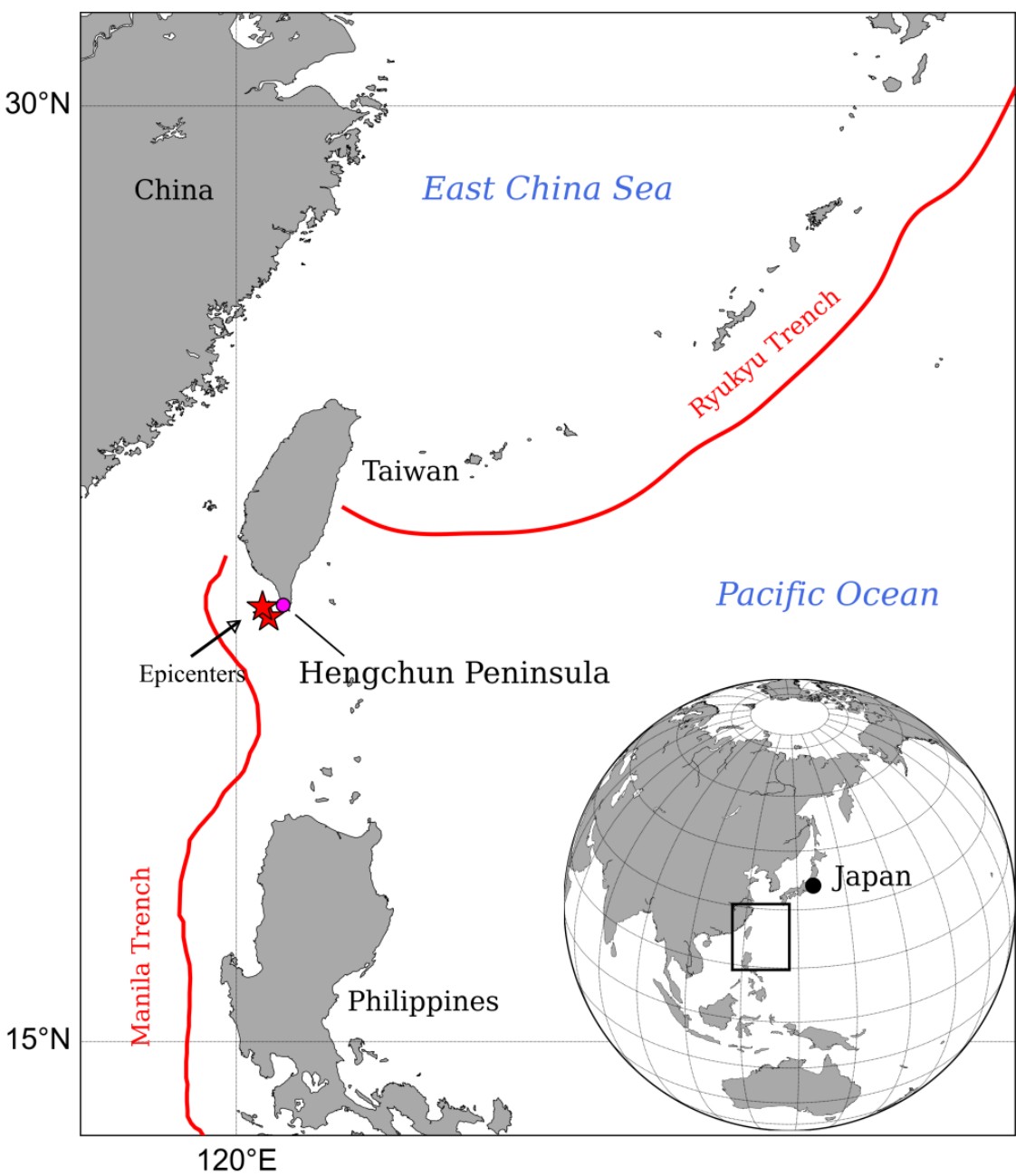


**Figure 1. Map of the Hengchun Peninsula, Taiwan. The red stars illustrate the epicenters of the doublet earthquakes, and the solid red lines illustrate the subduction zones of the Manila Trench and the Ryukyu Trench.**


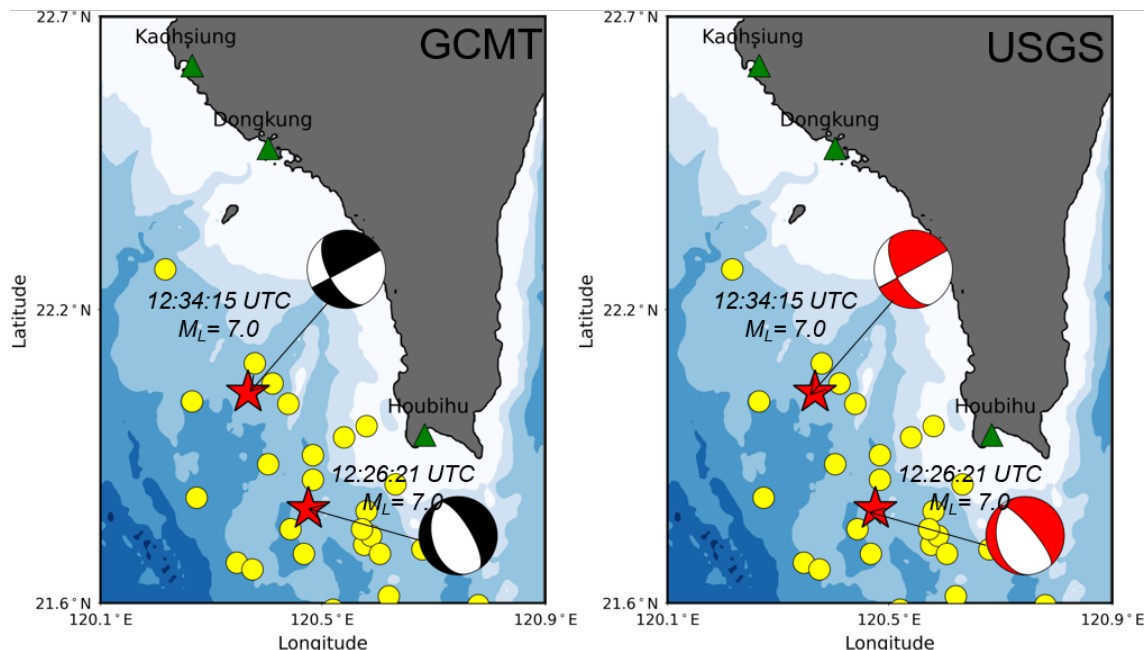


**Figure 2. The tectonic settings of the 2006 earthquake doublet. The red stars denote the**
**epicenters of the successive earthquakes. The beachballs denote the focal mechanisms of the two**
**earthquakes estimated from the GCMT and USGS moment tensor solutions. The yellow circles**
**show the aftershock distribution for one day from the USGS earthquake catalog. The green**
**triangles represent the locations of the CWB tide gauge stations.**

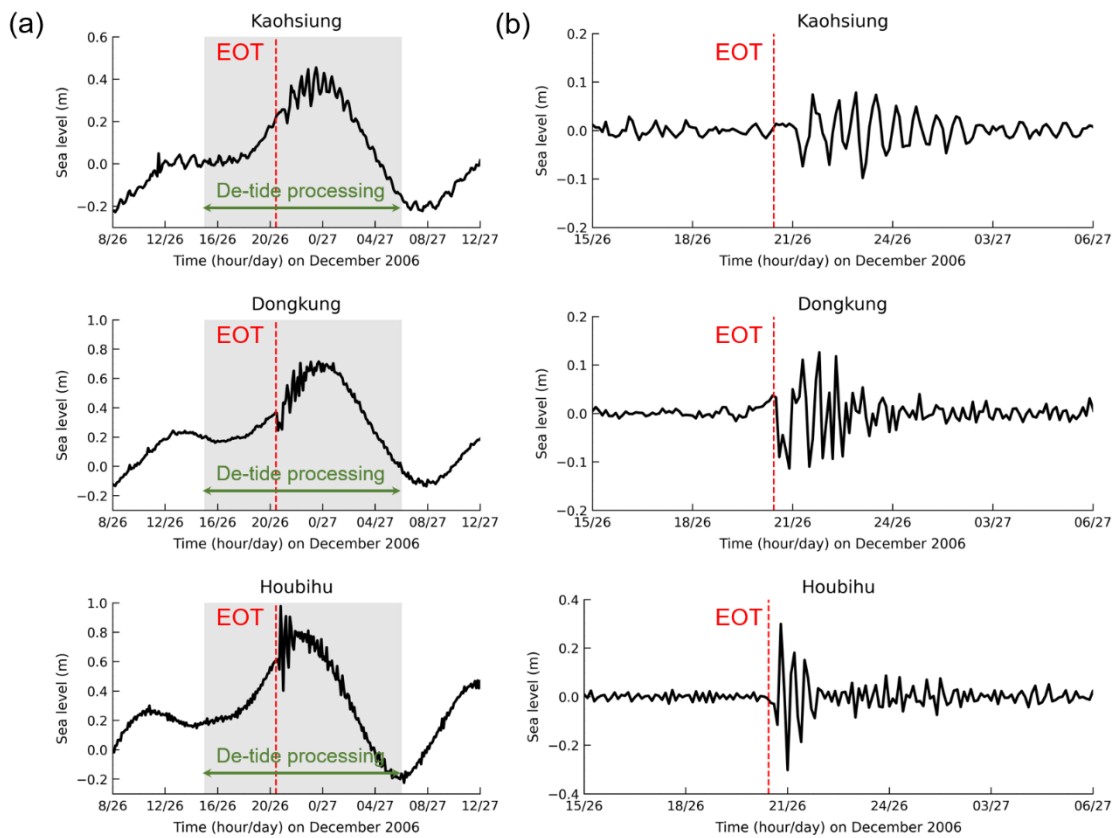


**Figure 3. The (a) original and (b) de-tided sea levels recorded at tide gauge stations in southern**
**Taiwan during the 26 December 2006 tsunami event. The vertical, dashed red lines indicate the**
**earthquake occurrence time (EOT). The gray shaded areas illustrate the tide gauge data used**
**for de-tide processing. The data shown in the graphs were drawn based on Taiwan Standard**
**Time.**


**Table 1. Calculation conditions for the numerical tsunami simulation.**

|  | Calculation condition for the numerical tsunami simulation |
| --- | --- |
| Governing equation | Two-dimensional nonlinear shallow water equations (TUNAMI-N2 model) |
| Numerical integration method | Leap-frog finite difference method |
| Initial condition | Initial water level calculated form fault parameters using the theory of Okada, 1985 considering the contribution of horizontal coseismic displacement |
| Coordination system | Cartesian coordinate system |
| Boundary condition | Radiation boundary condition |
| Stability criterion | Courant–Friedrichs–Lewy (CFL) condition |
| Time interval | 0.1 s |
| Mesh size | 450 m |
| Mesh number (x, y) | (538, 631) |



**Table 2. Focal mechanisms for successive earthquakes estimated by GCMT and USGS.**

|  |  | Earthquake 1 | | Earthquake 2 | |
|  |  | NP1 | NP2 | NP1 | NP2 |
| --- | --- | --- | --- | --- | --- |
| GCMT | Long (° E) | 120.52 | | 120.4 | |
|  | Lat (° N) | 21.81 | | 22.02 | |
|  | Strike (deg) | 165 | 329 | 151 | 61 |
|  | Dip (deg) | 30 | 61 | 48 | 90 |
|  | Rake (deg) | -76 | -98 | 0 | 138 |
|  | Depth (km) | 20 | | 33 | |
| USGS | Long (° E) | 120.55 | | 120.49 | |
|  | Lat (° N) | 21.8 | | 21.97 | |
|  | Strike (deg) | 171 | 319 | 151 | 61 |
|  | Dip (deg) | 24 | 69 | 48 | 90 |
|  | Rake (deg) | -61 | -102 | 0 | 138 |
|  | Depth (km) | 25 | | 33 | |


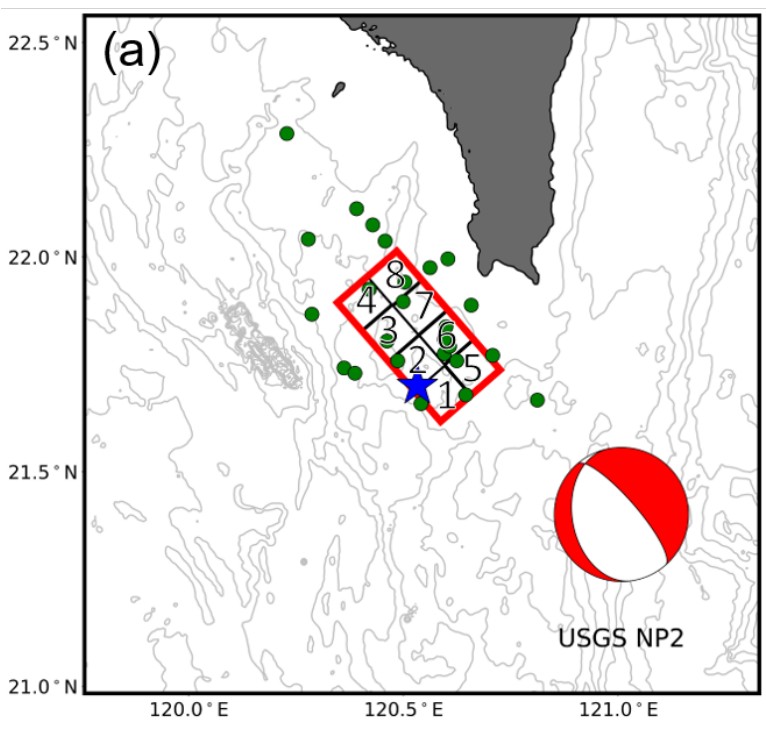

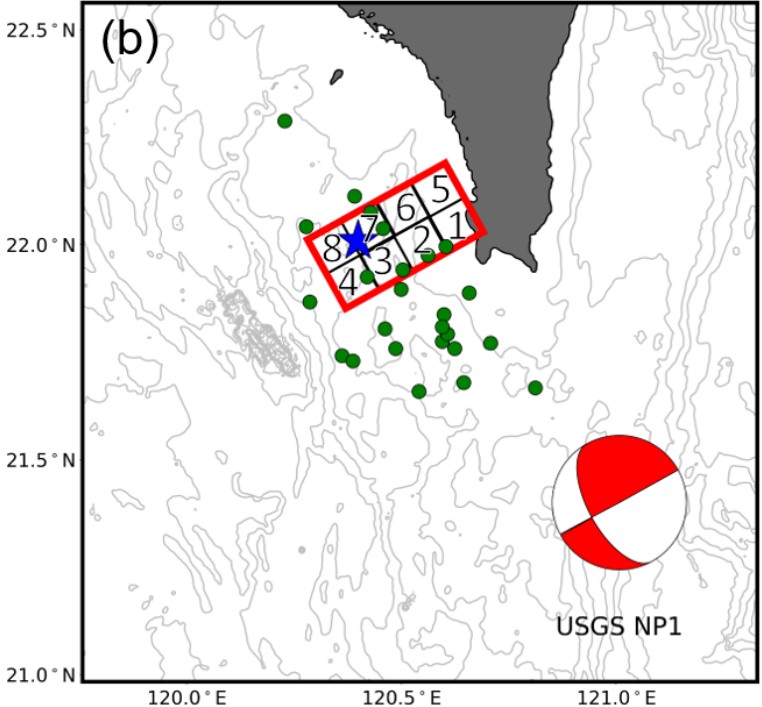


**Figure 4. Fault models for the two earthquakes. (a) Subfault locations of the first earthquake**
**($M_w$ 7.0) using NP2 of USGS's moment tensor solution. (b) Subfault locations of the second**
**earthquake ($M_w$ 6.9) using NP1 of USGS's moment tensor solution.**

 **Table 3. Parameters of the subfaults for the two earthquakes of the 2006 earthquake doublet.**

| | Sub fault | Long (°E) | Lat (°N) | Length (km) | Width (km) | Depth (km) | Strike (°) | Dip (°) | Rake (°) |
|---|---|---|---|---|---|---|---|---|---|
| Earthquake 1 | 1 | 120.619 | 21.588 | 10 | 10 | 15.3 | 319 | 69 | -102 |
| | 2 | 120.556 | 21.657 | 10 | 10 | 15.3 | 319 | 69 | -102 |
| | 3 | 120.492 | 21.724 | 10 | 10 | 15.3 | 319 | 69 | -102 |
| | 4 | 120.429 | 21.792 | 10 | 10 | 15.3 | 319 | 69 | -102 |
| | 5 | 120.692 | 21.648 | 10 | 10 | 24.7 | 319 | 69 | -102 |
| | 6 | 120.629 | 21.716 | 10 | 10 | 24.7 | 319 | 69 | -102 |
| | 7 | 120.565 | 21.784 | 10 | 10 | 24.7 | 319 | 69 | -102 |
| | 8 | 120.501 | 21.852 | 10 | 10 | 24.7 | 319 | 69 | -102 |
| Earthquake 2 | 1 | 120.726 | 21.989 | 10 | 10 | 29.1 | 151 | 48 | 0 |
| | 2 | 120.642 | 21.946 | 10 | 10 | 29.1 | 151 | 48 | 0 |
| | 3 | 120.557 | 21.902 | 10 | 10 | 29.1 | 151 | 48 | 0 |
| | 4 | 120.473 | 21.858 | 10 | 10 | 29.1 | 151 | 48 | 0 |
| | 5 | 120.680 | 22.068 | 10 | 10 | 29.1 | 151 | 48 | 0 |
| | 6 | 120.595 | 22.024 | 10 | 10 | 36.5 | 151 | 48 | 0 |
| | 7 | 120.510 | 21.980 | 10 | 10 | 36.5 | 151 | 48 | 0 |
| | 8 | 120.426 | 21.936 | 10 | 10 | 36.5 | 151 | 48 | 0 |



**Table 4a. Details of the average slip, large slip, and background slip for the two earthquakes.**

|  | Earthquake 1 | Earthquake 2 |
|---|---|---|
| Moment magnitude ($M_w$) | 7.0 | 6.9 |
| Entire fault size (km$^2$) | 800 | 800 |
| Rigidity (GPa) | 30 | 30 |
| Average slip $D_a$ (m) | 1.66 | 1.17 |
| Large slip $2D_a$ (m) | 3.32 | 2.35 |
| Background slip (m) | 1.11 | 0.78 |


**Table 4b. Asperity locations of multiple fault models for the two earthquakes.**

| Scenario | Asperity location of Earthquake 1 | | | Asperity location of Earthquake 2 | | |
|---|---|---|---|---|---|---|
| | North | Central | South | North | Central | South |
| LS1 | ○ | | | | ○ | |
| LS2 | | ○ | | | ○ | |
| LS3 | | | ○ | | ○ | |
| LS4 | ○ | | | ○ | | |
| LS5 | | ○ | | ○ | | |
| LS6 | | | ○ | ○ | | |
| LS7 | ○ | | | | | ○ |
| LS8 | | ○ | | | | ○ |
| LS9 | | | ○ | | | ○ |



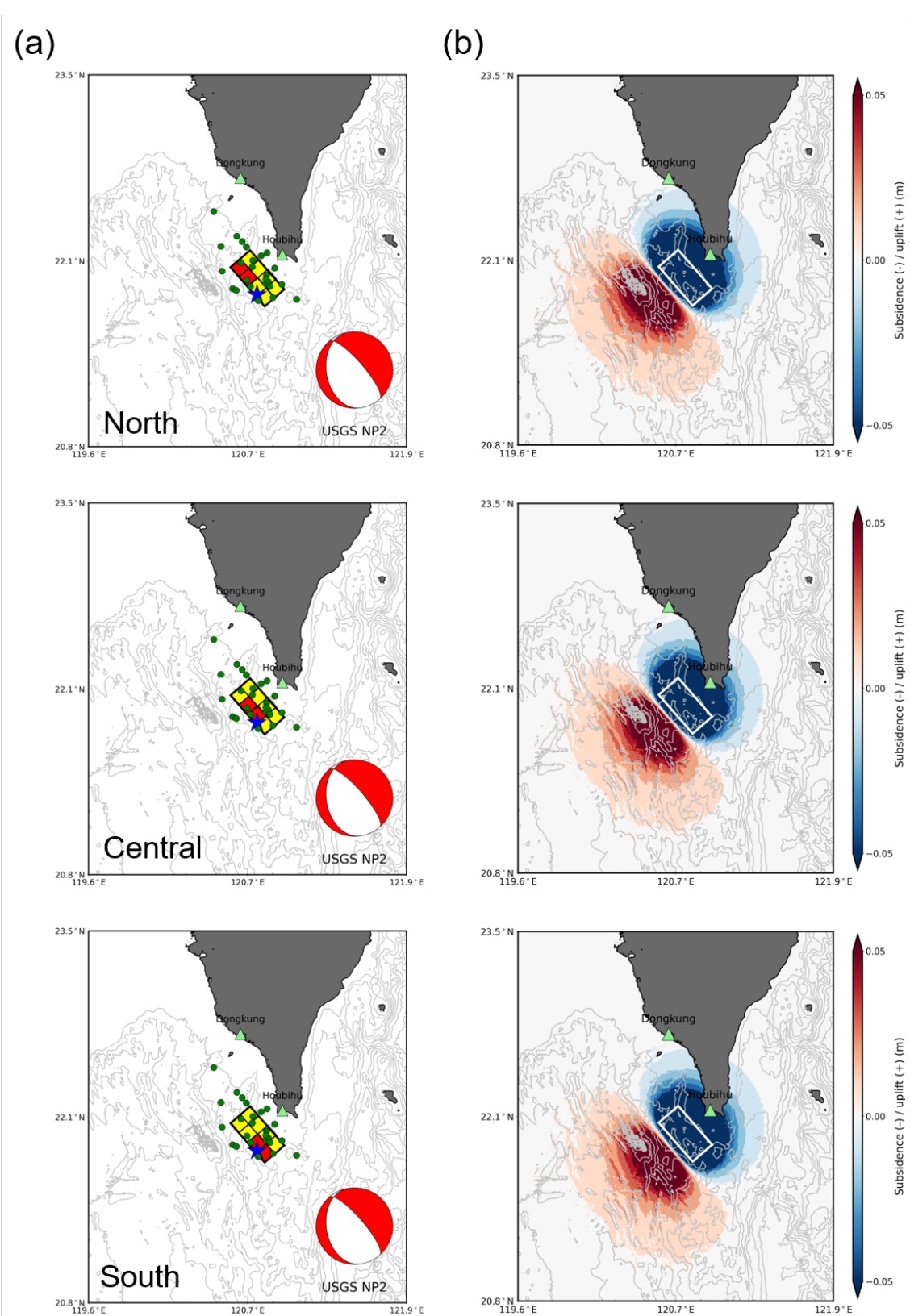

**Figure 5. (a) Map of subfault boundaries with different asperity locations for the first earthquake (M_w 7.0). (b) Coseismic crustal vertical displacement calculated using the fault**

parameters of the subfaults. The beachball denotes the focal mechanisms of USGS's NP2 nodal
planes for the first earthquake. The subfaults in red represent large slip areas, and the subfaults
in yellow represent background slip areas. The large slip area was located only at the shallow
part of the entire fault area. The blue stars represent the epicenter of the first earthquake, and
the green circles represent the aftershocks. The tide gauge stations are plotted as green triangles.

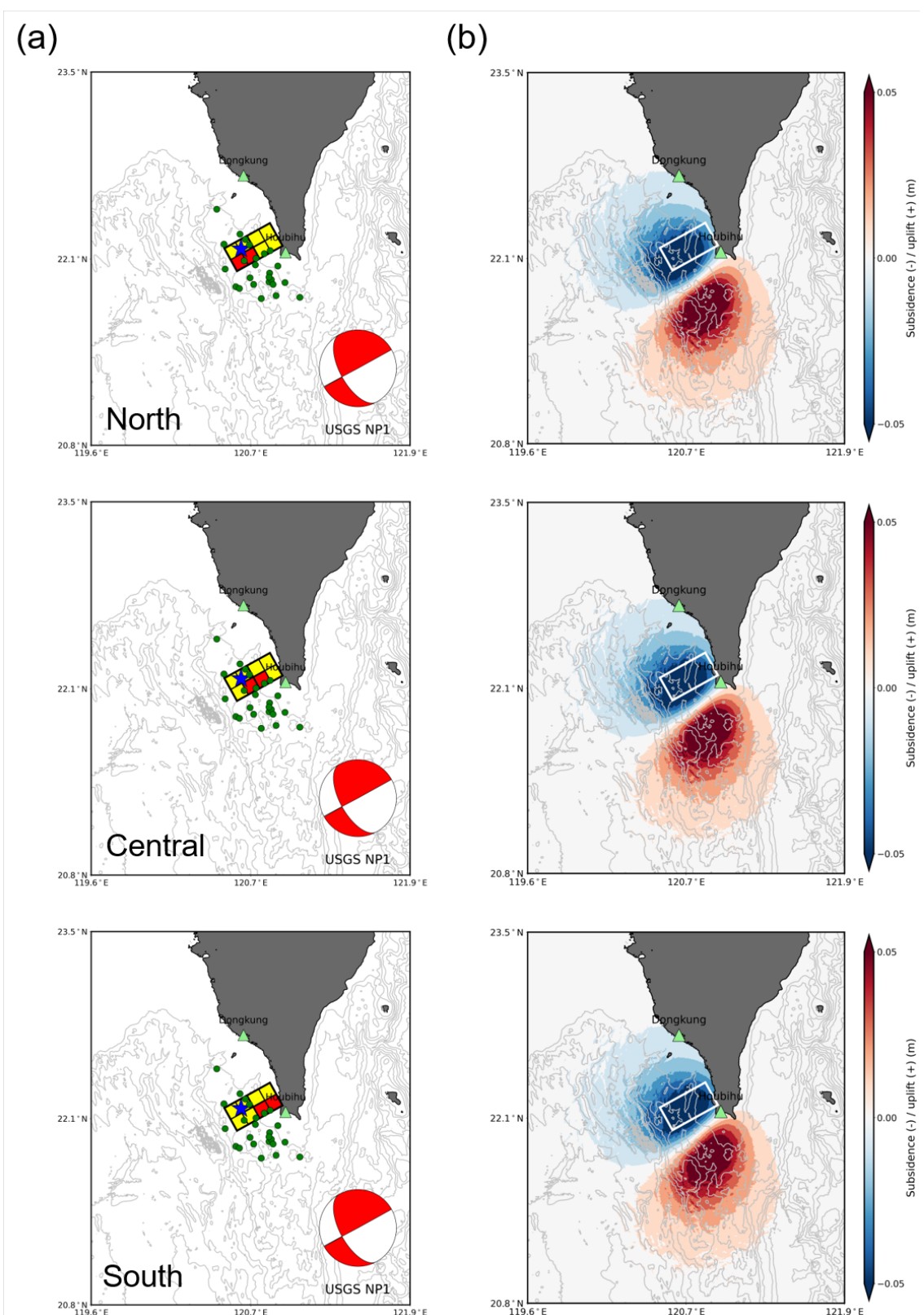

**Figure 6. (a) Map of subfault boundaries with three different locations of large slip areas for the**
**second earthquake (M_w 6.9). (b) Coseismic crustal vertical displacement calculated using the**
**fault parameters of the subfaults. The beachball denotes the focal mechanisms of USGS's NP2**
**nodal planes for the first earthquake. The subfaults in red represent large slip areas, and the**
**subfaults in yellow represent background slip areas. The large slip area was located only at the**
**shallow part of the entire fault area. The blue stars represent the epicenter of the first earthquake,**
**and the green circles represent the aftershocks. The tide gauge stations are plotted as green**
**triangles.**

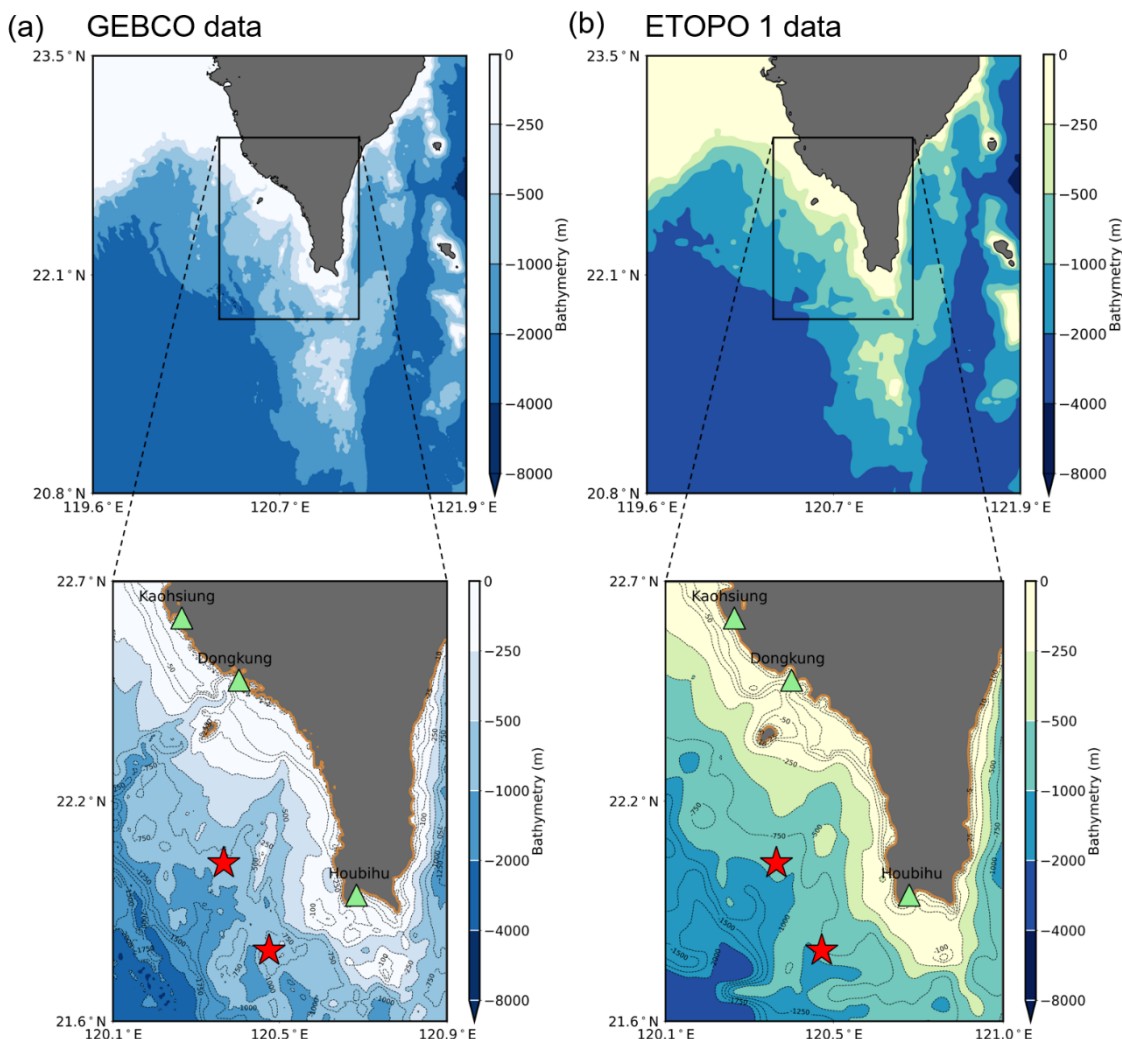

**Figure 7. Bathymetry map of the model domain from GEBCO and ETOPO1 bathymetry data.**
**The green triangles denote the locations of the tide gauge stations. The red stars represent the**
**epicenters of the two earthquakes.**

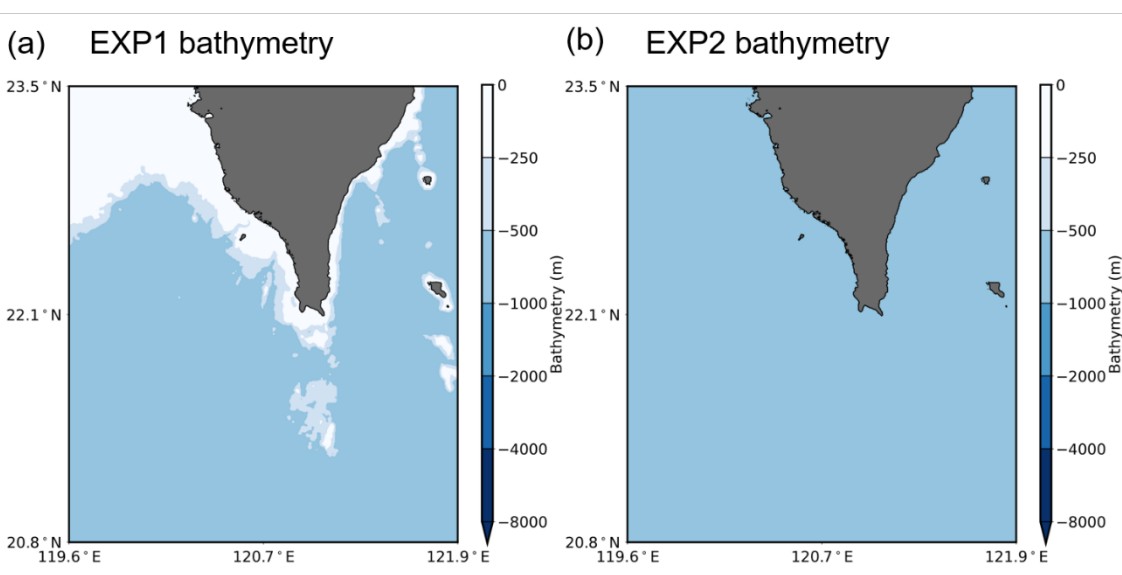


**Figure 8. Maps of the manipulated bathymetry of the model domain for numerical experiments**

**(a) EXP1 and (b) EXP2.**


**Table 5. Details of the bathymetry data used for the numerical experiments MS, EXP1, and**
**EXP2.**

| | Numerical experiments | | |
|---|---|---|---|
| | MS | EXP1 | EXP2 |
| Bathymetry source | | GEBCO data | |
| Grid size | | 450 m | |
| Mesh number (x, y) | | (538, 631) | |
| Description of bathymetry conditions | Sea depths from GEBCO data | Sea depths larger than 500 m were replaced with 500 m depths | Sea depths of entire domain were replaced with 500 m depths. |



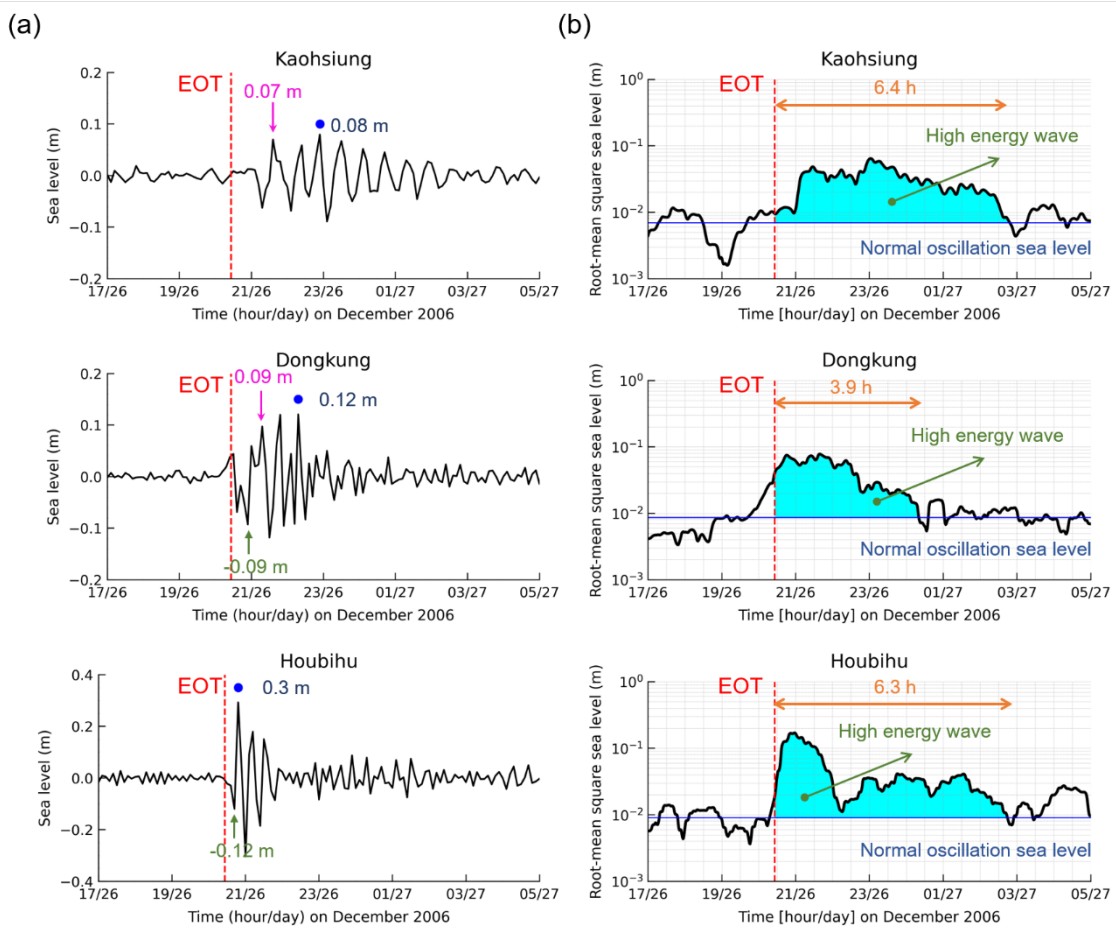

**Figure 9. (a) The observed tsunami waveforms and (b) diagrams of root mean square (RMS) sea**
**levels of the 2006 tsunami at the Kaohsiung, Dongkung, and Houbihu tide gauge stations. The**
**vertical, dashed red lines indicate the earthquake occurrence time (EOT). The blue circles**
**denote the arrival of the maximum crest wave that was recorded at all sites. The pink arrows**
**mark the first wave crest. The green arrows represent the trough sign of the first wave arrival.**
**The blue solid lines represent the normal oscillation sea level before the tsunami arrived (i.e.,**
**the mean value of sea level before the earthquake occurrence). The high-energy wave is**
**illustrated in cyan blue-shaded areas. The orange arrows show the elapsed time of tsunami**
**duration.**

**Table 6a. Details of the tide gauge stations and physical characteristics of tsunami waveforms**
**during the 2006 tsunami.**

| Station | Longitude (°E) | Latitude (°N) | Tsunami wave amplitude (m) | | |
|---|---|---|---|---|---|
| | | | First trough sign | First wave crest | Maximum wave crest |
| Kaohsiung | 120.28 | 22.61 | -0.06 | 0.07 | 0.08 |
| Dongkung | 120.43 | 22.46 | -0.09 | 0.09 | 0.12 |
| Houbihu | 120.74 | 21.94 | -0.12 | 0.3 | 0.3 |


**Table 6b. Details of the tide gauge stations and physical characteristics of tsunami waveforms**
**during the 2006 tsunami.**

| Station | Arrival time (Taiwan Standard Time) | | | Delay of maximum wave crest (min) | Visible wave period (min) |
|---|---|---|---|---|---|
| | First trough sign | First wave crest | Maximum wave crest | | |
| Kaohsiung | 21:18 | 21:44 | 22:54 | 70 | 30-48 |
| Dongkung | 20:54 | 21:18 | 22:18 | 60 | 18-24 |
| Houbihu | 20:42 | 20:48 | 20:48 | 0 | 18-24 |



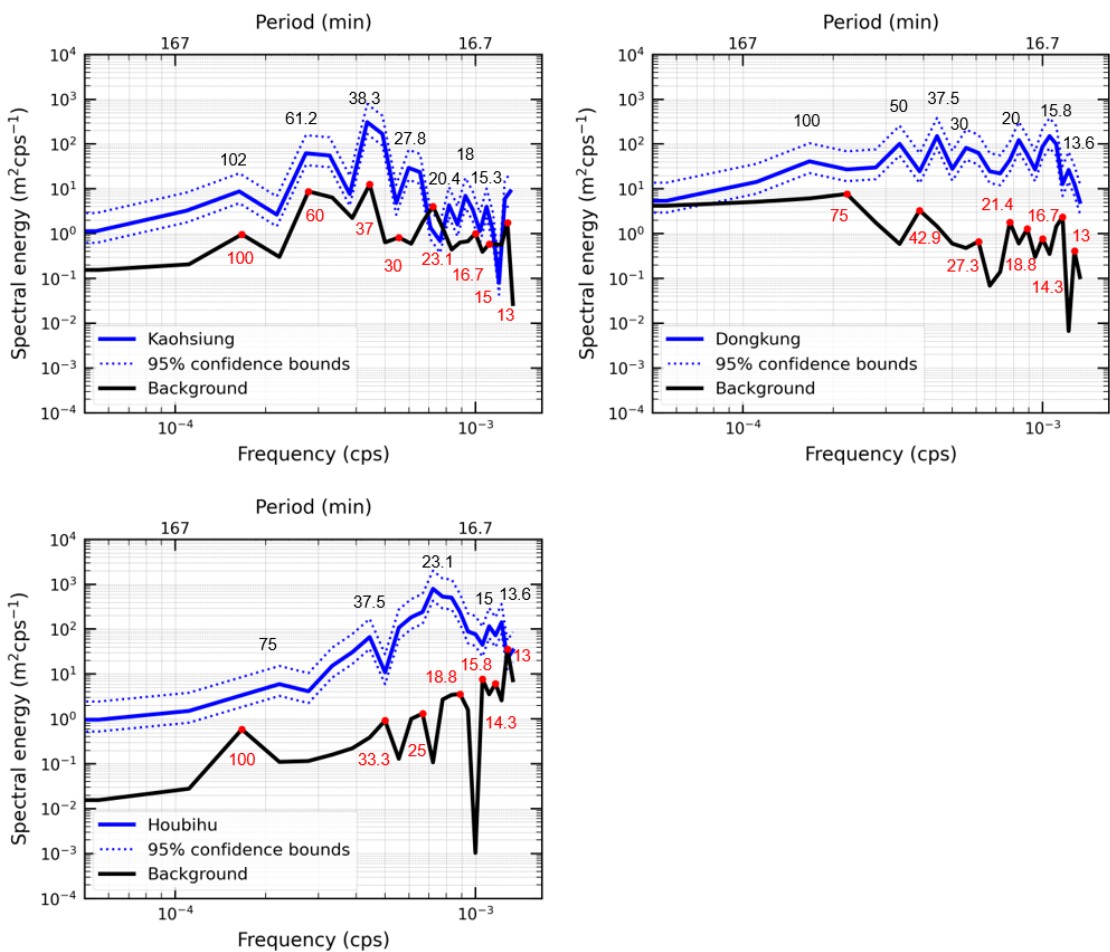


**Figure 10. Respective spectra of the observed tsunami waveform (solid blue lines) at each tide gauge station. The solid black lines are spectra for the background signals before tsunami arrivals at each station. The red circles denote the dominant periods of the background spectra.**


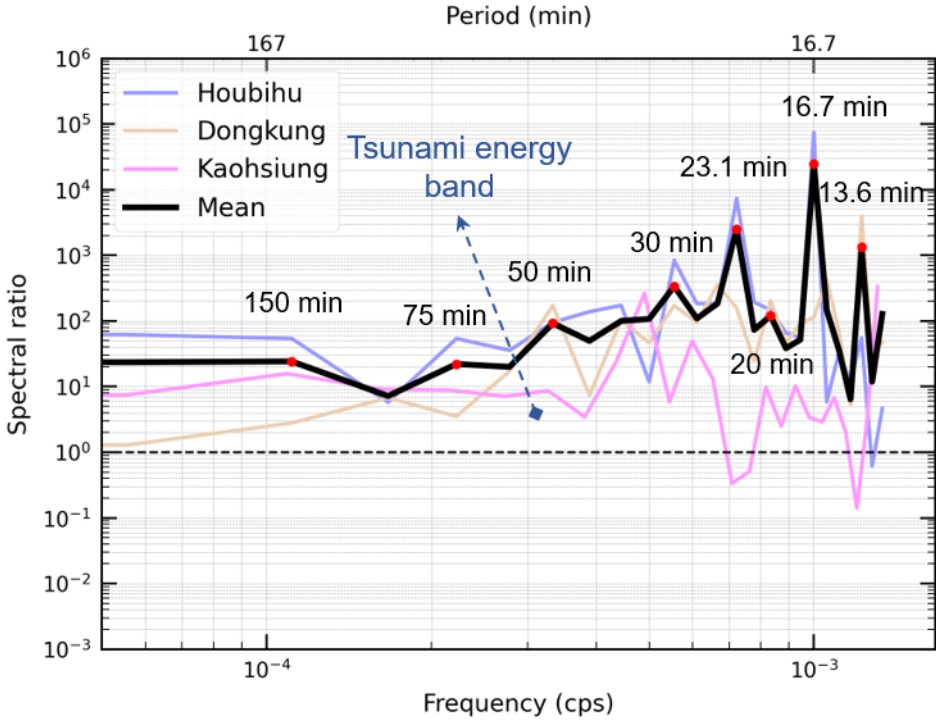


**Figure 11. Respective spectral ratio for the tide gauge spectra. The solid black line is the**
**calculated mean spectral ratio of the three tide gauge spectra. The red circles represent the**
**dominant periods of the mean spectral ratio.**

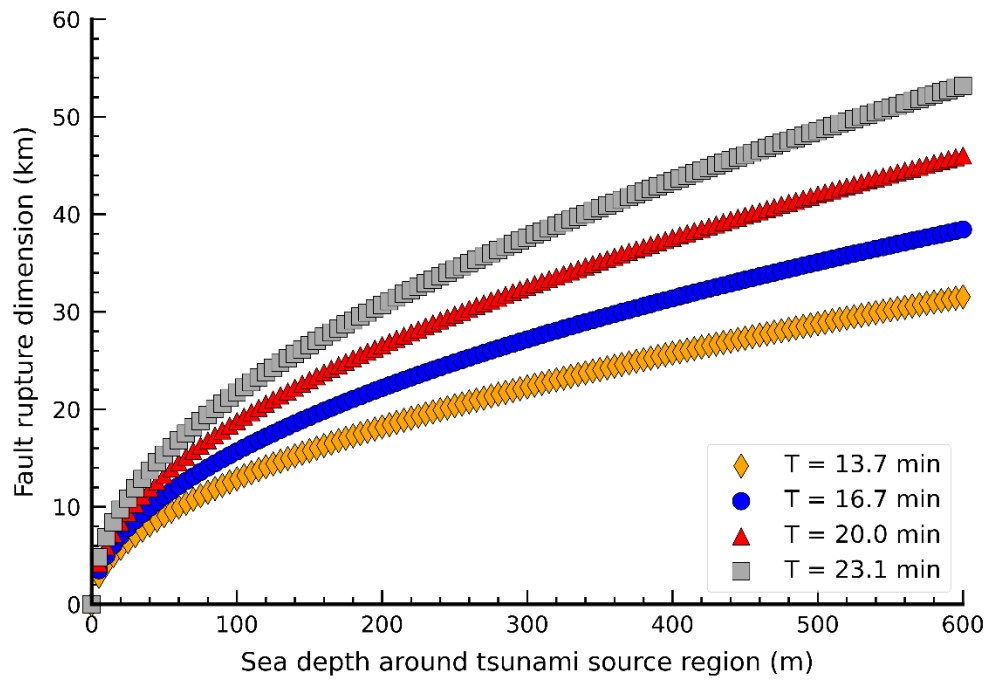


**Figure 12. Correlation of earthquake fault dimensions and sea depth around the tsunami source region derived from the empirical formula proposed by Rabinovich (1997).**




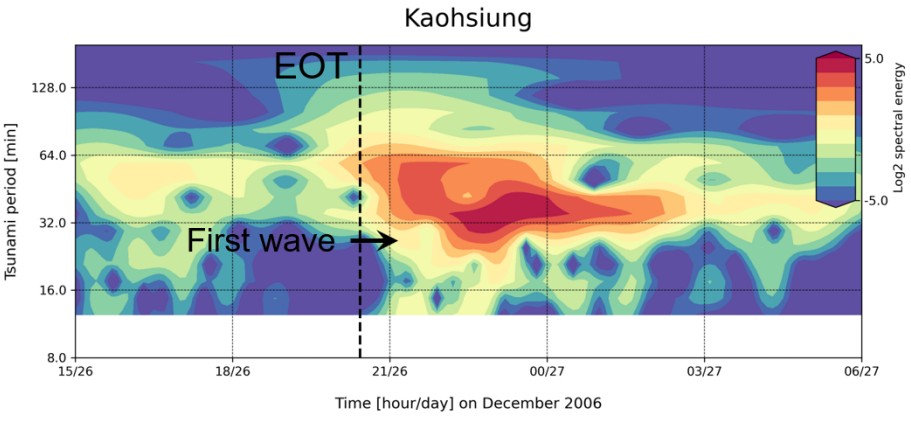

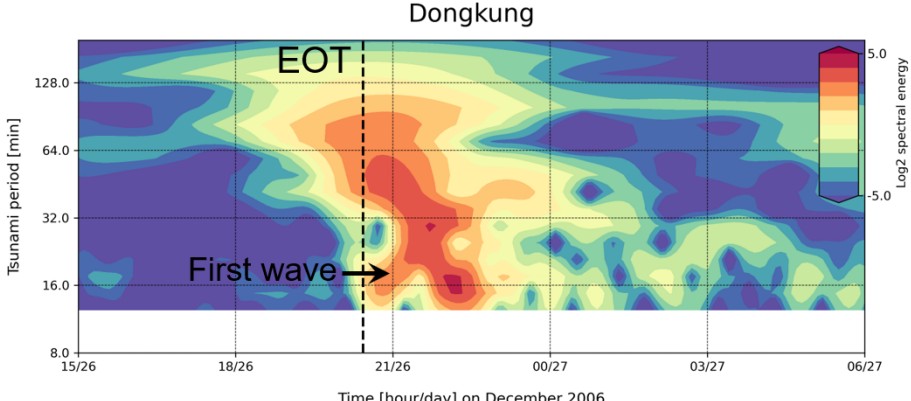

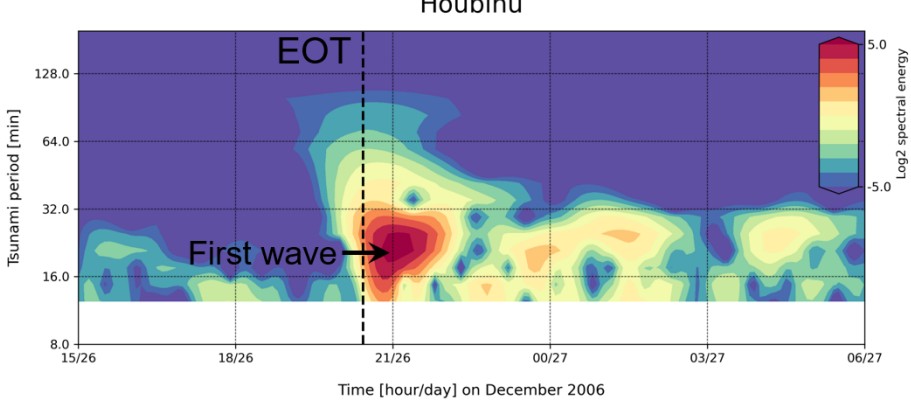


**Figure 13. Wavelet (time-frequency) diagrams of tsunami data for the 26 December 2006 tsunami event at the (a) Kaohsiung, (b) Dongkung, and (c) Houbihu tide gauge stations. The colormap represents the log2 spectral energy at various times and tsunami periods. The vertical, dashed black lines indicate the earthquake occurrence time (EOT). The black arrows denote the arrival time of the first tsunami wave at each station.**

947

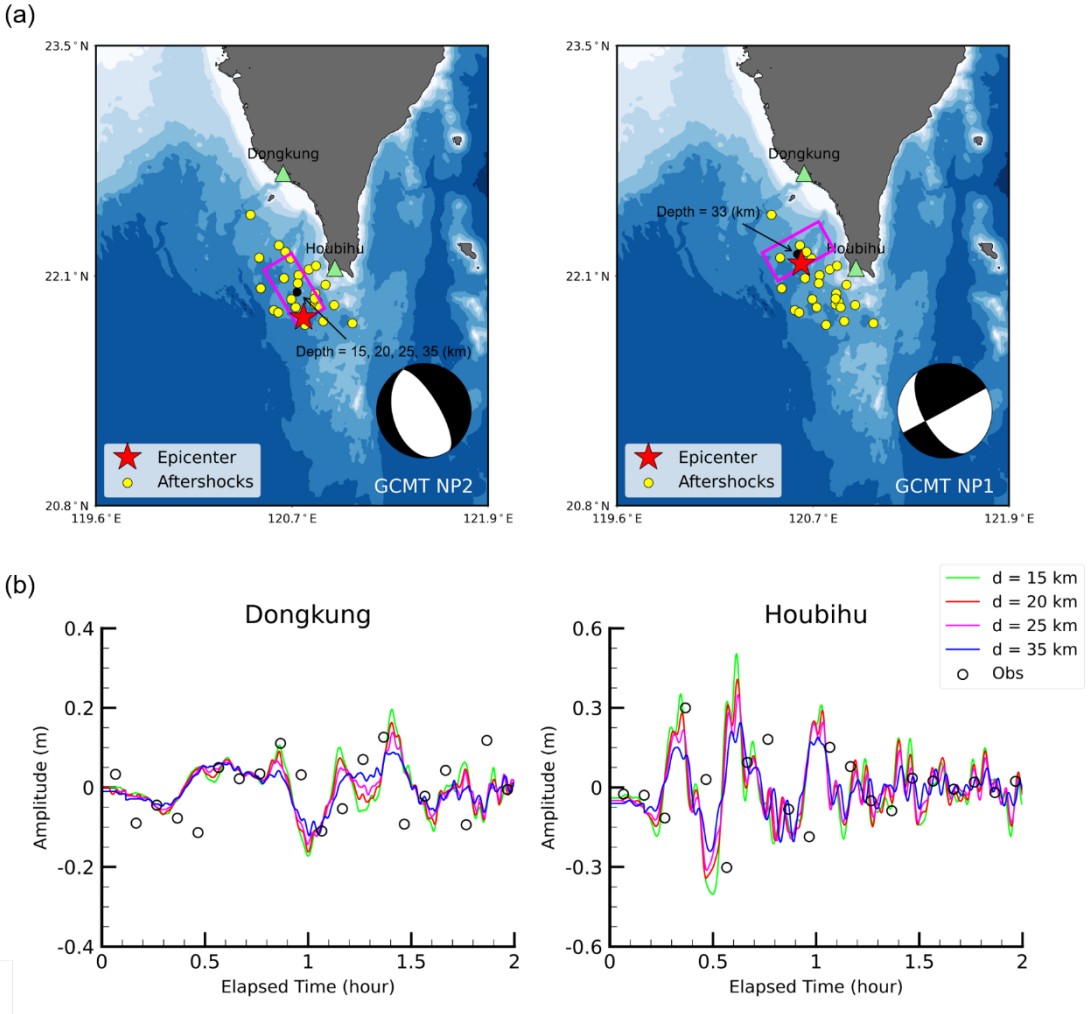

948

**Figure 14. (a) Single fault models with fault dimensions (length × width) of 40 km × 20 km of the first earthquake using the GCMT NP2 nodal plane and the second earthquake using the GCMT NP1 nodal plane. The central fault depths of the single fault models for the first earthquake are set as 15 km, 20 km, 25 km, and 35 km, and the central fault depth is fixed at 33 km for the single fault models of the second earthquake for the tsunami sensitivity test. (b) Observed and simulated tsunami waveforms at the Dongkung and Houbihu stations using single fault models with the different central fault depths of the first earthquake.**


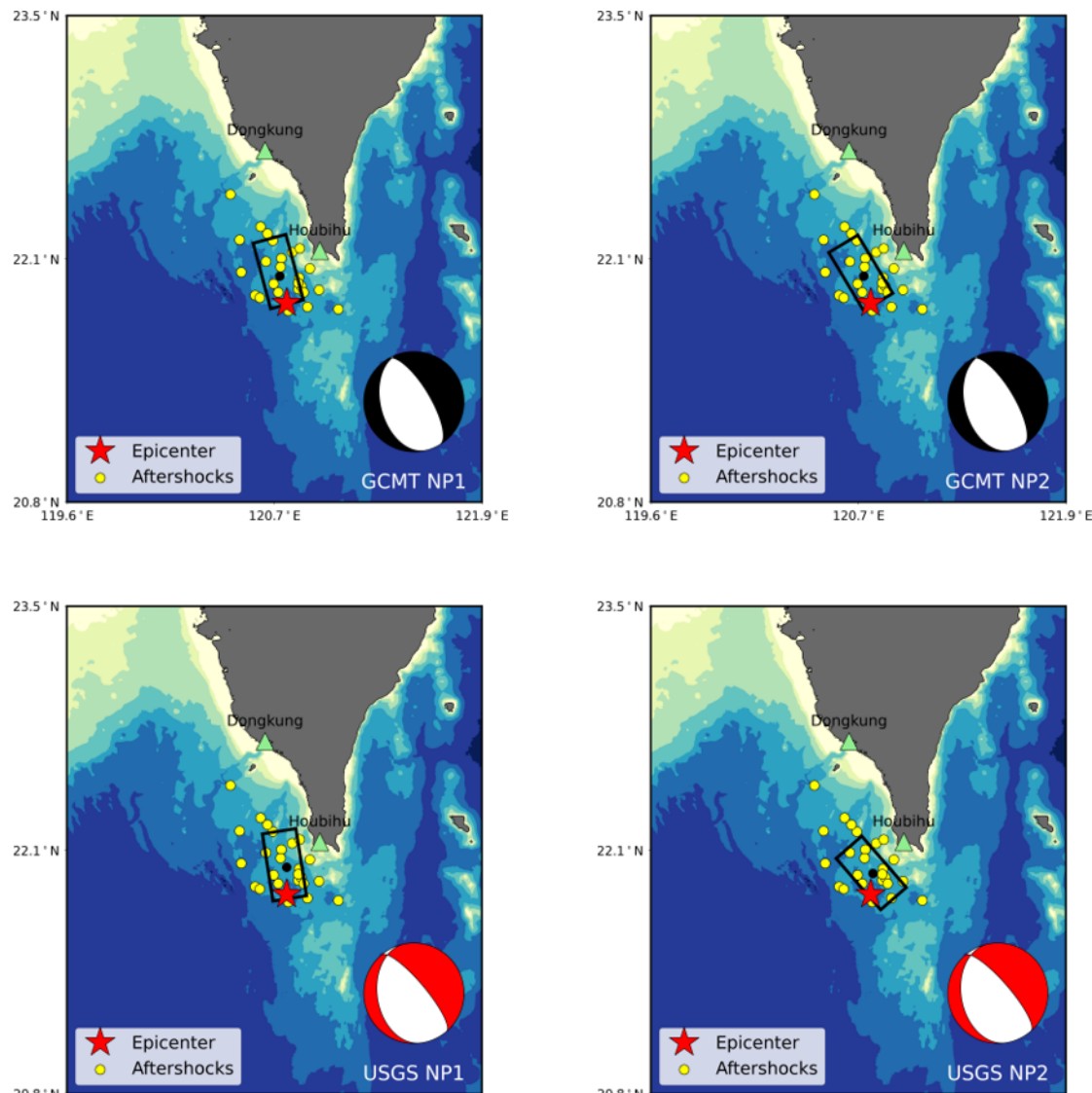


**Figure 15. Simple fault models of the first earthquake (M_w 7.0) using the focal mechanisms from**

**GCMT and USGS. The green triangles indicate the tide gauge stations, red stars indicate the**

**epicenter, yellow circles indicate aftershocks, and the black rectangles indicate the fault model.**


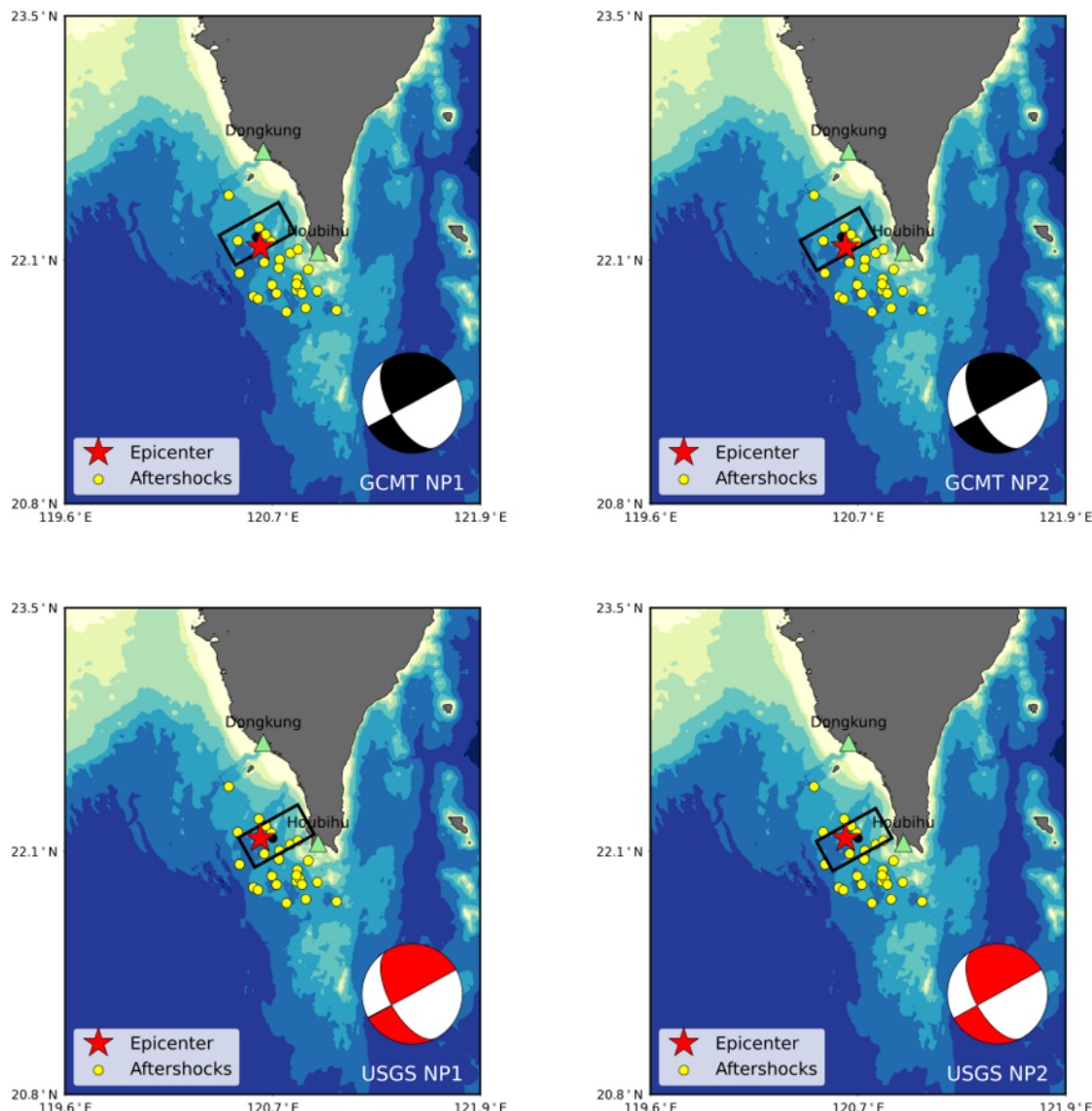


**Figure 16. Simple fault models of the second earthquake (M$_w$ 6.9) using the focal mechanisms**
**from GCMT and USGS. The green triangles indicate the tide gauge stations, red stars indicate**
**the epicenter, yellow circles indicate aftershocks, and the black rectangles indicate the fault**
**model.**


**Table 7. Validation of the simulated tsunami waveforms using single fault models with eight**
**different models of focal mechanisms estimated by GCMT and USGS.**

| Scenario | Moment tensor solution | Nodal plane | | Misfit of simulated tsunami waveforms |
| --- | --- | --- | --- | --- |
| | | Earthquake 1 | Earthquake 2 | |
| S1 | | NP1 | NP1 | 0.591 |
| S2 | | NP1 | NP2 | 0.632 |
| S3 | GCMT | NP2 | NP1 | 0.530 |
| S4 | | NP2 | NP2 | 0.661 |
| S5 | | NP1 | NP1 | 0.529 |
| S6 | | NP1 | NP2 | 0.604 |
| S7 | USGS | NP2 | NP1 | 0.493 |
| S8 | | NP2 | NP2 | 0.735 |



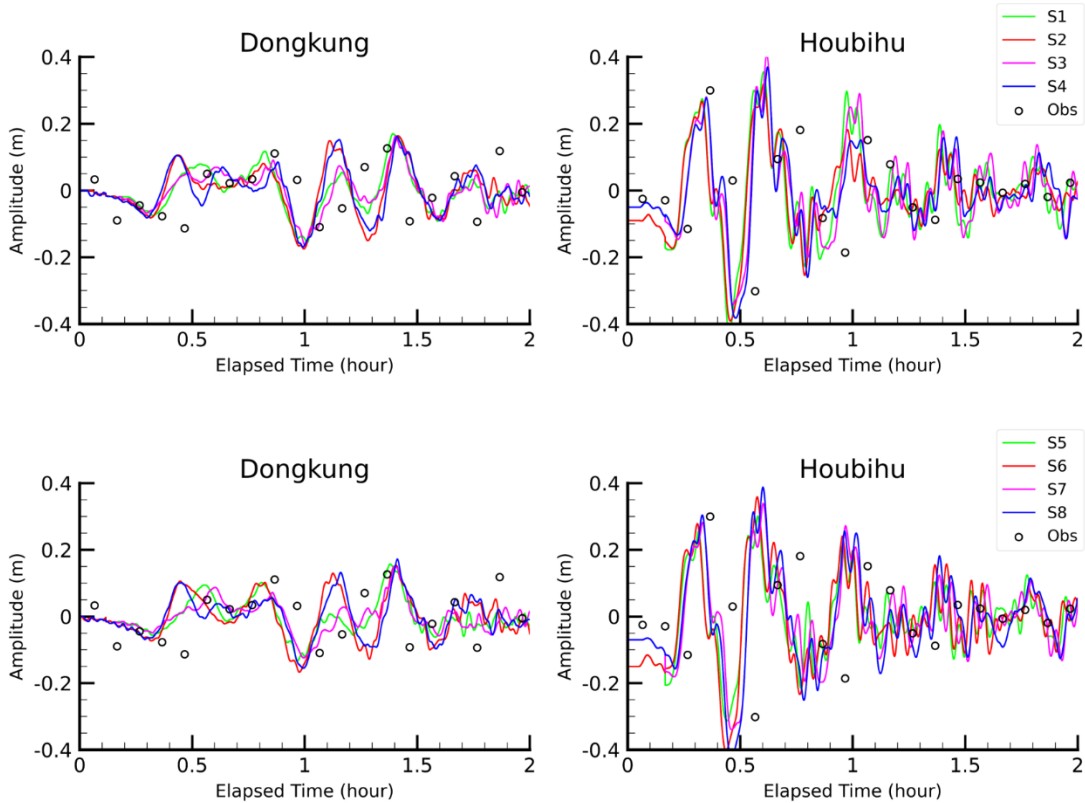

**Figure 17. Comparison of simulated tsunami waveforms at the Dongkung and Houbihu stations**
**using single fault models with eight different models of focal mechanisms estimated by GCMT**
**and USGS.**

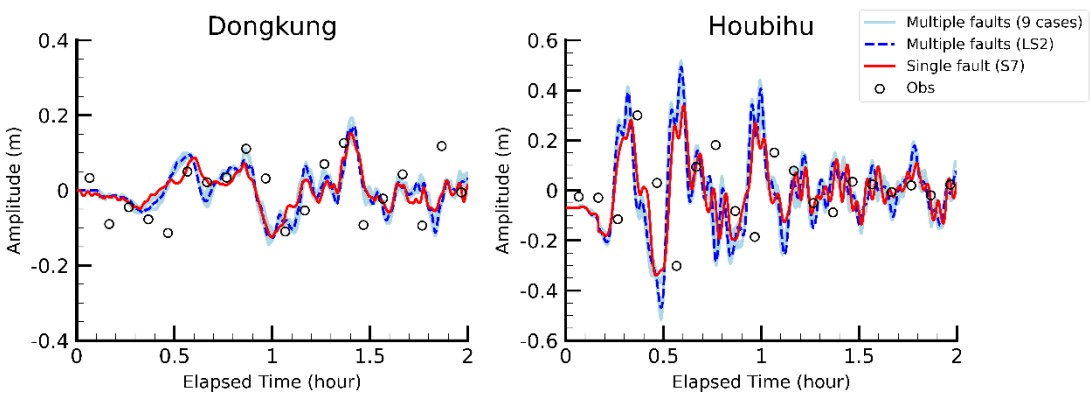


**Figure 18. Comparison of simulated tsunami waveforms at the Dongkung and Houbihu stations**
**using 9 cases of multiple fault models (solid blue lines) and a single fault model of S7 (solid red**
**lines). The simulated tsunami waveforms using the multiple fault model (LS2) are shown as**
**dashed blue lines. The white circles represent the observational data.**

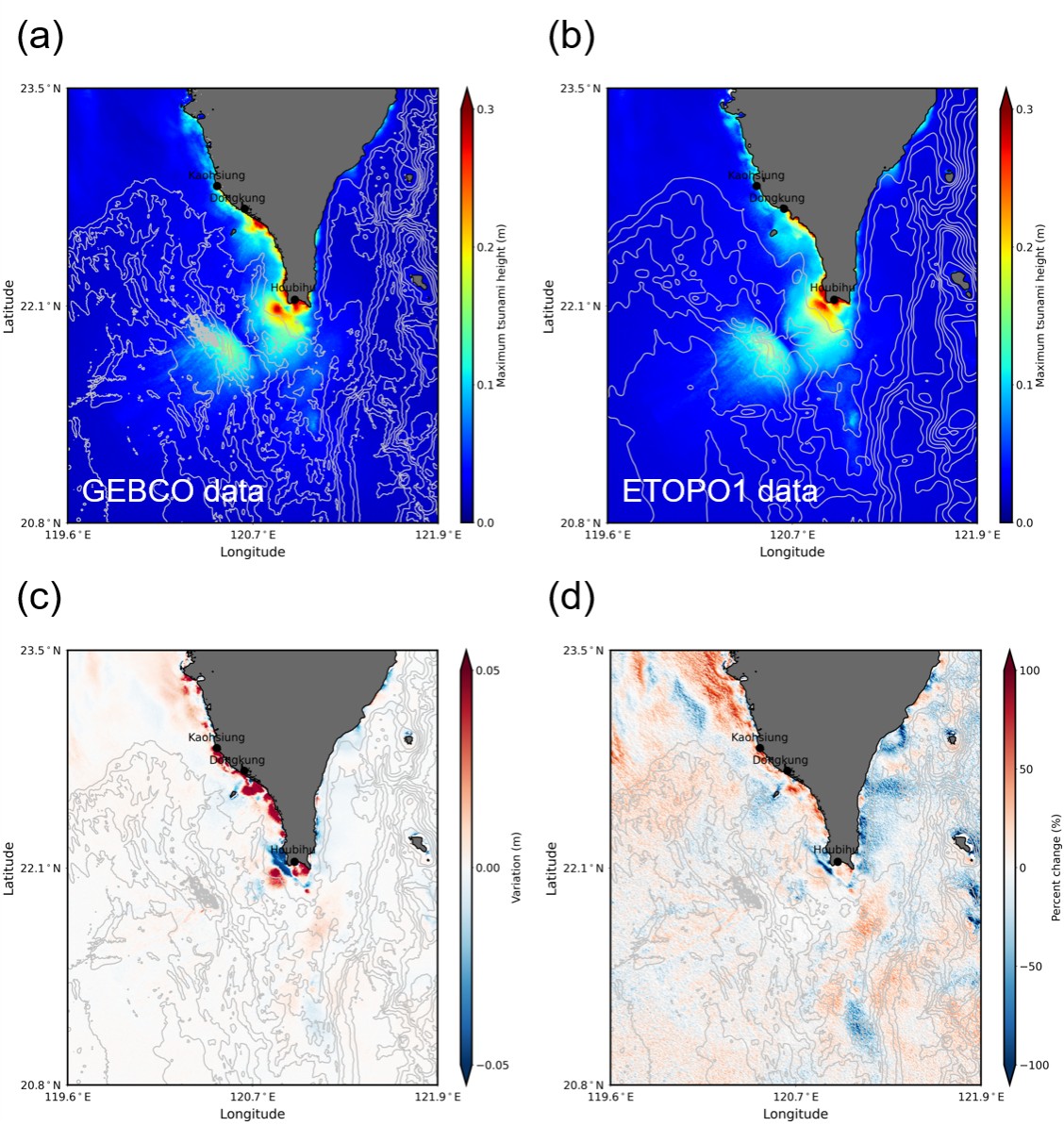

Figure 19. Simulated maximum tsunami height using open-source bathymetry data: (a) GEBCO and (b) ETOPO1 data. (c) The variation and (d) the percent variation in the simulated maximum tsunami height using two sources of bathymetry data. The black circles indicate the locations of the tide gauge stations. The bathymetry contour is 500 m based on the GEBCO or ETOPO1 bathymetric data.

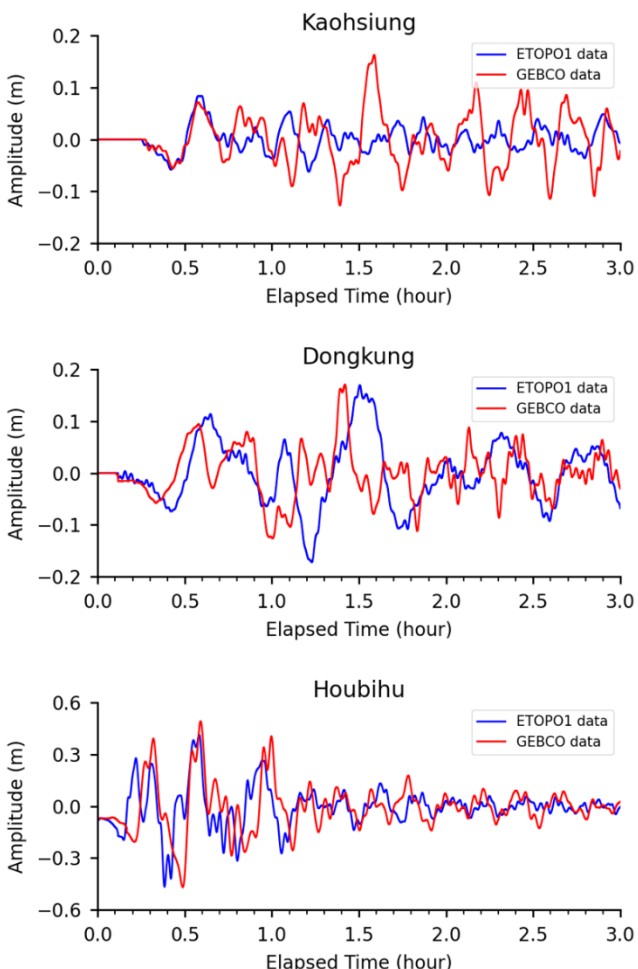


**Figure 20. Simulated tsunami waveforms at the (a) Kaohsiung, (b) Dongkung, and (c) Houbihu**
**stations using two different open-source bathymetry datasets, GEBCO and ETOPO1.**

**Table 8. Details of the locations of the simulated tsunami waveforms and misfit of model**
**results using different open-source bathymetry data at three tide gauge stations.**

| Station | Sea depth (m) | | Simulated wave peak (m) | | $Var_{peak}$ | $\%Var_{peak}$ |
|---|---|---|---|---|---|---|
| | GEBCO | ETOPO1 | GEBCO | ETOPO1 | | |
| Kaohsiung | 10 | 8 | 0.163 | 0.084 | 0.079 | 48.45 |
| Dongkung | 9 | 14 | 0.171 | 0.17 | 0.001 | 0.58 |
| Houbihu | 4 | 11 | 0.493 | 0.414 | 0.079 | 16.02 |



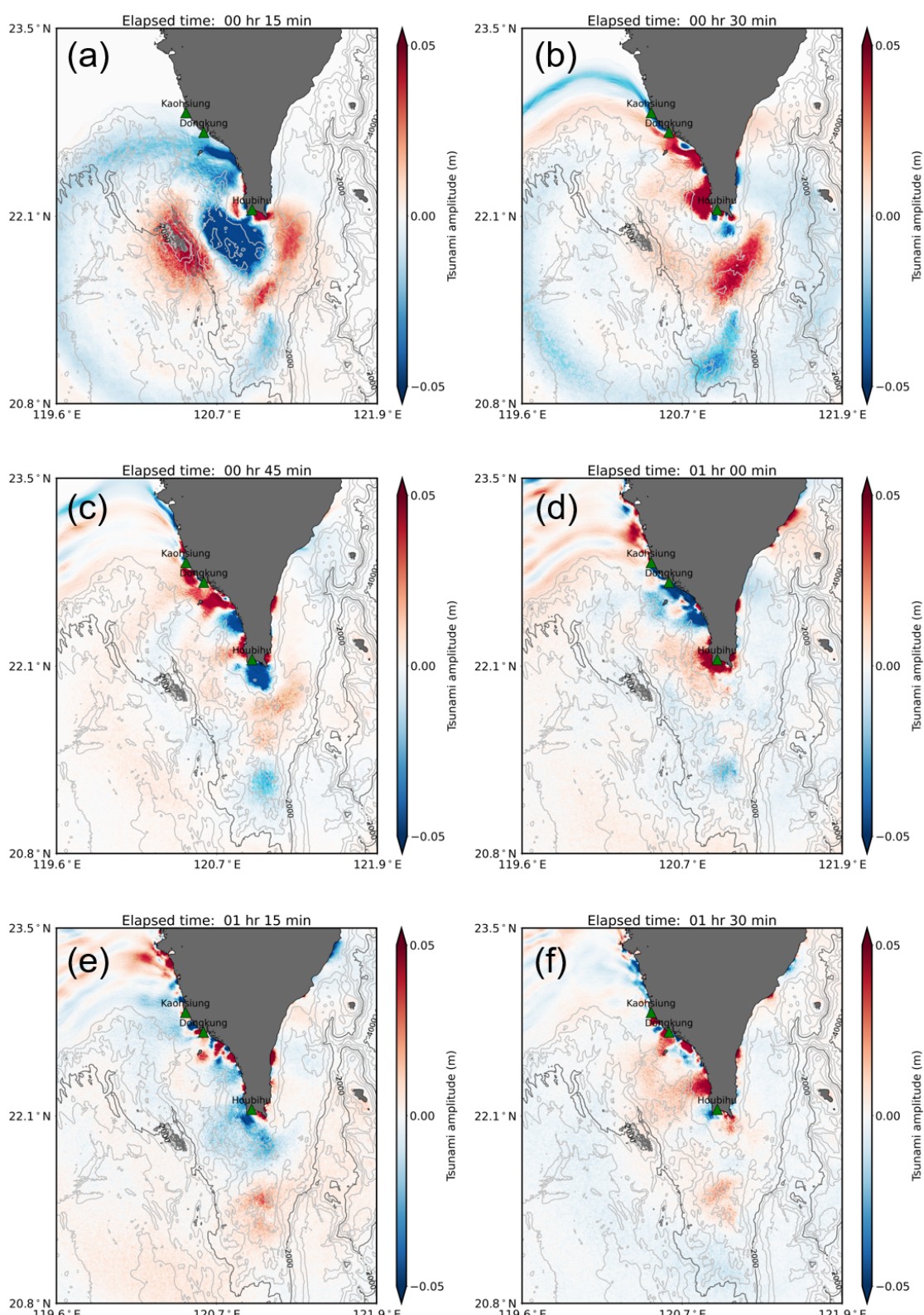

**Figure 21. Tsunami propagation snapshots from the numerical experiment MS. The tide gauge stations are plotted in green triangles. The bathymetry contour is 500 m.**

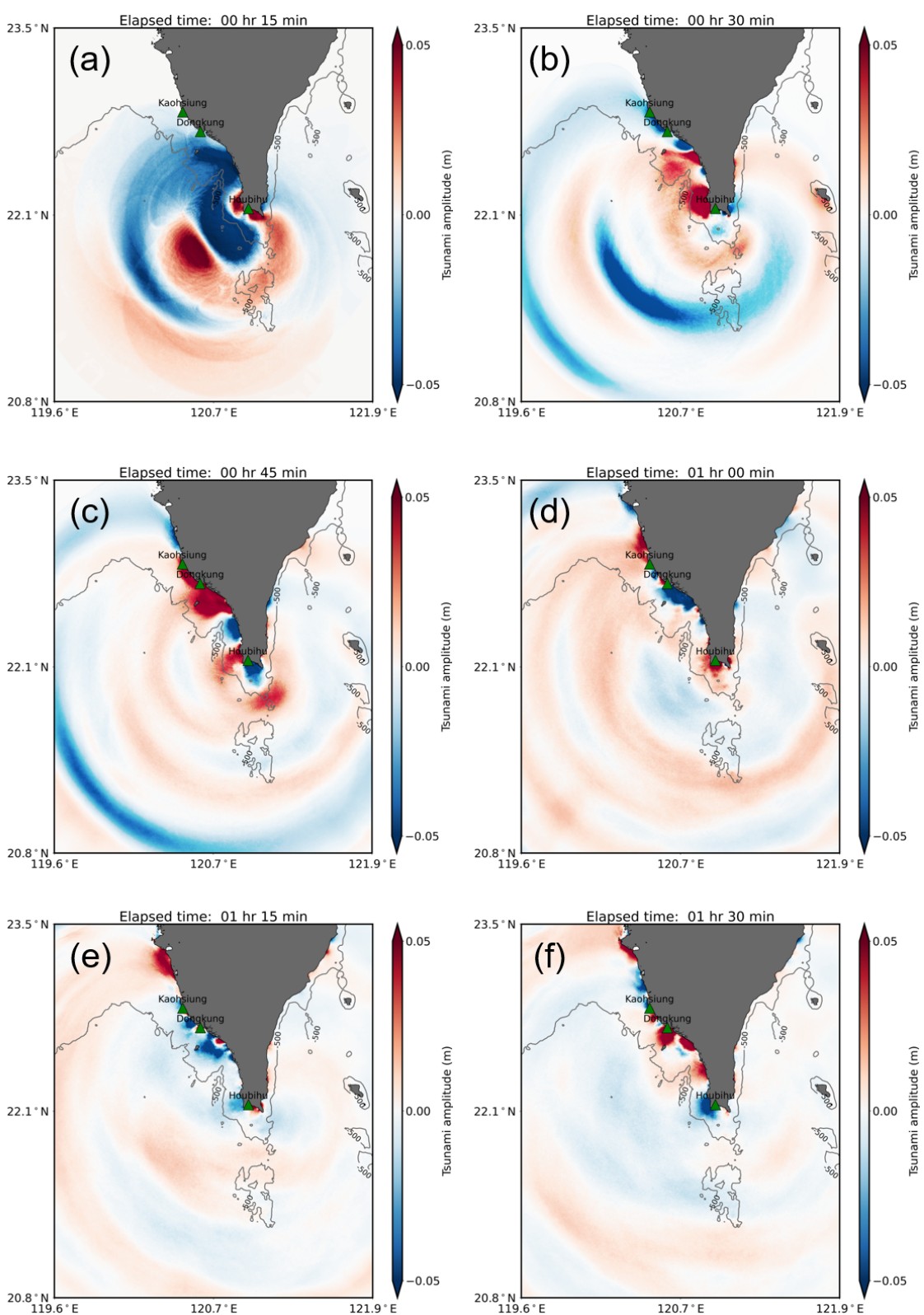

**Figure 22. Tsunami propagation snapshots from the numerical experiment EXP1. The tide gauge**
**stations are plotted as green triangles. The bathymetry contour at a depth of 500 m is shown as**
**a solid gray line.**

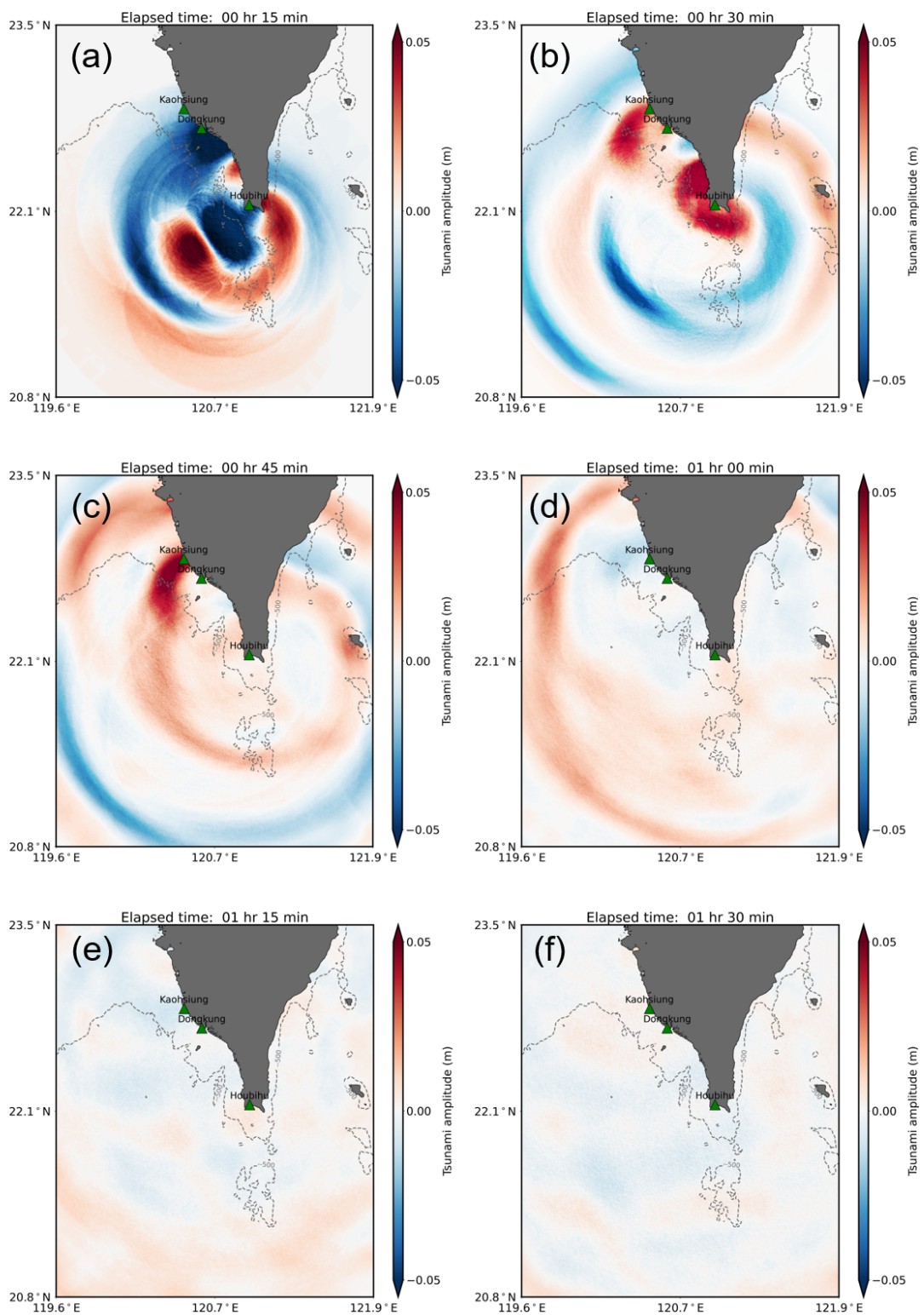

**Figure 23. Tsunami propagation snapshots from the numerical experiment EXP2. The tide gauge**
**stations are plotted as green triangles. The corresponding bathymetry contour of 500 m depth**
**from GEBGO data is shown as a dashed gray line.**

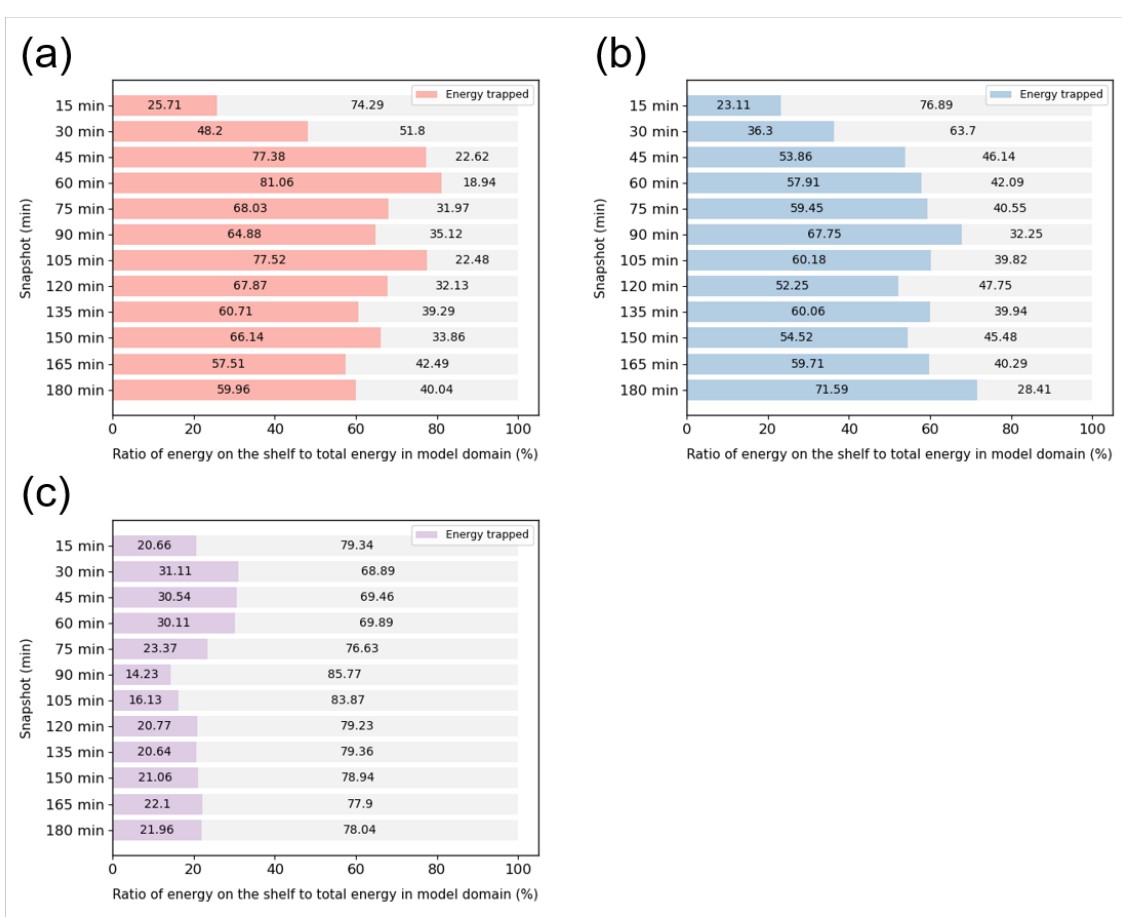

**Figure 24. Trapped ratio of tsunami wave energy calculated from tsunami propagation**
**snapshots every 15 min from numerical experiments (a) MS, (b) EXP1, and (c) EXP2.**

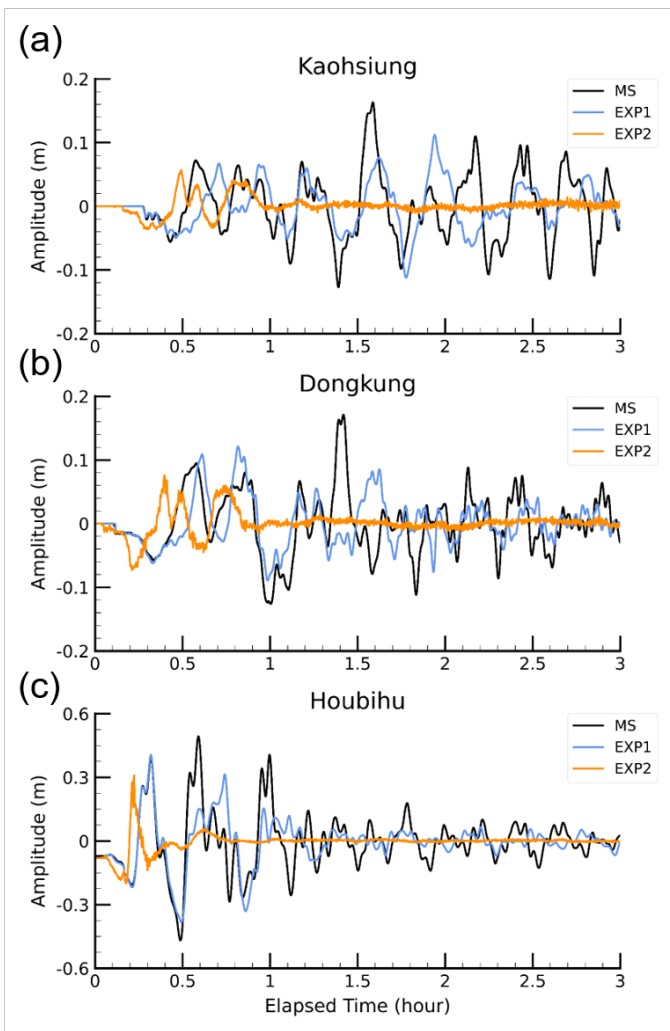


**Figure 25. Simulated tsunami waveforms at the (a) Kaohsiung, (b) Dongkung, and (c) Houbihu**
**stations from numerical experiments MS, EXP1, and EXP2.**

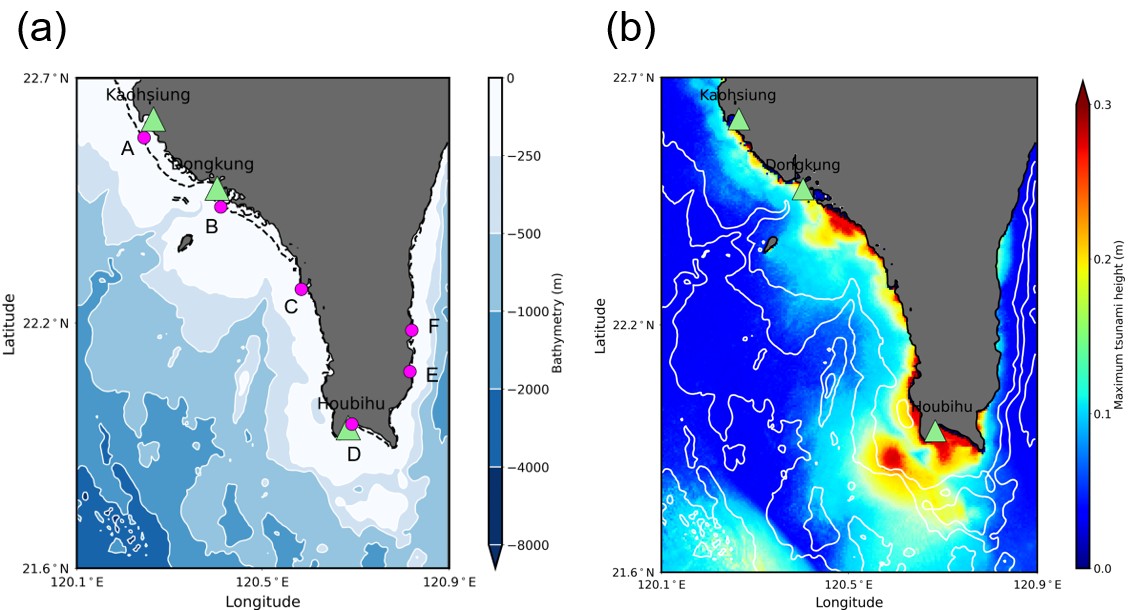

**Figure 26. Zoomed map of the (a) bathymetry around southern Taiwan and (b) simulated**
**maximum tsunami height using a multiple fault model (LS2). Green triangles indicate the**
**locations of tide gauge stations, and pink circles denote numerical wave gauges at a sea depth of**
**20 m. The solid white lines are contour lines, and the dashed black line represents the**
**bathymetric contour at a depth of 20 m.**

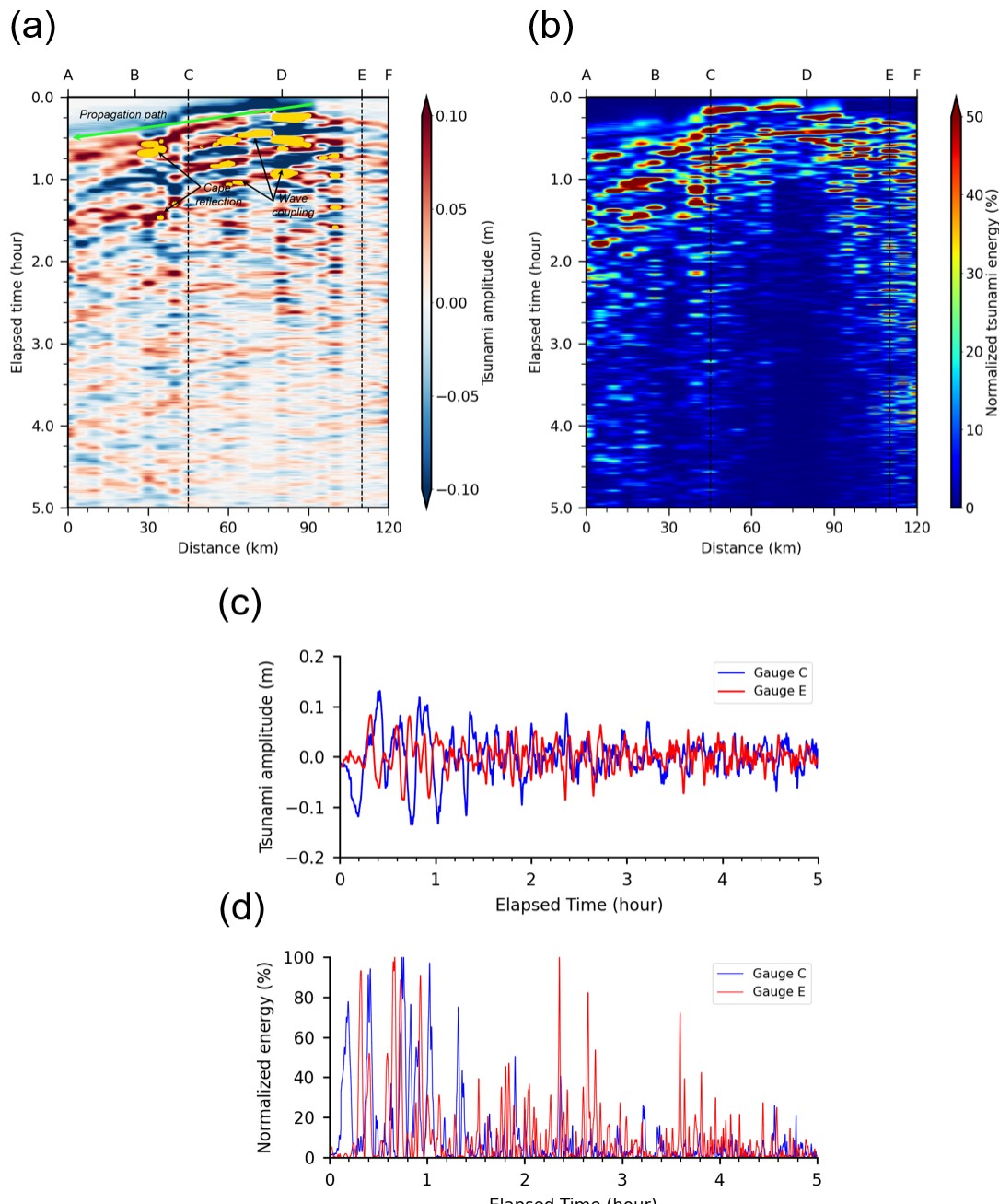


**Figure 27. Time-distance diagram of the (a) tsunami wave and (b) normalized energy along the**
**20 m bathymetry contour from numerical wave gauges A to F and time series measurements of**
**the (c) tsunami amplitude and (d) normalized energy at numerical wave gauges C and E. The**
**dashed black lines indicate the distances of numerical wave gauges C and E from A. For**
**interpretation of the references, please refer to Figure 26a.**

1032

1033