# Peer review of "Characteristics of consecutive tsunamis and resulting"

_Natural Hazards and Earth System Sciences, 2022_

## Author Comment (AC1)

**Dear Editor,**

Thank you for the time and sending us your decision. We have made responses and corrections to reviewers' comments and suggestions as shown below. Corrections made based on comments and suggestions are shown in red.

**Reply to reviewer no. 1**

We highly appreciate your time spent in reviewing the manuscript as well as your valuable comments and suggestions. We are glad that you are interested in our work and your positive feedback. Please find our line-by-line responses and corrections to your comments and suggestions. All responses, corrections and improvements are shown in red in the revised manuscript.

| Reply to general comments |
|---|
| Thank you very much for pointing out these important issues. We totally agreed that the sensitivity and variability aspects of the source models and the bathymetry should be sufficiently discussed, Also, additional investigations should be applied to strengthen the conclusion related to tsunami wave trapping. In order to this, we have applied additional analyses mainly in **section 2.4, 2.5 and 2.6,** and related **sections 4.1**, **4.2** and **4.3.**

In addition, to improve the clarity of the text, we have added more explanations to s**ection 2.3, 6.1, 6.2, 6.3,** as well as additional Tables and Figures to support the explanations. The manuscript was carefully re-written, and the English spellings were to our best to be improved. Please see more details below on our answers and responses. |

| Reviewer comments | Our answers | Corrected manuscript |
|---|---|---|
| Line 15: Please remove 'for the first time' | We thank the reviewer for pointing this out. We corrected it by removing the word. | **Line 14:**
A small tsunami was generated, and recorded at tide gauge stations . |
| Line 44: I suggest putting in a reference to Figure 1 already here. | We thank and agree with the reviewer. We corrected it by linking a reference to Figure 1. | **From line 48 to line 49:**
The locations of Hengchun Peninsula and the epicenters of the successive earthquakes are shown in Figure 1. |
| Line 51: The Lay and Kanamori refence is general but the way the | We thank the reviewer for pointing this out. The sentence | **From line 50 to line 56:**
The respective magnitudes of |

| | | |
|---|---|---|
| sentence reads it sounds like the paper refers to this event. Please rephrase, and include a specific reference work (e.g. from seismology) that consider the 2006 event in particular. | was rephrased, and additional references about earthquakes doublet in seismological perspective of view were included. | these two earthquakes were suggested to be $M_L = 7.0$ ($M_w = 7.0$ in the Global CMT catalog) for the former, and $M_L = 7.0$ ($M_w = 6.9$ in the Global CMT catalog) for the latter. From seismological perspective of view, pairs of large earthquakes with equivalent rupture size and occurred in a similar spatial and temporal proximity were specified as doublet (Lay and Kanamori, 1980; Kagan and Jackson, 1999). Sharing comparable earthquake magnitudes, and very close epicenters and occurrence times, the successive earthquakes are referred as an event of doublet (Ma and Liang, 2008; Wu et al., 2009). |
| Line 51: 'Casualties', do you mean 'fatalities'? The former also refer to injuries, the latter only to loss of life. | Thank you very much for the suggestion. According to the report of National Disaster prevention and Protection Commission, R.O.C., 2007, the 26 December 2006 earthquakes caused 44 injuries, including 2 fatal ones, 3 building collapse, and massive damages of submarine communication cables. To that sense, we considered to use the vocabulary 'Casualties' here. | |
| Line 57: 'propagated toward' à 'propagated towards' | We are very sorry for making this spelling mistake. We | **Line 61:**
A small tsunami was generated |

| | corrected it. | after the successive strong motions of these earthquakes. The tsunami propagated towards, and reached the western coast of southern Taiwan immediately after the earthquakes. |
|---|---|---|
| Line 60: Rephrase sentence, my suggestion 'as it was rare because it was generated by earthquakes in short succession'. | We thank and agreed with the reviewer. We corrected it by rephrasing the sentence. | **Line 63-66:**
The December 2006 tsunami was an important event and attracted public interest, as it was rare because it was generated by earthquakes in short succession, and was a new issue among social communities and ordinary persons in Taiwan about tsunamis. |
| Line 62: 'heightens' à 'increased' | We thank the reviewer for pointing this out. We corrected it. | **Line 67:**
This recent tsunami not only corroborates the tsunami risk in Taiwan, but also increased the awareness of disaster risk management, such as preparedness, and mitigation countermeasures for the next tsunamis. |
| Line 65: Several repeats of the above in this paragraph, I suggest shortening. | We thank the reviewer for pointing this out. We shortened the paragraph. | Please see **line 69**.
The tsunami observations reported following the 26 December 2006 tsunami also opened some questions. |
| Line 67: Please delete sentence starting with 'It has been common understanding…'. This can certainly be disputed and the scientific community is | We thank the reviewer for pointing this out. The sentence starting with 'It has been common understanding…' have been deleted, and the sentences | **Line 70:**
First, the first tsunami wave crest was not shown as the largest in some stations. |

| | | |
|---|---|---|
| definitely aware that later wave arrivals can be larger than the first. | were rephrased. | |
| Line 71: 'prolonged'? Prolonged compared to what? | We apologize for our confusing expression. We meant that some stations recorded the tsunami durations for more than 6 hours during the 2006 earthquake tsunami. We have removed the word 'prolonged', and rephrased the sentence to improve the lack clarity. | **Line 73:**

 Second, tsunami durations for more than 6 h were recorded at some stations following the earthquakes. |
| Lines 80-81: Something is missing in these statements, please rephrase so the meaning is more apparent. | We thank the reviewer for pointing this out. We rephrase the sentences and the meaning. | **Please see line 77-88**

 The other issue was that which source models could better explain the successive tsunamis to the recorded observations in southern Taiwan. Wu et al., 2008 simulated the tsunami from this event using single fault models. They numerically computed the tsunami propagation on a nested grid system with finest grids of 0.125 min resolution bathymetry data and compared their results with observation data from tide gauge stations. Although the source models to this tsunami event have been specified and modeled in previous study, the uncertainty and variability aspects of the source models and bathymetry have not been investigated thoroughly. Such uncertainties in earthquake fault parameters and significant |

| | | difference among the open-source bathymetries can exaggerate the modeled results rather than the predictions from previous study to the 2006 tsunami. Therefore, it is critical to discuss such model's performances from viewpoint of sensibility perspective because it is desirable to obtain a tsunami source model and to understand the reliability of bathymetry data utilized for numerical simulation for reasonably estimating the tsunami wave activities during the 2006 tsunami. |
|---|---|---|
| Line 91: 'justify' à 'hindcast' | We thank the reviewer for pointing this out. We corrected it. | **Line 99:** The December 2006 earthquake tsunami represents a unique and recent incident in Taiwan; therefore, the hindcasting and findings could not only help to further understand the tsunami generation and important behaviors responsible for the tsunami hazards to the island of Taiwan, but also have implications for tsunami warning and disaster risk management. |
| Line 99: Please delete 'In general', and replace the statement 'possible method to study' with 'one source of information we can use to study'. The point is that it can only be | We thank and agreed with the point of view of the reviewer. We corrected it by rephrasing the sentence. | **Line 105:** Time history data of sea levels recorded at coastal sites provide one source of information that we can use to study tsunami patterns. |

| supplementary to other methods, it is usually not enough by itself. | | |
|---|---|---|
| Line 112: 'represent the duration' à 'represent the observation' (duration written twice in sentence) | We thank the reviewer for pointing this out. We corrected it. | **Line 118:**
The tsunami durations represent the observation time of high-energy tsunami waves persisting in a coastal site of observation. |
| Line 113: Remove 'of observation'. 'duration' à 'durations', and 'was' à 'were' | We thank the reviewer for pointing this out. We corrected it. | **Line 119:**
The tsunami durations at all stations were identified based on the calculation of RMS sea levels, indicating the elapsed time of the wave amplitude above the level of normal oscillation before the tsunami wave arrived (Heidarzadeh, 2021). |
| Line 127: 'The' Fourier analysis … | We thank the reviewer for pointing this out. We corrected it. | **Line 132-136:**
The Fourier analysis and the wavelet (time-frequency) analysis. The Fourier analysis is based on the fast Fourier transform (FFT) algorithm, applied based on the updated open-source library Numpy in the Python package (Harris et al., 2020). The Fourier analysis was performed to estimate the spectral components of the time history data of the tsunami waveform. |
| Line 137: 'the' wavelet analysis … | We thank the reviewer for pointing this out. We corrected it. | **Line 133:**
The Fourier analysis and the wavelet (time-frequency) analysis. |
| Line 144: The first sentence in | Thank you very much for the | Please see **section 2.3 (from line** |

| the paragraph is somewhat misleading. I would rather say it is a computer-based method describing the equations of motion for the tsunami wave propagation. You could also add that there are various methods, but that the shallow water model is most used, although dispersive models are more and more used as well. | valuable comments. We rephrased it to improve the clarity of the numerical methods. | **150-157)** |
|---|---|---|
| Line 149: I would say that TUNAMI also cover far-field tsunamis, with limitations of course. | Thank you very much for the valuable comments. We add additional information to this part. | Please see **section 2.3 (from line 154 to 157)** |
| Line 155: You do not describe mesh refinement anywhere. How do you ensure convergence? What is your grid resolution, and what exactly is the CFL number? It should be a minimum to test convergence at least with two different (optimally three) mesh sizes. | We simulated the tsunami propagation using a 450 m bathymetric grid. The mesh size in x and y directions are 538 and 631. The CFL condition is presented as: $$\Delta t \leq \frac{\Delta x}{\sqrt{2gh_{max}}}$$ Where the $\Delta t$ is the time interval, $\Delta x$ is the grid spacings, and $h_{max}$ is the maximum water depth in the model domain. | Please see **section 2.3 (from line 167 to 172)** |
| Line 160: You have stated this before. I suggest to delete this sentence that only repeats what is already written in the intro. | We thank and agreed with the reviewer. We deleted the sentence. | |
| Line 168: Are you simulating with uniform slip? Could you gain anything with adding non-uniform slide and simulate different realisations of the slip | Thank you very much for the valuable suggestions. The tsunami sensitivity to non-uniform fault slip distribution is evaluated. | For the approach, please see section 2.4.2 (**from line 219 to 246**) and for the results of sensitivity analysis, please see section 5.2 (**from line 460 to** |

| | | |
|---|---|---|
| distribution? This deserves to be discussed more. | | **471**) |
| Line 186: 'horizontal effect' à 'horizontal deformation contribution to tsunami generation' | We appreciated the reviewer for the correction. The sentence was revised. | **Please see line 173-175:** The horizontal deformation contribution to tsunami generation in the steep bathymetric slopes (Tanioka and Satake, 1996) is included. |
| Line 191: Why could this not have been caused by landslides? Please elaborate / substantiate, or otherwise skip this statement if you cannot back it up more explicitly. | The statement was skipped. | |
| Line 193: Add 'simulated' before 'initial'. | The vocabulary was revised. | **Please see line 172-173:** As the simulated initial condition inputted for numerical tsunami simulation, the initial water level distribution is calculated from the earthquake fault parameters using the theory of Okada, 1985. |
| Line 203: You may need to elaborate what you mean by 'two bathymetric scenarios'. You probably mean tsunami simulations applying two different bathymetries. You may motivate your work by mentioning how wrong the open source bathy was for 2018 Palu. Similar for 2018 Anak Krakatoa (e.g. Zengaffinen et al., 2021). | For the bathymetric scenarios stated here, we meant the actual and manipulated bathymetries used in numerical simulations to examine the how bathymetry can influence the tsunami wave directivity and wave trapping. In addition, the variability aspects of open source bathymetry to model results was examined. | For the clarity of bathymetric scenarios, please see **section 2.6 (from line 274 to 289).** The details of actual and manipulated bathymetries used in numerical simulations were summarized in **Table 5**. For the examination of tsunami sensitivity to open source bathymetry, the 2018 Palu and the 2018 Anak Krakatoa tsunami were referred as backgrounds and the approach and results could be found in **section 2.5** |

| | | **(from line 249 to 271)** and **section 5.3 (from 474 to 494)**, respectively. |
|---|---|---|
| Line 207: Both are scenarios in a way. I would rephrase, and rather say 'manipulated bathymetry' rather than 'hypothetical scenario'. | We appreciated the reviewer for the comments. The sentences were rephrased. | Please see **section 2.6 (from line 274-286)**. |
| Line 211: You only investigate two different bathymetries, and this might be a bit thin to conclude in general. I suggest that the uncertainty related to the bathymetry is discussed more. | Thank you very much for the valuable suggestions. We agreed with the reviewer.
In addition to the two different bathymetries (i.e., actual and manipulated bathymetry by replacing sea depths larger than 500 m to 500 m), a rather hypothetical situation was examined using the manipulated bathymetry of flatted sea bottom of 500 m depth. | Please see **section 2.6 (from line 274-286)** and **section 6.1 (from line 498-528).** |
| Line 231: Please rephrase 'different mechanism of tsunami waves was' à 'different propagation effects were' | We appreciated the reviewer for pointing this out. The sentence was revised. | **Please see line 306:**
These results suggest that the different propagation effects were active at these coastal sites during the passage of the 2006 tsunami. |
| Line 237: The aspects of the wave recordings should be move more up front, at least within this subsection, it is important background. | We appreciated the reviewer for the valuable comments. The aspects of the wave recordings were moved and considered as important background for simulating scenarios with non-uniform fault slip distributions. | **Please see line 451-457.**
While the single fault models can produce the simulated tsunami waveforms well consistent to the observations, the badly sampled (i.e., 6 min interval) signals recorded in coastal stations also raise some questions, as one would expect some potential high tsunami |

| | | waves behind the observed signals. To that sense, overestimation of modeled results was expected, but the simulated tsunami waveforms using single fault models present the opposite. This indicates that the single fault models (i.e., with uniform fault slip) may not be sufficient and the asperity area (i.e., with large fault slip) on the fault should be evaluated. The tsunami sensitivity to asperity locations of multiple fault models will be discussed in next section. |
|---|---|---|
| Line 254: You say 'abnormally long', but compared to what? | We apologize for our confusing expression. We meant that Kaohsiung and Houbihu station recorded the tsunami durations for more than 6 hours during the 2006 tsunami. We have removed the word 'prolonged', and rephrased the sentence to improve the lack clarity. | **Please see line 326-327**

 The calculated tsunami duration at Dongkung was as much as 3.9 h, while the tsunami continued for more than 6 h in Kaohsiung and Houbihu. |
| Line 271: What does the background spectra contain? Are they de-tided? Please clarify. | We apologize for our lack expression. The background spectra are the spectral components calculated from de-tided observed data of 5 h before the tsunami arrival. | **Please see line 346**

 The background spectra are the spectral components calculated from de-tided observed data of 5 h before the tsunami arrival, and the spectral components of the observed tsunami waveform were computed using 5 h data recorded at tide gauge after tsunami wave arrived. |
| Line 293: I think this is stating | Thank you very much for | |

| | | |
|---|---|---|
| the obvious, and it could perhaps be skipped? | pointing this out. We skipped this statement. | |
| Line 329: 'determined' à 'estimated' | Thank you very much for pointing this out. The vocabulary was revised. | **Please see line 389** Assuming the mean sea depths around tsunami source region is 300 m, the fault rupture dimensions for the two earthquakes could be estimated to 20- 40 km. |
| Line 372: I would say it is the opposite: The data can be used to validate the numerical simulations. | Thank you very much for the valuable comments. The sentence was rephrased. | **Please see line 180-183** The multiple forward tsunami simulations were conducted using the single fault models with different fault depths and fault orientations. The main attempt of the multiple forward tsunami simulations is to find a single fault model that can produce the tsunami waveforms in the best agreement of fit to the observations made by tide gauge stations in southern Taiwan. |
| Line 377: If there is undersampling, you would normally expect the numerical simulations to overestimate the wave measurements, because the measurements would miss out on larger amplitude waves. Here it seems to be the other way around, implying that the simulations are lower than you would expect from the measurements. The authors need to elaborate on this. For instance, why was not alternative | We appreciate the reviewer for the valuable suggestions on this issue. We established and simulated the non-uniform slip scenarios to examine whether the measurements have missed out on larger amplitude waves. | **Please see section 2.4.2 (from line 220 to 247) for the approach and section 5.2 (from line 461 to 472) for the results.** |

| | | |
|---|---|---|
| scenarios or random / heterogeneous slip investigated with several scenarios? | | |
| Line 388: Replace 'It is commonly understood that' with 'The longest wave component'. Then add an 'a' ahead of 'velocity'. | Thank you very much for the valuable comments. The vocabulary was revised, and sentence was rephrased. | **Please see line 499**
The longest wave component of tsunami travel with a velocity that is mainly governed by seafloor depths. |
| Line 390: Add 'through diffraction' after 'wave direction'. | We appreciate the reviewer for the correction. The vocabulary was added. | **Please see line 502**
The significant change in propagation speed allows the tsunami to change its wave direction through diffraction. |
| Line 391: 'of the' à 'using' | Thank you very much for the suggestion. The vocabulary was revised. | **Please see line 504**
Simulated snapshots of tsunami wave propagation using actual (MS) bathymetry are shown in Figure 21. |
| Line 395: I found it difficult to follow the authors in this paragraph. I suggest that the authors review the text and try to rephrase it, at least the first 6-7 lines. | We apologize for the confusing expression in this paragraph. The paragraph was re-written. | Please see **section 6.1 (from line 499 to 529).** |
| Line 422: I suggest to comment on previous studies investigating fits and misfits using open source bathymetry and topography data, e.g. Griffin et al., (2015). | Thank you very much for this valuable suggestion. We examined the misfits of modeled results using open-accessible bathymetry and topography. | Please see **section 5.3 (from line 474 to 494)** |
| Line 426: The sentence starting with 'These results further confirmed …' I found was formulated too conclusive. The number of investigations are rather limited, and there should | We appreciate the reviewer for the valuable comments. To strength the conclusion related to wave trapping, we applied additional analysis including energy trapping ratio, and the | Please see **section 6.2** and **6.3 (from line 531 to 565)** |

| | | |
|---|---|---|
| be room for additional investigations to strengthen the conclusion related to wave trapping. | comparison of calculated waveforms. | |
| Line 439-441: What the authors write here is not clear from the figures. If there is additional not shown that back this up please state this explicitly. | We apologize for the unclarity of the figure. We replotted the figure and rephased the statement in this paragraph. | Pease see **section 6.4 (from line 568 to 600)** and **Figure 27.** |
| Line 482: 'characterized' à 'analyzed' | Thank you very much for the suggestion. The vocabulary was revised. | The physical characteristics of tsunami waveforms in all three tide gauge stations in southern Taiwan during the December 2006 tsunami were analyzed. |

**2.3 Numerical tsunami simulation**

Numerical simulation is a computer-based method, which describes the equations of the motion for the tsunami wave propagation. The tsunami wave propagation can be numerically modeled based on various theories, including shallow water and dispersive wave theories. Among those theories, the shallow water equations are one of the most commonly used methods to model the tsunami propagation from the source to nearshore. There are various computational models developed to solve the shallow water equations and the TUNAMI (Tohoku University Numerical Analysis Model for Investigation of tsunamis) code is one of the models that widely used to simulate both far-field and near-field tsunamis (Suppasri et al., 2010; Suppasri et al., 2014). The No.2 version of the TUNAMI code (TUNAMI-N2) is mainly developed to deal with near-field tsunamis by applying nonlinear theory of shallow water equations, which is solved using a leap-frog scheme (Imamura, 1995). In this study, the TUNAMI-N2 model was used to simulate the 2006 tsunami with nonlinear shallow water equations. The nonlinear shallow water equations on the Cartesian coordinate system are presented in equation (2)-(4) and the nonlinear equations are solved by applying the finite difference method.

$$\frac{\partial \eta}{\partial t} + \frac{\partial M}{\partial x} + \frac{\partial N}{\partial y} = 0 \tag{2}$$

$$\frac{\partial M}{\partial t} + \frac{\partial}{\partial \chi}\left(\frac{M^2}{D}\right) + \frac{\partial}{\partial y}\left(\frac{MN}{D}\right) + gD\frac{\partial \eta}{\partial \chi} + \frac{gn^2}{D^{\frac{7}{3}}}M\sqrt{M^2 + N^2} = 0 \tag{3}$$

$$\frac{\partial N}{\partial t} + \frac{\partial}{\partial \chi}\left(\frac{MN}{D}\right) + \frac{\partial}{\partial y}\left(\frac{N^2}{D}\right) + gD\frac{\partial \eta}{\partial y} + \frac{gn^2}{D^{\frac{7}{3}}}N\sqrt{M^2 + N^2} = 0 \tag{4}$$

In the governing equations, $\eta$ is the water level, $M$ and $N$ are the discharge fluxes in the $x$ and $y$ directions, $D$ is the total water depth, $g$ is the gravitational acceleration, and $n$ is Manning's roughness coefficient. The bottom friction term was represented by the Manning roughness coefficient, which was set equal to 0.025 s m$^{-1/3}$, assuming the seafloor in the model domain with a perfect condition. The numerical tsunami simulations were conducted with a time interval of 0.1 s and grid spacings of 450 m. The entire model domain covered the source region and the southern Taiwan, which comprised mesh numbers of 538 and 631 in x and y directions. The time interval and grid spacings were set up to satisfy the Courant-Friedrichs-Lewy (CFL) condition to ensure the stability of simulation. The CFL condition is presented in equation (5).

$$\Delta t \leq \frac{\Delta x}{\sqrt{2gh_{max}}} \tag{5}$$

Where the $\Delta t$ is the time interval, $\Delta x$ is the grid spacings, and h$_{max}$ is the maximum water depth in the model domain. As the initial condition inputted for numerical tsunami simulations, the simulated initial water level distributions were calculated from the earthquake fault parameters using the theory of Okada, 1985. In addition, the horizontal deformation contribution to tsunami generation in the steep bathymetric slopes (Tanioka and Satake, 1996) was included. The calculation conditions for the numerical tsunami simulation

were summarized in Table 1.

**2.4 Sensitivity analysis of source models**

**2.4.1 Single fault models**

The multiple forward tsunami simulations were conducted using the single fault models with different fault depths and fault orientations. The main attempt of the multiple forward tsunami simulations is to find a single fault model that can produce the tsunami waveforms in the best agreement of fit to the observations made by tide gauge stations in southern Taiwan.

There are two moment tensor solutions available from the Global Centroid Moment Tensor project (GCMT) and United States Geological Survey (USGS) to the successive earthquakes on 26 December 2006 (Figure 2.). Each solution suggests two possible fault planes to respective earthquakes. The focal mechanisms for the two earthquakes estimated by GCMT and USGS are summarized in Table 2.

Through the analysis of simulated tsunami waveforms made by multiple forward tsunami simulations, one of those fault planes can be chosen as the appropriate fault plane to the respective earthquakes of the 2006 earthquake doublet. This approach has been applied in previous study to obtain the optimum fault plane for the 2016 Fukushima normal-faulting earthquake (Gusman et al., 2017).

Wu et al., 2008 computed the synthetic tsunami waveforms based on single fault models using different fault planes of GCMT's moment tensor solutions. They found that the nodal plane (NP) of NP2 to the first earthquake, with a strike of 329°, dip of 61°, rake of -98°, and the fault plane of NP1 to the second earthquake, with a strike of 151°, dip of 48°, and rake of 0° produced the tsunami waveforms better fit to observed data.

Based on the study conducted by Wu et al., 2008, the focal mechanisms of NP2 to the first and NP1 to the second earthquake from GCMT's solution were used for sensitivity analysis to fault depths. The approximated fault area of length 40 km and width 20 km (800 km$^2$) were estimated to the successive earthquakes based on the empirical formula with tsunami source periods. The methods by which fault area of the two earthquakes have been obtained will be discussed in section 4.1. For the given moment magnitude ($M_w$) of 7.0 and 6.9 earthquake, the amount of average slip can be estimated to 1.66 m to the first (i.e., $M_w$ 7.0) and 1.17 m to the second earthquake ($M_w$ 6.9) assuming the rigidity of 30 GPa. The centroid depth of the GCMT (20 km) and the USGS (25 km) to the first earthquake were significantly different, while similar depth of 33 km was proposed to the second earthquake. Therefore, for the sensitivity analysis of fault central depth, the central fault depth of 15, 20, 25, and 35 km of the first earthquake were evaluated.

After determining the best fault central depth for the single fault models of the two earthquakes, the multiple tsunami forward simulations were applied to all possible fault planes from the moment tensor solutions estimated by GCMT and USGS using single fault. The misfit of observed and simulated tsunami waveforms from the multiple tsunami forward simulations were calculated and compared to examine the focal mechanisms that better explain the observed tsunami data. The misfit of observed and simulated

tsunami waveforms can be calculated using equation (6).

$$\varepsilon = \frac{1}{N} \sqrt{\sum_{i=1}^{N} \frac{(Obs_i - Sim_i)^2}{(Obs_i)^2}} \tag{6}$$

Where $\varepsilon$ is the misfit of observed and synthetic tsunami waveform, $N$ is the total number of data points, $Obs_i$ is the observed data at time step $i$, $Sim_i$ is the simulated data at time step $i$. The equation (8) calculated $\varepsilon$ for one station. For cases with several stations, the overall misfit is gained from the mean of $\varepsilon$ computed from all stations.

**2.4.2 Multiple fault models**

After determining the best central fault depths and fault orientations of single fault, the area of each single fault was subdivided into 8 sub-faults with areas of 10 km $\times$ 10 km, with 4 and 2 sub-faults along the strike and dip axis, respectively. The location of each sub-faults in the fault model of the two earthquakes are shown in Figure 4. The top depths for the two earthquakes are 15.3 km and 29.1 km, respectively, which corresponds to sub-fault 1-4 in each fault models (Figure 4a, b). The rest of the depths from shallowest to the deepest portion along dip axis are derived using fault parameters of width dimensions and dip angles. The respective fault parameters of each sub-fault in fault models of the two earthquakes were summarized in Table 3.

The tsunami sensitivity to non-uniform slip distribution of the fault model was evaluated. To that purpose, two levels of slip to each sub-fault were established, which were the large (asperity) slip and background slip regions of the entire fault. The large slip and background slip region should satisfy the moment magnitude ($M_w$) to avoid overestimation. The slip amount in each region was obtained from following procedures. First, the amount of average slip ($D_a$) was calculated using moment magnitude ($M_w$), entire fault area (S), and rigidity ($\mu$) of 30 GPa following the equations introduced by Kanamori, 1977. The equations are presented in equation (7)-(8).

$$M_w = \frac{\log M_0 - 9.1}{1.5} \tag{7}$$

$$D_a = \frac{M_0}{\mu S} \tag{8}$$

Next, the slip amount of large slip ($2D_a$) was assumed as twice the average slip based on a report of Tsunami receipt, 2017. The total area of large slip area (S') was set to be 25% of the entire fault area, and the seismic moment of large slip area ($M_0$') can be obtained using equation (8). Then, the slip amount of background area ($D_b$) was estimated using the area of background region ($S_b$) and following equation (8)-(9).

$$S_b = S - S' \tag{8}$$

$$D_b = \frac{M_0 - M_0'}{\mu S_b} \tag{9}$$

The details of slip amount in each region for the two earthquakes were summarized in Table 4a.

After determining the slip amount of asperity and background regions, the tsunami sensitivity to asperity location was studied. The asperity area with large slip is placed in the shallow portion of entire fault area based on the lesson of the 2011 Tohoku-Oki earthquake (Satake et al., 2013; Fukutani et al., 2021), focusing on north (sub-fault 3-4), central (sub-fault 2-3), and south (sub-fault 1-2) part of each earthquake fault model. Assuming different location of asperity to the two earthquakes, the total number of 9 scenarios were simulated. The multiple fault models and the generated tsunamis of each earthquake were shown in Figure 5. and 6. The asperity location of multiple fault models for the two earthquakes in each scenario were summarized in Table 4b.

**2.5 Tsunami simulation using open-source bathymetry data**

Aside from fault parameters of source models, the bathymetry data is needed for simulating tsunami wave propagation. The simulated results of tsunami propagation are known sensitive to accuracy and resolution of bathymetry data. Despite it can be expected that the bathymetry data with a higher accuracy and resolution can produce the simulated results better fit to actual values, such a data is not always available and freely accessible. Due to this limitation, the open-source datasets are often utilized for modeling tsunamis in many previous studies (Koshimura et al., 2008; Suppasri et al., 2012; Li et al., 2016; Otake et al., 2020).

Unfortunately, the open-source datasets are sometimes problematic and not sufficient to accurately simulate the tsunami waves because of the lack of data accuracy and quality (Griffin et al., 2015). Similar issue has been reported by Zengaffinen et al., 2021 and Heidarzadeh et al., 2019 in simulating the 2018 Anak Krakatoa tsunami and the 2018 Sulawesi tsunami. Significant differences from various sources of datasets can also result in the modeled results opposite to estimated values from previous studies for purpose of tsunami hazard assessment. Therefore, it is important to assess and to comment on different available open-source bathymetries in subject to their model performances taking the 2006 tsunami as a case study.

For such purpose, the tsunami simulation was applied to two different sources of bathymetry data GEBCO data and ETOPO1 data, separately, and the misfit between the modeled results was evaluated. The GEBCO data contains the bathymetry data with a grid spacings of 15 arc seconds while ETOPO1 data has sea depth data with a resolution of 1 arc minute. To fairly investigate the model performances from different datasets, bathymetry data from two datasets were derived to 450 m grids and used as input for numerical tsunami simulations. Figure 7. shows the bathymetry data of modeled domain obtained from GEBCO and ETOPO1 data. As the initial condition, the initial water distribution of tsunami generated by the proposed multiple fault model (LS2) was used for these simulations, which the asperity location of the two earthquakes were assumed at the central of entire fault area.

**2.6 Evaluation of bathymetry effect to tsunami wave trapping**

To examine any significant change in tsunami wave transmission that could be recognized due to the bathymetry effect during the passage of the 2006 tsunami, the numerical experiments (MS, EXP1, EXP2) of tsunami propagation were conducted using actual and manipulated bathymetry. For the numerical experiment of the main simulation (MS), the actual bathymetry data with a resolution of 450 m derived from sea depth data with a grid spacings of 15 arc seconds from General Bathymetric Chart of Oceans (GEBCO) was used. For the manipulated bathymetry data used for numerical experiment EXP1, the sea depths larger than 500 m was replaced with 500 m depths. For the numerical experiment EXP2, the bathymetry data was manipulated by removing the sea depth with flattened sea bottom at a depth of 500 m. It is noted that the 500 m depth was defined as indicative because the bathymetric slopes for sea depths shallower than 500 m in front of southern Taiwan is great gentle, and therefore considered as the shelf region. Figure 8. shows the map manipulated bathymetry of model domain for numerical experiments EXP1 and EXP2. The details of bathymetry data used for numerical experiment MS, EXP1, and EXP2 was summarized in Table 5.

The results of numerical experiments were compared to examine how tsunami wave directivity could be changed due to the bathymetric effect, and to evaluate how much performance could the tsunami wave energy be coastally trapped in different bathymetric conditions during the passage of 2006 tsunami.

**4 Sensitivity analysis to source models and bathymetry data**

**4.1 Single fault models**

**4.1.1 Tsunami sensitivity to fault depths**

The sensitivity of simulated tsunami waveforms to fault depth was evaluated by varying the central fault depths of the first earthquake. The fault dimensions of 40 km $\times$ 20 km was applied to the two earthquakes. The single fault model of two earthquakes were constructed using GCMT's moment tensor solution of nodal plane NP2 to the first and NP1 to the second earthquake. The tide gauge stations of Dongkung and Houbihu were chosen for this sensitivity analysis. Dongkung and Houbihu stations were chosen because those are the near-field stations from the source region, and therefore more sensitive to the tsunami source. The single fault models of two earthquakes and the locations of near-field tide gauge stations used for sensitivity analysis of fault depths were shown in Figure 14a.

Figure 14b showed the observed and simulated tsunami waveforms at Dongkung and Houbihu station using different fault depths of the first earthquake. At Dongkung station, the first circle of simulated tsunami waveforms matches the observed data well regardless the fault depths. At Houbihu station, the first wave crest of simulated waveform from the fault depth of 35 km is half the observed value. The simulated tsunami waveforms with shallower depths of 15 km and 20 km produce significantly higher amplitude during the

arrival of first crest wave. These revealed that the coastal site with a rather shorter distance to the source is more sensitive to the earthquake fault depths. The simulated waveforms from central fault depth of 20 km better fit to observed data than others and therefore considered as the best fault depths for simulation.

**4.1.2     Comparison of eight models**

The single fault models with fault dimensions of 40 km × 20 km, and central depth of 20 km for the first and 33 km for the second earthquake were used in tsunami simulations using eight different sets of focal mechanisms to the two earthquakes estimated from GCMT and USGS. The single fault models of the two earthquakes with different focal mechanisms were plotted in Figure 15. and 16. The details of eight different sets of earthquake focal mechanisms was listed in Table 7.

In general, the simulated tsunami waveforms from all sets of earthquake focal mechanisms match the observed data well. Figure 17. shows the observed and simulated tsunami waveforms at Dongkung and Houbihu station using eight different sets of earthquake focal mechanisms. The simulated tsunami waveform from earthquake focal mechanisms of S3 (misfit = 0.530), S5 (misfit = 0.529), and S7 (misfit = 0.493) show better agreement of fit to observations than others (Table 7). Among them, the earthquake focal mechanisms of S7 were found to be the best fit scenario with the smallest misfit from the observations. The scenario S7 contains fault orientations of nodal plane (NP) NP2 of the first and NP1 of the second earthquake from USGS's moment tensor solution (Figure 15d, 16c).

While the single fault models can produce the simulated tsunami waveforms well consistent to the observations, the badly sampled (i.e., 6 min interval) signals recorded in coastal stations also raise some questions, as one would expect some potential high tsunami waves behind the observed signals. To that sense, overestimation of modeled results was expected, but the simulated tsunami waveforms using single fault models present the opposite. This indicates that the single fault models (i.e., with uniform fault slip) may not be sufficient and the asperity area (i.e., with large fault slip) on the fault should be evaluated. The tsunami sensitivity to asperity locations of multiple fault models will be discussed in next section.

**4.2  Tsunami sensitivity to uniform and non-uniform fault slip models**

The sensitivity of simulated tsunami waveforms to non-uniform fault slip distribution was evaluated. The single fault models were also modeled to identify the significant differences on modeled results using single (i.e., uniform slip) and multiple (i.e., non-uniform slip) fault models.

Figure 18. shows the observed and simulated tsunami waveforms at Dongkung and Houbihu station using non-uniform slip models (total 9 cases) and uniform slip model. At Dongkung station, the simulated tsunami waveforms from multiple fault models are not much different from the single fault models. Both models can produce the tsunami waveforms in a good agreement to the observed values recorded at this station. At Houbihu station, the non-uniform slip models produce the first wave crest significantly higher than observations. The simulated wave peaks from non-uniform slip models produce the wave heights

approximately twice of those simulated using uniform slip. These results support that near-field station of Houbihu is rather sensitive to the effect of fault slip distribution, and some high tsunami waves might have been missed out in recorded signals from Houbihu station during the 2006 tsunami.

**4.3 Tsunami simulation using open-source data of bathymetry**

To analyze the tsunami sensitivity on different sources of open-accessible bathymetry data, the numerical simulations were applied using GEBCO data and ETOPO1 data. The differences between the modeled results using different bathymetry data are evaluated. This is a comparison of modeled wave peaks and waveforms by using GEBCO data and ETOPO1 data, in the 2006 tsunami.

Figure 19a. and 19b. show the spatial distribution of maximum wave heights simulated using two bathymetric grids, GEBCO data and ETOPO1 data. To evaluate the differences between modeled wave peaks, the variation and percent change of variation are calculated, which can be defined in equation (12) and (13).

$$Var_{peak} = Peak_{GEBCO} - Peak_{ETOPO1} \tag{12}$$

$$\% Var_{peak} = \frac{Peak_{GEBCO} - Peak_{ETOPO1}}{Peak_{GEBCO}} \times 100 \tag{13}$$

Where $Var_{peak}$ is the variation of modeled wave peaks calculated at each computational grid with GEBCO and ETOPO1 data, $Peak_{GEBCO}$ and $Peak_{ETOPO1}$ are defined as the calculated wave peaks of progressive wave in a unit area of the free surface. Figure 19c and 19 d illustrated the spatial distribution of the variation and percent change of variation of the modeled wave peaks in model domain, indicating the differences of modeled results using two bathymetries. The results suggest that the variation of modeled wave peaks using two bathymetries is more than 0.05 m, and the percent change is more than 50 % to GEBCO's modeled results for the area shallower than 500 m in sea depth.

Figure 20. shows the modeled tsunami waveforms at three coastal stations (i.e., black circles in Figure 19.) using two bathymetric grids. At Kaohsiung, the modeled waveforms from two bathymetries match well to each other, however, the modeled wave peak from ETOPO1 is significantly smaller than GEBCO. The bathymetries from GEBCO and ETOPO1 data can produce the tsunami waveforms at Dongkung and Houbihu similar in both wave periods and peaks. The Table 8. summarized the details of coastal stations and the peak variation percentage of modeled results from two bathymetries.

**6. The mechanism of tsunami wave trapping**

**6.1 Bathymetry effect on tsunami wave directivity**

It is commonly understood that tsunami travel with velocity that is mainly governed by seafloor depths. A tsunami propagates at a slower speed when the tsunami wave enters the shallow water from deeper water. The significant change in propagation speed allows the tsunami to change its wave direction. To assess the

bathymetry effect on tsunami wave directivity during propagation, the simulations were applied using actual (MS) and manipulated bathymetry (EXP1 and EXP2).

Simulated snapshots of tsunami wave propagation using actual (MS) bathymetry are shown in Figure 21. The shelves in the front of Hengchun Peninsula have shallow depths compared to the open ocean. Figure 21a. and b. present how tsunami waves repeatedly changed their directions among the shelves and then refracted into the embayment of west coast. The tsunami waves are reflected from the coast after arrival and tended to be radiated offshore. However, it was not fully radiated offshore, instead, was again reflected at the boundary of the shelf, and refracted north toward as far as Kaohsiung and Dongkung (Figure 21c, d). The high energy waves repeatedly reflected and refracted among the shelves. Only rare tsunamis were transmitted back to the open ocean or to the east coast. These indicated that the tsunami waves are trapped over the shelves during its passage in the 2006 tsunami event. Due to this fluctuation, the high-energy tsunami wave remained along the western coast for a long time, which could be clearly seen at 75 min and 90 min after the occurrence of the first earthquake (Figure 21e, f).

Figure 22. shows the snapshots of simulated tsunami wave propagation using manipulated (EXP1) bathymetry. In this situation, the transmission of tsunami waves in the shallow area are similar to those simulated using the actual (MS) bathymetry, which the tsunami waves are persistent, and repeatedly reflected and refracted among the shelves, but with more reflected waves from the coast are radiated to the open sea (Figure 22b-f). This is because the tsunami source is located at the area with sea depths deeper than 500 m and for bathymetry with sea depths deeper than 500 m are replaced with 500 m depth in this hypothetical situation.

Aside from the numerical experiment EXP1, a rather hypothetical situation (EXP2) was conducted, which simulates the tsunami wave propagation on a bathymetry with flatted sea bottom with sea depth 500 m. Figure 23 shows the snapshots of simulated tsunami wave propagation using manipulated (EXP2) bathymetry. An inspection of tsunami wave transmission in the shallow area indicates the reflected tsunami waves from the coast are radiated homogeneously to offshore and the wave reflection and refraction could not be seen clearly. In addition, the tsunami waves propagate in a rather fast speed (i.e., in comparing to MS and EXP1) and mostly radiated out of the model domain at 75 min and 90 min after the occurrence of the first earthquake (Figure 23 d, e).

**6.2 Tsunami wave energy trapped on the shelf**

While the past section specified that the tsunami waves are trapped over the shelves due to the wave directivity change associated to the configuration of coastal bathymetry, the question remains how much wave energy can be trapped over the shelves in front of southern Taiwan during the passage of tsunami. To quantitatively evaluate the wave energy trapped over the shelves, the trapped ratio is introduced and used to indicate the performance of tsunami energy trapped in bathymetric situations and can be determined by equation (14).

$$R_T = \frac{E_{Shelf}}{E_{Total}} \times 100 \qquad (14)$$

Where $R_T$ is the ratio of tsunami energy trapped, $E_{Shelf}$ is the calculated tsunami potential energy on the shelves (i.e., shallow area with sea depths shallower than 500 m), $E_{Total}$ is the calculated total tsunami potential energy of model domain of each time step. The tsunami potential energy is determined assuming that the energy flux of the tsunami wave progressed in a unit region of free sea surface (Nosov et al., 2014) and can be determined using equation (15).

$$E_p = \oiint \frac{1}{2} \rho g \eta^2 \, dx \, dy \qquad (15)$$

Where $E_p$ is the tsunami potential energy, $\rho$ is the water density of the ocean, $g$ is the gravity acceleration and is set as 9.81 m s$^{-2}$, and $\eta$ represents the surface integral of the ocean surface disturbance at each time step. The trapped ratio of tsunami energy is calculated from the snapshots of tsunami simulation using actual (MS) and manipulated (EXP1 and EXP2) bathymetry. Figure 24. shows the calculated trapped ratio from simulated tsunami propagation snapshots in every 15 min using actual (MS) and manipulated (EXP1 and EXP2) bathymetry. It is note that for calculating trapped ratio from simulations using manipulated bathymetry (EXP1 and EXP2), the shelf region corresponding to actual bathymetry (MS) is used (i.e., the shallow area illustrated by black solid and dashed lines shown in Figure 22. and 23.). According to the equation (14) and (15), the simulations yield a tsunami energy trapped for more than 50 % when using actual bathymetry (MS) and manipulated bathymetry (EXP1), but with a smaller trapped ratio of 20 % when using manipulated bathymetry (EXP2). These quantitatively provide another confirmation to that tsunami wave energy is coastally trapped related to the shape of bathymetry.

**6.3 Comparison of simulated tsunami waveforms**

To understand any significant change on tsunami waveforms can be recognized with and without the wave trapping, the tsunami waveforms simulated from actual (MS) and manipulated bathymetry (EXP1 and EXP2) are compared. Figure 25. shows the simulated tsunami waveforms at three coastal stations in southern Taiwan using actual and manipulated bathymetry.

Using the manipulated bathymetry (EXP1), the first few circles of simulated tsunami waveforms at all stations are well consistent with those simulated using actual bathymetry (MS), but produce the later phase amplitudes slightly smaller. An inspection of simulated waveforms using manipulated bathymetry (EXP2) indicates a sooner arrival time of first wave and a rather smaller amplitudes of later phase to results from simulation using actual (MS) bathymetry. These ensure that the persistent high energy waves along south coast of Taiwan was associated to the mechanism of tsunami wave trapping.

**6.4 Amplified and persisting high energy waves along the coast**

As described in previous sections, the tsunami wave was trapped over the shelves and transmitted along

the coast as edge waves during the 2006 tsunami. This section studies how tsunami waves behaved as the edge waves, and to what extent such wave fluctuations influenced the amplified and persisting high energy waves along the south coast of Taiwan. Figure 26. shows the shelves in front of south Taiwan and the simulated tsunami heights of the 2006 tsunami from main simulation (MS).

To study the behaviors of edge waves along south coast during the 2006 tsunami, the time-distance diagram of tsunami wave is shown. Figure 27a. shows the time-distance diagram of the tsunami wave along the contour of sea depth 20 m (i.e., black dashed line in Figure 26a.). Based on the phase shift of the tsunami wave, the propagation path and the travel time curve of edge waves is illustrated (i.e., green arrow in Figure 27a.). According to the travel time curve, the edge waves propagate along the coast at the speed of 50 m s$^{-1}$. The edge waves propagate along the coast, which iteratively reflected at the shelf edge. The coupling of edge waves and later coming incident waves amplified tsunami waves and persists the wave oscillation in later phase. These are visible from simulated tsunami waveforms at numerical wave gauge C and E, as shown in Figure 27c.

To understand the persisting high energy waves along south coast of Taiwan during the 2006 tsunami, the decreasing tendency of tsunami wave energy along the contour of sea depth 20 m is analyzed. The temporal tsunami wave energy is first determined using equation (11) and then normalized according to the maximum temporal tsunami energy in the time series. Figure 27b. shows the time-distance diagram of the normalized tsunami energy long the contour of sea depth 20 m (i.e., black dashed line in Figure 26a.). Figure 27d. shows the normalized tsunami energy at numerical wave gauge C and E. At numerical wave gauge C, the normalized tsunami energy achieves its greatest value at approximately 40 min after the first earthquake occurrence. However, this high-energy channel did not decrease with time after the first wave arrived, instead, a persisting channel of strong energy was visible. This energy channel lasted for more than 60 min, and the wave energy repeatedly reached the maximum value in this channel. Beyond this channel, the energy commenced to decrease with a rate of energy loss of 50% at 110 min, and 20% at 270 min after the occurrence time of first earthquake. At numerical wave gauge E, the normalized tsunami energy achieves its greatest value approximately 30 min and 120 min after the first wave arrived. Beyond this channel, the energy commenced to decrease with a rather rate of energy loss of 80 % at 150 min, and 70 % at 215 min after the occurrence time of first earthquake. Accordingly, the tsunami decay process in this region is expected to last for more than 300 min. These results indicate that the wave amplification and persistent high energy waves along the coast during the 2006 tsunami were connected to tsunami wave trapping and the influence of edge waves. According to these behaviors, southern Taiwan could be affected by intensified coastal hazards and severe impacts from tsunamis.

**Table 2. Focal mechanisms for successive earthquakes estimated by GCMT and USGS.**

|  |  | Earthquake 1 | | Earthquake 2 | |
|---|---|---|---|---|---|
|  |  | NP1 | NP2 | NP1 | NP2 |
| GCMT | Long (º E) | 120.52 | | 120.4 | |
|  | Lat (º N) | 21.81 | | 22.02 | |
|  | Strike (deg) | 165 | 329 | 151 | 61 |
|  | Dip (deg) | 30 | 61 | 48 | 90 |
|  | Rake (deg) | -76 | -98 | 0 | 138 |
|  | Depth (km) | 20 | | 33 | |
| USGS | Long (º E) | 120.55 | | 120.49 | |
|  | Lat (º N) | 21.8 | | 21.97 | |
|  | Strike (deg) | 171 | 319 | 151 | 61 |
|  | Dip (deg) | 24 | 69 | 48 | 90 |
|  | Rake (deg) | -61 | -102 | 0 | 138 |
|  | Depth (km) | 25 | | 33 | |

[Figure]

**Figure 4. Fault models for the two earthquakes. (a) Sub-fault locations of the first earthquake ($M_w$ 7.0) using NP2 of USGS's moment tensor solution. (b) Sub-fault locations of the second earthquake ($M_w$ 6.9) using NP1 of USGS's moment tensor solution.**

**Table 3. Parameters of sub-faults for the two earthquakes of the 2006 earthquakes doublet**

| | Sub fault | Long (°E) | Lat (°N) | Length (km) | Width (km) | Depth (km) | Strike (°) | Dip (°) | Rake (°) |
|---|---|---|---|---|---|---|---|---|---|
| | 1 | 120.619 | 21.588 | 10 | 10 | 15.3 | 319 | 69 | -102 |
| | 2 | 120.556 | 21.657 | 10 | 10 | 15.3 | 319 | 69 | -102 |
| | 3 | 120.492 | 21.724 | 10 | 10 | 15.3 | 319 | 69 | -102 |
| Earthquake | 4 | 120.429 | 21.792 | 10 | 10 | 15.3 | 319 | 69 | -102 |
| 1 | 5 | 120.692 | 21.648 | 10 | 10 | 24.7 | 319 | 69 | -102 |
| | 6 | 120.629 | 21.716 | 10 | 10 | 24.7 | 319 | 69 | -102 |
| | 7 | 120.565 | 21.784 | 10 | 10 | 24.7 | 319 | 69 | -102 |
| | 8 | 120.501 | 21.852 | 10 | 10 | 24.7 | 319 | 69 | -102 |
| | 1 | 120.726 | 21.989 | 10 | 10 | 29.1 | 151 | 48 | 0 |
| | 2 | 120.642 | 21.946 | 10 | 10 | 29.1 | 151 | 48 | 0 |
| | 3 | 120.557 | 21.902 | 10 | 10 | 29.1 | 151 | 48 | 0 |
| Earthquake | 4 | 120.473 | 21.858 | 10 | 10 | 29.1 | 151 | 48 | 0 |
| 2 | 5 | 120.680 | 22.068 | 10 | 10 | 29.1 | 151 | 48 | 0 |
| | 6 | 120.595 | 22.024 | 10 | 10 | 36.5 | 151 | 48 | 0 |
| | 7 | 120.510 | 21.980 | 10 | 10 | 36.5 | 151 | 48 | 0 |
| | 8 | 120.426 | 21.936 | 10 | 10 | 36.5 | 151 | 48 | 0 |

**Table 4a. Details of average slip, large slip, and background slip for the two earthquakes**

|  | Earthquake 1 | Earthquake 2 |
|---|---|---|
| Moment magnitude ($M_w$) | 7.0 | 6.9 |
| Entire fault size ($km^2$) | 800 | 800 |
| Rigidity (GPa) | 30 | 30 |
| Average slip $D_a$ (m) | 1.66 | 1.17 |
| Large slip $2D_a$ (m) | 3.32 | 2.35 |
| Background slip $D_b$ (m) | 1.11 | 0.78 |

**Table 4b. Asperity location of multiple fault models for the two earthquakes**

| Scenario | Asperity location of Earthquake 1 | | | Asperity location of Earthquake 2 | | |
|---|---|---|---|---|---|---|
|  | North | Central | South | North | Central | South |
| LS1 | ○ |  |  |  | ○ |  |
| LS2 |  | ○ |  |  | ○ |  |
| LS3 |  |  | ○ |  | ○ |  |
| LS4 | ○ |  |  | ○ |  |  |
| LS5 |  | ○ |  | ○ |  |  |
| LS6 |  |  | ○ | ○ |  |  |
| LS7 | ○ |  |  |  |  | ○ |
| LS8 |  | ○ |  |  |  | ○ |
| LS9 |  |  | ○ |  |  | ○ |

[Figure]

**Figure 5. (a) Map of sub-fault boundaries with different asperity location for the first earthquake (M_w 7.0). (b) Co-seismic crustal vertical displacement calculated using the fault parameters of sub-**

**faults. The beachball denotes the focal mechanisms of USGS NP2 nodal planes for the first earthquake. The sub-faults in red represent the large slip areas, and the sub-faults in yellow represent the background slip areas. The large slip area was placed only at shallow parts of entire fault area. The blue stars represent the epicenter of the first earthquake, and the green circles represent the aftershocks. The tide gauge stations are plotted in green triangles.**

[Figure]

**Figure 6. (a) Map of sub-fault boundaries with three different locations of large slip areas for the second earthquake (M$_w$ 6.9). (b) Co-seismic crustal vertical displacement calculated using the fault**

parameters of sub-faults. The beachball denotes the focal mechanisms of USGS NP2 nodal planes for the first earthquake. The sub-faults in red represent the large slip areas, and the sub-faults in yellow represent the background slip areas. The large slip area was placed only at shallow parts of entire fault area. The blue stars represent the epicenter of the first earthquake, and the green circles represent the aftershocks. The tide gauge stations are plotted in green triangles.

[Figure]

**Figure 7. Bathymetry map of model domain from GEBCO and ETOPO1 bathymetry data. The green triangles denote the location of tide gauge stations. The red stars present the epicenters of the two earthquakes.**

[Figure]

**Figure 8. Maps of the manipulated bathymetry of model domain for numerical experiments (a) EXP1 and (b) EXP2.**

**Table 5. Details of the bathymetry data used for numerical experiments MS, EXP1, and EXP2.**

| | Numerical experiments | | |
|---|---|---|---|
| | MS | EXP1 | EXP2 |
| Bathymetry source | | GEBCO data | |
| Grid size | | 450 m | |
| Mesh number (x, y) | | (538, 631) | |
| Description of bathymetry conditions | Sea depths from GEBCO data | Sea depths larger than 500 m were replaced with 500 m depths | Sea depths of entire domain were replaced with 500 m depths. |

[Figure]

**Figure 14. (a) Single fault models with fault dimensions (length × width) of 40 km × 20 km of the first earthquake using GCMT NP2 nodal plane, and the second earthquake using GCMT NP1 nodal plane. The central fault depth of single fault models for first earthquake are set from 15 km, 20 km, 25 km, 35 km, and the central fault depth is fixed at 33 km for the single fault models of second earthquake for tsunami sensitivity test. (b) Observed and simulated tsunami waveforms at the Dongkung and Houbihu stations using single fault models with different central fault depths of the first earthquake.**

[Figure]

**Figure 15. Simple fault models of the first earthquake ($M_w$ 7.0) using the focal mechanisms from GCMT and USGS. The green triangles indicate the tide gauge stations, red stars indicate the epicenter, yellow circles indicate aftershocks, and the black rectangles indicate the fault model.**

[Figure]

**Figure 16. Simple fault models of the second earthquake ($M_w$ 6.9) using the focal mechanisms from GCMT and USGS. The green triangles indicate the tide gauge stations, red stars indicate the epicenter, yellow circles indicate aftershocks, and the black rectangles indicate the fault model.**

**Table 7. Validation of the simulated tsunami waveforms using single fault models with eight different models of focal mechanisms estimated by GCMT and USGS.**

| Scenario | Moment Tensor Solution | Nodal plane | | Misfit of simulated tsunami waveforms |
|---|---|---|---|---|
| | | Earthquake 1 | Earthquake 2 | |
| S1 | GCMT | NP1 | NP1 | 0.591 |
| S2 | | NP1 | NP2 | 0.632 |
| S3 | | NP2 | NP1 | 0.530 |
| S4 | | NP2 | NP2 | 0.661 |
| S5 | USGS | NP1 | NP1 | 0.529 |
| S6 | | NP1 | NP2 | 0.604 |
| S7 | | NP2 | NP1 | 0.493 |
| S8 | | NP2 | NP2 | 0.735 |

[Figure]

**Figure 17. Comparison of simulated tsunami waveforms at Dongkung and Houbihu stations using single fault models with eight different models of focal mechanisms estimated by GCMT and USGS.**

[Figure]

**Figure 18. Comparison of simulated tsunami waveforms at Dongkung and Houbihu stations using 9 cases of multiple fault models (blue solid lines) and single fault model of S7 (red solid lines). The simulated tsunami waveforms using multiple fault model (LS2) was shown in blue dashed lines. The white circles represent the observation data.**

[Figure]

**Figure 19. Simulated maximum tsunami height using open source bathymetry data (a) GEBCO and (b) ETOPO1 data. (c) The variation and (d) the percent variation of simulated maximum tsunami height using two source of bathymetry data. The black circles indicate the location of tide gauge stations. The bathymetry contour is 500 m based on bathymetry data of GEBCO or ETOPO1 data.**

[Figure]

**Figure 20. Simulated tsunami waveforms at (a) Kaohsiung (b) Dongkung, and (c) Houbihu station using two different open source bathymetry GEBCO and ETOPO1 data.**

**Table 8. Details of tide gauge stations for location of simulating tsunami waveforms and misfit of model results using different open source bathymetry data.**

| Station | Sea depth (m) | | Simulated wave peak (m) | | $Var_{peak}$ | $\%Var_{peak}$ |
|---------|-------|--------|-------|--------|--------------|----------------|
|         | GEBCO | ETOPO1 | GEBCO | ETOPO1 |              |                |
| Kaohsiung | 10 | 8 | 0.163 | 0.084 | 0.079 | 48.45 |
| Dongkung | 9 | 14 | 0.171 | 0.17 | 0.001 | 0.58 |
| Houbihu | 4 | 11 | 0.493 | 0.414 | 0.079 | 16.02 |

[Figure]

**Figure 21. Tsunami propagation snapshots from the numerical experiment MS. The tide gauge stations are plotted in green triangles. The bathymetry contour is 500 m.**

[Figure]

**Figure 22. Tsunami propagation snapshots from the numerical experiment EXP1. The tide gauge stations are plotted in green triangles. The bathymetry contour of 500 m depth is shown in gray solid line.**

[Figure]

**Figure 23. Tsunami propagation snapshots from the numerical experiment EXP2. The tide gauge stations are plotted in green triangles. The corresponding bathymetry contour of 500 m depth from GEBGO data is shown in gray dashed line.**

[Figure]

**Figure 24. Trapped ratio calculated from tsunami propagation snapshots in every 15 min from numerical experiment (a) MS, (b) EXP1, and (c) EXP2.**

[Figure]

**Figure 25. Simulated tsunami waveforms at (a) Kaohsiung (b) Dongkung, and (c) Houbihu station from numerical experiment MS, EXP1, and EXP2.**

[Figure]

**Figure 26. Zoomin map of (a) the bathymetry around southern Taiwan, and (b) the simulated maximum tsunami height using multiple fault model (LS2). Green triangles indicate the locations of tide gauge stations, and pink circles denote numerical wave gauges at sea depth 20 m. The white solid lines are contour lines, and the black dashed line presents the bathymetric contour at a depth of 20 m.**

[Figure]

**Figure 27.** Time-distance diagram of (a) tsunami wave and (b) normalized energy along the bathymetry contour of 20 m from numerical wave gauge A to F, and time series measurements of (c) tsunami amplitude and (d) normalized energy at numerical wave gauge C and E. The black dashed lines indicate the distances of numerical wave gauge C and E from A. For interpretation of the references, the reader is referred to the Figure 26a.

---

## Author Comment (AC2)

**Dear Editor,**

Thank you for the time and sending us your decision. We have made responses and corrections to reviewers' comments and suggestions as shown below. Corrections made based on comments and suggestions are shown in red.

**Reply to reviewer no. 2**

We highly appreciate your time spent in reviewing the manuscript as well as your valuable comments and suggestions. We are glad that you are interested in our work and your positive feedback. Please find our line-by-line responses and corrections to your comments and suggestions. All responses, corrections and improvements are shown in red in the revised manuscript.

| Reply to general comments |
|---|
| We apologize for the English issues and spelling errors on the manuscript. In order to this, the manuscript was carefully re-written, and the English spellings were to our best to be improved. Please see more detail below on our answers and responses. The revised English in the article will be shown in the revised manuscript. |

| Reviewer comments | Our answers | Corrected manuscript |
|---|---|---|
| Title: As the two earthquakes have different magnitudes (M6.9 and M7.0), I think they cannot be called doublet. Usually doublet us used for two earthquakes with the same size that occur with short interval. You can simply say "…by two Mw6.9 and Mw7.0 consecutive earthquakes". | We apologize for our confusing expression. We added some more information in seismological perspective of view to improve the clarity. The two successive earthquakes are suggested to be $M_L = 7.0$ ($M_w = 7.0$ in the Global CMT catalog) for the former, and $M_L = 7.0$ ($M_w = 6.9$ in the Global CMT catalog) for the latter. From seismological perspective of view, pairs of large earthquakes with equivalent rupture size and occurred in a | **From line 50 to line 56:** The respective magnitudes of these two earthquakes were suggested to be $M_L = 7.0$ ($M_w = 7.0$ in the Global CMT catalog) for the former, and $M_L = 7.0$ ($M_w = 6.9$ in the Global CMT catalog) for the latter. From seismological perspective of view, pairs of large earthquakes with equivalent rupture size and occurred in a similar spatial and temporal proximity were specified as doublet (Lay and Kanamori, 1980; Kagan and |

| | | |
|---|---|---|
| | similar spatial and temporal proximity were specified as doublet (Lay and Kanamori, 1980; Kagan and Jackson, 1999). Sharing comparable earthquake magnitudes, and very close epicenters and occurrence times, the successive earthquakes are referred as an event of doublet (Ma and Liang, 2008; Wu et al., 2009). | Jackson, 1999). Sharing comparable earthquake magnitudes, and very close epicenters and occurrence times, the successive earthquakes are referred as an event of doublet (Ma and Liang, 2008; Wu et al., 2009). |
| L16: waveforms and conducted numerical simulations… | We apologized for the English errors made in the manuscript. We corrected it. | Please see **Line 16 to 17.** This study analyzed tide gauge tsunami waveforms and conducted numerical simulations to understand the source characteristics and resulting tsunami behaviors. |
| L39: and to cause severe …. | We apologized for the English errors made in the manuscript. We corrected it. | The Manila Trench and Ryukyu Trench are located offshore Taiwan, and have been identified as hazardous tsunamigenic regions, as both have the potential to generate megathrust earthquakes and to cause severe tsunami impacts on coast plains (Liu et al., 2009; Megawati et al., 2009; Wu and Huang, 2009; Li et al., 2016; Sun et al., 2018; Qiu et al., 2019). |

---

## Author Comment (AC3)

**Dear Editor,**

Thank you for the time and sending us your decision. We have made responses and corrections to reviewers' comments and suggestions as shown below. Corrections made based on comments and suggestions are shown in red.

**Reply to reviewer no. 1**

We highly appreciate your time spent in reviewing the manuscript as well as your valuable comments and suggestions. We are glad that you are interested in our work and your positive feedback. Please find our line-by-line responses and corrections to your comments and suggestions. All responses, corrections and improvements are shown in red in the revised manuscript.

**Reply to general comments**

Thank you very much for pointing out these important issues. We totally agreed that the sensitivity and variability aspects of the source models and the bathymetry should be sufficiently discussed, Also, additional investigations should be applied to strengthen the conclusion related to tsunami wave trapping. In order to this, we have applied additional analyses mainly in section 2.4, 2.5 and 2.6, and related sections 5.1, 5.2 and 5.3.

In addition, to improve the clarity of the text, we have added more explanations to section 2.3, 6.1, 6.2, 6.3, as well as additional Tables and Figures to support the explanations. The manuscript was carefully re-written, and the English spellings were to our best to be improved. Please see more details below on our answers and responses.

| Reviewer comments                  | Our answers                        | Corrected manuscript                     |
|------------------------------------|------------------------------------|------------------------------------------|
| Line 15: Please remove 'for the    | We thank the reviewer for          | Line 14:                                 |
| first time'                        | pointing this out. We corrected it | A small tsunami was generated,           |
|                                    | by removing the word.              | and recorded at tide gauge               |
|                                    |                                    | stations <del>for the first time</del> . |
| Line 44: I suggest putting in a    | We thank and agree with the        | From line 48 to line 49:                 |
| reference to Figure 1 already      | reviewer. We corrected it by       | The locations of Hengchun                |
| here.                              | linking a reference to Figure 1.   | Peninsula and the epicenters of          |
|                                    |                                    | the successive earthquakes are           |
|                                    |                                    | shown in Figure 1.                       |
| Line 51: The Lay and Kanamori      | We thank the reviewer for          | From line 50 to line 56:                 |
| refence is general but the way the | pointing this out. The sentence    | The respective magnitudes of             |

| sentence reads it sounds like the     | was rephrased, and additional    | these two earthquakes were           |
|---------------------------------------|----------------------------------|--------------------------------------|
| paper refers to this event. Please    | references about earthquakes     | suggested to be $M_L = 7.0 \ (M_w =$ |
| rephrase, and include a specific      | doublet in seismological         | 7.0 in the Global CMT catalog)       |
| reference work (e.g. from             | perspective of view were         | for the former, and $M_L = 7.0 (M_w$ |
| seismology) that consider the         | included.                        | = 6.9 in the Global CMT catalog)     |
| 2006 event in particular.             |                                  | for the latter. From seismological   |
|                                       |                                  | perspective of view, pairs of        |
|                                       |                                  | large earthquakes with               |
|                                       |                                  | equivalent rupture size and          |
|                                       |                                  | occurred in a similar spatial and    |
|                                       |                                  | temporal proximity were              |
|                                       |                                  | specified as doublet (Lay and        |
|                                       |                                  | Kanamori, 1980; Kagan and            |
|                                       |                                  | Jackson, 1999). Sharing              |
|                                       |                                  | comparable earthquake                |
|                                       |                                  | magnitudes, and very close           |
|                                       |                                  | epicenters and occurrence times,     |
|                                       |                                  | the successive earthquakes are       |
|                                       |                                  | referred as an event of doublet      |
|                                       |                                  | (Ma and Liang, 2008; Wu et al.,      |
|                                       |                                  | 2009).                               |
| Line 51: 'Casualties', do you         | Thank you very much for the      |                                      |
| mean 'fatalities'? The former also    | suggestion. According to the     |                                      |
| refer to injuries, the latter only to | report of National Disaster      |                                      |
| loss of life.                         | prevention and Protection        |                                      |
|                                       | Commission, R.O.C., 2007, the    |                                      |
|                                       | 26 December 2006 earthquakes     |                                      |
|                                       | caused 44 injuries, including 2  |                                      |
|                                       | fatal ones, 3 building collapse, |                                      |
|                                       | and massive damages of           |                                      |
|                                       | submarine communication          |                                      |
|                                       | cables. To that sense, we        |                                      |
|                                       | considered to use the vocabulary |                                      |
|                                       | 'Casualties' here.               |                                      |
| Line 57: 'propagated toward' à        | We are very sorry for making     | Line 61:                             |
| 'propagated towards'                  | this spelling mistake. We        | A small tsunami was generated        |

| -                                  | r                                   | r                                    |
|------------------------------------|-------------------------------------|--------------------------------------|
|                                    | corrected it.                       | after the successive strong          |
|                                    |                                     | motions of these earthquakes.        |
|                                    |                                     | The tsunami propagated               |
|                                    |                                     | towards, and reached the western     |
|                                    |                                     | coast of southern Taiwan             |
|                                    |                                     | immediately after the                |
|                                    |                                     | earthquakes.                         |
| Line 60: Rephrase sentence, my     | We thank and agreed with the        | Line 64-66:                          |
| suggestion 'as it was rare because | reviewer. We corrected it by        | The December 2006 tsunami            |
| it was generated by earthquakes    | rephrasing the sentence.            | was an important event and           |
| in short succession'.              |                                     | attracted public interest, as it was |
|                                    |                                     | rare because it was generated by     |
|                                    |                                     | earthquakes in short succession,     |
|                                    |                                     | and was a new issue among            |
|                                    |                                     | social communities and ordinary      |
|                                    |                                     | persons in Taiwan about              |
|                                    |                                     | tsunamis.                            |
| Line 62: 'heightens' à 'increased' | We thank the reviewer for           | Line 67:                             |
| 6                                  | pointing this out. We corrected it. | This recent tsunami not only         |
|                                    | r 8                                 | corroborates the tsunami risk in     |
|                                    |                                     | Taiwan, but also increased the       |
|                                    |                                     | awareness of disaster risk           |
|                                    |                                     | management. such as                  |
|                                    |                                     | preparedness and mitigation          |
|                                    |                                     | countermeasures for the next         |
|                                    |                                     | tsunamis                             |
| Line 65: Several repeats of the    | We thank the reviewer for           | Please see line 69            |
| above in this paragraph I          | pointing this out We shortened      | The tsunami observations             |
| suggest shortening                 | the paragraph                       | reported following the 26            |
| suggest shortening.                | the paragraph.                      | December 2006 teunomi also           |
|                                    |                                     | December 2000 tsunann also           |
| Line 67. Dieses delete contant     | We thenk the merioder for           | Jine 70.                             |
| Line 0/: Please delete sentence    | we thank the reviewer for           | Line /U:                             |
| starting with 'It has been         | pointing this out. The sentence     | First, the first tsunami wave crest  |
| common understanding'. This        | starting with 'It has been          | was not shown as the largest in      |
| can certainly be disputed and the  | common understanding' have          | some stations.                       |
| scientific community is            | been deleted, and the sentences     |                                      |

| definitely aware that later wave  | were rephrased.                  |                                   |
|-----------------------------------|----------------------------------|-----------------------------------|
| arrivals can be larger than the   |                                  |                                   |
| first.                            |                                  |                                   |
| Line 71: 'prolonged'? Prolonged   | We apologize for our confusing   | Line 72 to line 73:               |
| compared to what?                 | expression. We meant that some   | Second, tsunami durations for     |
|                                   | stations recorded the tsunami    | more than 6 h were recorded at    |
|                                   | durations for more than 6 hours  | some stations following the       |
|                                   | during the 2006 earthquake       | earthquakes.                      |
|                                   | tsunami. We have removed the     |                                   |
|                                   | word 'prolonged', and rephrased  |                                   |
|                                   | the sentence to improve the lack |                                   |
|                                   | clarity.                         |                                   |
| Lines 80-81: Something is         | We thank the reviewer for        | Please see line 77-88             |
| missing in these statements,      | pointing this out. We rephrase   | The other issue was that which    |
| please rephrase so the meaning is | the sentences and the meaning.   | source models could better        |
| more apparent.                    |                                  | explain the successive tsunamis   |
|                                   |                                  | to the recorded observations in   |
|                                   |                                  | southern Taiwan. Wu et al., 2008  |
|                                   |                                  | simulated the tsunami from this   |
|                                   |                                  | event using single fault models.  |
|                                   |                                  | They numerically computed the     |
|                                   |                                  | tsunami propagation on a nested   |
|                                   |                                  | grid system with finest grids of  |
|                                   |                                  | 0.125 min resolution bathymetry   |
|                                   |                                  | data and compared their results   |
|                                   |                                  | with observation data from tide   |
|                                   |                                  | gauge stations. Although the      |
|                                   |                                  | source models to this tsunami     |
|                                   |                                  | event have been specified and     |
|                                   |                                  | modeled in previous study, the    |
|                                   |                                  | uncertainty and variability       |
|                                   |                                  | aspects of the source models and  |
|                                   |                                  | bathymetry have not been          |
|                                   |                                  | investigated thoroughly. Such     |
|                                   |                                  | uncertainties in earthquake fault |
|                                   |                                  | parameters and significant        |

|                                   |                                   | 1                                                     |
|-----------------------------------|-----------------------------------|-------------------------------------------------------|
|                                   |                                   | difference among the open-
source bathymetries can |
|                                   |                                   | exaggerate the modeled results                        |
|                                   |                                   | rather than the predictions from                      |
|                                   |                                   | previous study to the 2006                            |
|                                   |                                   | tsunami. Therefore, it is critical                    |
|                                   |                                   | to discuss such model's                               |
|                                   |                                   | performances from viewpoint of                        |
|                                   |                                   | sensibility perspective because it                    |
|                                   |                                   | is desirable to obtain a tsunami                      |
|                                   |                                   | source model and to understand                        |
|                                   |                                   | the reliability of bathymetry data                    |
|                                   |                                   | utilized for numerical simulation                     |
|                                   |                                   | for reasonably estimating the                         |
|                                   |                                   | tsunami wave activities during                        |
|                                   |                                   | the 2006 tsunami.                                     |
| Line 91: 'justify' à 'hindcast'   | We thank the reviewer for         | Line 98 to line 101:                                  |
|                                   | pointing this out. We rephased    | The December 2006 earthquake                          |
|                                   | the sentence.                     | tsunami represents a unique and                       |
|                                   |                                   | recent incident in Taiwan;                            |
|                                   |                                   | therefore, these findings could                       |
|                                   |                                   | not only help further clarify                         |
|                                   |                                   | tsunami generation and the                            |
|                                   |                                   | important behaviors responsible                       |
|                                   |                                   | for tsunami hazards facing the                        |
|                                   |                                   | island of Taiwan but also have                        |
|                                   |                                   | implications for tsunami                              |
|                                   |                                   | warning and disaster risk                             |
|                                   |                                   | management.                                           |
| Line 99: Please delete 'In        | We thank and agreed with the      | Line 105 to line 106:                                 |
| general', and replace the         | point of view of the reviewer. We | Time history data of sea levels                       |
| statement 'possible method to     | corrected it by rephrasing the    | recorded at coastal sites provide                     |
| study' with 'one source of        | sentence.                         | one source of information that                        |
| information we can use to study'. |                                   | we can use to study tsunami                           |
| The point is that it can only be  |                                   | patterns.                                             |
| supplementary to other methods    |                                   |                                                       |

| it is usually not enough by itself. |                                     |                                    |
|-------------------------------------|-------------------------------------|------------------------------------|
| Line 112: 'represent the duration'  | We thank the reviewer for           | Line 118:                          |
| à 'represent the observation'       | pointing this out. We corrected it. | The tsunami durations represent    |
| (duration written twice in          |                                     | the observation time of high-      |
| sentence)                           |                                     | energy tsunami waves persisting    |
|                                     |                                     | in a coastal site of observation.  |
| Line 113: Remove 'of                | We thank the reviewer for           | Line 118 to line 121:              |
| observation'. 'duration' à          | pointing this out. We corrected it. | The tsunami durations represent    |
| 'durations', and 'was' à 'were'     |                                     | the observation time of high-      |
|                                     |                                     | energy tsunami waves persisting    |
|                                     |                                     | at a coastal site. The tsunami     |
|                                     |                                     | durations at all the stations were |
|                                     |                                     | identified based on a calculation  |
|                                     |                                     | of root mean square (RMS) sea      |
|                                     |                                     | levels, indicating the elapsed     |
|                                     |                                     | time of the wave amplitude         |
|                                     |                                     | above the normal oscillation       |
|                                     |                                     | level before the tsunami wave      |
|                                     |                                     | arrived (Heidarzadeh, 2021).       |
| Line 127: 'The' Fourier analysis    | We thank the reviewer for           | Line 132-136:                      |
|                                     | pointing this out. We corrected it. | The Fourier analysis and the       |
|                                     |                                     | wavelet (time-frequency)           |
|                                     |                                     | analysis. The Fourier analysis is  |
|                                     |                                     | based on the fast Fourier          |
|                                     |                                     | transform (FFT) algorithm,         |
|                                     |                                     | applied based on the updated       |
|                                     |                                     | open-source library Numpy in       |
|                                     |                                     | the Python package (Harris et al., |
|                                     |                                     | 2020). The Fourier analysis was    |
|                                     |                                     | performed to estimate the          |
|                                     |                                     | spectral components of the time    |
|                                     |                                     | history data of the tsunami        |
|                                     |                                     | waveform.                          |
| Line 137: 'the' wavelet analysis    | We thank the reviewer for           | Line 133:                          |
|                                     | pointing this out. We corrected it. | The Fourier analysis and the       |
|                                     |                                     | wavelet (time-frequency)           |

|                                   |                                                       | analysis.                         |
|-----------------------------------|-------------------------------------------------------|-----------------------------------|
| Line 144: The first sentence in   | Thank you very much for the                           | Please see section 2.3 (from line |
| the paragraph is somewhat         | valuable comments. We                                 | 149-177)                          |
| misleading. I would rather say it | rephrased it to improve the                           |                                   |
| is a computer-based method        | clarity of the numerical methods.                     |                                   |
| describing the equations of       |                                                       |                                   |
| motion for the tsunami wave       |                                                       |                                   |
| propagation. You could also add   |                                                       |                                   |
| that there are various methods,   |                                                       |                                   |
| but that the shallow water model  |                                                       |                                   |
| is most used, although dispersive |                                                       |                                   |
| models are more and more used     |                                                       |                                   |
| as well.                          |                                                       |                                   |
| Line 149: I would say that        | Thank you very much for the                           | Please see section 2.3 (from line |
| TUNAMI also cover far-field       | valuable comments. We add                             | 149 to 177)                       |
| tsunamis, with limitations of     | additional information to this                        |                                   |
| course.                           | part.                                                 |                                   |
| Line 155: You do not describe     | We simulated the tsunami                              | Please see section 2.3 (from line |
| mesh refinement anywhere.         | propagation using a 450 m                             | 149 to 177)                       |
| How do you ensure                 | bathymetric grid. The mesh size                       |                                   |
| convergence? What is your grid    | in x and y directions are 538 and                     |                                   |
| resolution, and what exactly is   | 631. The CFL condition is                             |                                   |
| the CFL number? It should be a    | presented as:                                         |                                   |
| minimum to test convergence at    | $\Delta t \leq \Delta x$                              |                                   |
| least with two different          | $\Delta t \leq \sqrt{2gh_{max}}$                      |                                   |
| (optimally three) mesh sizes.     | Where the $\Delta t$ is the time interval,            |                                   |
|                                   | $\Delta x$ is the grid spacings, and $h_{\text{max}}$ |                                   |
|                                   | is the maximum water depth in                         |                                   |
|                                   | the model domain.                                     |                                   |
| Line 160: You have stated this    | We thank and agreed with the                          |                                   |
| before. I suggest to delete this  | reviewer. We deleted the                              |                                   |
| sentence that only repeats what   | sentence.                                             |                                   |
| is already written in the intro.  |                                                       |                                   |
| Line 168: Are you simulating      | Thank you very much for the                           | For the approach, please see      |
| with uniform slip? Could you      | valuable suggestions. The                             | section 2.4.2 (from line 220 to   |
| gain anything with adding non-    | tsunami sensitivity to non-                           | 248) and for the results of       |

| uniform slide and simulate          | uniform fault slip distribution is | sensitivity analysis, please see  |
|-------------------------------------|------------------------------------|-----------------------------------|
| different realisations of the slip  | evaluated.                         | section 5.2 (from line 464 to     |
| distribution? This deserves to be   |                                    | 478)                              |
| discussed more.                     |                                    |                                   |
| Line 186: 'horizontal effect' à     | We appreciated the reviewer for    | Please see line 175-176:          |
| 'horizontal deformation             | the correction. The sentence was   | The horizontal deformation        |
| contribution to tsunami             | revised.                           | contribution to tsunami           |
| generation'                         |                                    | generation on the steep           |
|                                     |                                    | bathymetric slopes (Tanioka and   |
|                                     |                                    | Satake, 1996) was included.       |
| Line 191: Why could this not        | The statement was skipped.         |                                   |
| have been caused by landslides?     |                                    |                                   |
| Please elaborate / substantiate, or |                                    |                                   |
| otherwise skip this statement if    |                                    |                                   |
| you cannot back it up more          |                                    |                                   |
| explicitly.                         |                                    |                                   |
| Line 193: Add 'simulated' before    | The vocabulary was revised.        | Please see line 173-174:          |
| 'initial'.                          |                                    | As the simulated initial          |
|                                     |                                    | condition inputted for numerical  |
|                                     |                                    | tsunami simulation, the initial   |
|                                     |                                    | water level distribution is       |
|                                     |                                    | calculated from the earthquake    |
|                                     |                                    | fault parameters using the theory |
|                                     |                                    | of Okada, 1985.                   |
| Line 203: You may need to           | For the bathymetric scenarios      | For the clarity of bathymetric    |
| elaborate what you mean by 'two     | stated here, we meant the actual   | scenarios, please see section 2.6 |
| bathymetric scenarios'. You         | and manipulated bathymetries       | (from line 276 to 291). The       |
| probably mean tsunami               | used in numerical simulations to   | details of actual and manipulated |
| simulations applying two            | examine the how bathymetry can     | bathymetries used in numerical    |
| different bathymetries. You may     | influence the tsunami wave         | simulations were summarized in    |
| motivate your work by               | directivity and wave trapping.     | Table 5.                          |
| mentioning how wrong the open       | In addition, the variability       | For the examination of tsunami    |
| source bathy was for 2018 Palu.     | aspects of open source             | sensitivity to open source        |
| Similar for 2018 Anak Krakatoa      | bathymetry to model results was    | bathymetry, the 2018 Palu and     |
| (e.g. Zengaffinen et al., 2021).    | examined.                          | the 2018 Anak Krakatoa tsunami    |
|                                     |                                    | were referred as backgrounds      |

| Line 207: Both are scenarios in a
way. I would rephrase, and rather
say 'manipulated bathymetry'
rather than 'hypothetical
scenario'.                                                          | We appreciated the reviewer for
the comments. The sentences
were rephrased.                                                                                                                                                                                                                                                                 | and the approach and results
could be found in section 2.5
(from line 250 to 274) and
section 5.3 (from 480 to 502),
respectively.
Please see section 2.6 (from line
276-291).                                                           |
|------------------------------------------------------------------------------------------------------------------------------------------------------------------------------------------------------------|---------------------------------------------------------------------------------------------------------------------------------------------------------------------------------------------------------------------------------------------------------------------------------------------------------------------------------------------------|------------------------------------------------------------------------------------------------------------------------------------------------------------------------------------------------------------------------------------------------------------|
| Line 211: You only investigate
two different bathymetries, and
this might be a bit thin to
conclude in general. I suggest
that the uncertainty related to the
bathymetry is discussed more. | Thank you very much for the valuable suggestions. We agreed with the reviewer.
In addition to the two different bathymetries (i.e., actual and manipulated bathymetry by replacing sea depths larger than 500 m to 500 m), a rather hypothetical situation was examined using the manipulated bathymetry of flatted sea bottom of 500 m depth. | Please see section 2.6 (from line 276-291) and section 6.1 (from line 505-535).                                                                                                                                                                            |
| Line 231: Please rephrase
'different mechanism of tsunami
waves was' à 'different
propagation effects were'                                                                                       | We appreciated the reviewer for
pointing this out. The sentence
was revised.                                                                                                                                                                                                                                                                | Please see line 307 top line 308:
These results suggest that the
different propagation effects
were active at these coastal sites
during the passage of the 2006
tsunami.                                                                   |
| Line 237: The aspects of the
wave recordings should be move
more up front, at least within this
subsection, it is important
background.                                                        | We appreciated the reviewer for
the valuable comments. The
aspects of the wave recordings
were moved and considered as
important background for
simulating scenarios with non-
uniform fault slip distributions.                                                                                                                | Please see line 455-462.
While the single fault models
can produce the simulated
tsunami waveforms well
consistent to the observations,
the badly sampled (i.e., 6 min
interval) signals recorded in
coastal stations also raise some |

|                                                                                                                                                               |                                                                                                                                                                                                                                                                                                                                                                                                                                                                                  | questions, as one would expect                                                                                                                                                                                                                                                                                                                                                                                                                                                                                                   |
|---------------------------------------------------------------------------------------------------------------------------------------------------------------|----------------------------------------------------------------------------------------------------------------------------------------------------------------------------------------------------------------------------------------------------------------------------------------------------------------------------------------------------------------------------------------------------------------------------------------------------------------------------------|----------------------------------------------------------------------------------------------------------------------------------------------------------------------------------------------------------------------------------------------------------------------------------------------------------------------------------------------------------------------------------------------------------------------------------------------------------------------------------------------------------------------------------|
|                                                                                                                                                               |                                                                                                                                                                                                                                                                                                                                                                                                                                                                                  | some potential high tsunami                                                                                                                                                                                                                                                                                                                                                                                                                                                                                                      |
|                                                                                                                                                               |                                                                                                                                                                                                                                                                                                                                                                                                                                                                                  | waves behind the observed                                                                                                                                                                                                                                                                                                                                                                                                                                                                                                        |
|                                                                                                                                                               |                                                                                                                                                                                                                                                                                                                                                                                                                                                                                  | signals. To that sense,                                                                                                                                                                                                                                                                                                                                                                                                                                                                                                          |
|                                                                                                                                                               |                                                                                                                                                                                                                                                                                                                                                                                                                                                                                  | overestimation of modeled                                                                                                                                                                                                                                                                                                                                                                                                                                                                                                        |
|                                                                                                                                                               |                                                                                                                                                                                                                                                                                                                                                                                                                                                                                  | results was expected, but the                                                                                                                                                                                                                                                                                                                                                                                                                                                                                                    |
|                                                                                                                                                               |                                                                                                                                                                                                                                                                                                                                                                                                                                                                                  | simulated tsunami waveforms                                                                                                                                                                                                                                                                                                                                                                                                                                                                                                      |
|                                                                                                                                                               |                                                                                                                                                                                                                                                                                                                                                                                                                                                                                  | using single fault models present                                                                                                                                                                                                                                                                                                                                                                                                                                                                                                |
|                                                                                                                                                               |                                                                                                                                                                                                                                                                                                                                                                                                                                                                                  | the opposite. This indicates that                                                                                                                                                                                                                                                                                                                                                                                                                                                                                                |
|                                                                                                                                                               |                                                                                                                                                                                                                                                                                                                                                                                                                                                                                  | the single fault models (i.e., with                                                                                                                                                                                                                                                                                                                                                                                                                                                                                              |
|                                                                                                                                                               |                                                                                                                                                                                                                                                                                                                                                                                                                                                                                  | uniform fault slip) may not be                                                                                                                                                                                                                                                                                                                                                                                                                                                                                                   |
|                                                                                                                                                               |                                                                                                                                                                                                                                                                                                                                                                                                                                                                                  | sufficient and the asperity area                                                                                                                                                                                                                                                                                                                                                                                                                                                                                                 |
|                                                                                                                                                               |                                                                                                                                                                                                                                                                                                                                                                                                                                                                                  | (i.e., with large fault slip) on the                                                                                                                                                                                                                                                                                                                                                                                                                                                                                             |
|                                                                                                                                                               |                                                                                                                                                                                                                                                                                                                                                                                                                                                                                  | fault should be evaluated. The                                                                                                                                                                                                                                                                                                                                                                                                                                                                                                   |
|                                                                                                                                                               |                                                                                                                                                                                                                                                                                                                                                                                                                                                                                  | tsunami sensitivity to asperity                                                                                                                                                                                                                                                                                                                                                                                                                                                                                                  |
|                                                                                                                                                               |                                                                                                                                                                                                                                                                                                                                                                                                                                                                                  | locations of multiple fault                                                                                                                                                                                                                                                                                                                                                                                                                                                                                                      |
|                                                                                                                                                               |                                                                                                                                                                                                                                                                                                                                                                                                                                                                                  | models will be discussed in next                                                                                                                                                                                                                                                                                                                                                                                                                                                                                                 |
|                                                                                                                                                               |                                                                                                                                                                                                                                                                                                                                                                                                                                                                                  | section.                                                                                                                                                                                                                                                                                                                                                                                                                                                                                                                         |
|                                                                                                                                                               |                                                                                                                                                                                                                                                                                                                                                                                                                                                                                  |                                                                                                                                                                                                                                                                                                                                                                                                                                                                                                                                  |
| Line 254: You say 'abnormally                                                                                                                                 | We apologize for our confusing                                                                                                                                                                                                                                                                                                                                                                                                                                                   | Please see line 326-328                                                                                                                                                                                                                                                                                                                                                                                                                                                                                                          |
| Line 254: You say 'abnormally long', but compared to what?                                                                                                    | We apologize for our confusing
expression. We meant that                                                                                                                                                                                                                                                                                                                                                                                                                      | Please see line 326-328
The calculated tsunami duration                                                                                                                                                                                                                                                                                                                                                                                                                                                                       |
| Line 254: You say 'abnormally
long', but compared to what?                                                                                                 | We apologize for our confusing
expression. We meant that
Kaohsiung and Houbihu station                                                                                                                                                                                                                                                                                                                                                                                     | Please see line 326-328
The calculated tsunami duration
at Dongkung was as much as 3.9                                                                                                                                                                                                                                                                                                                                                                                                                                     |
| Line 254: You say 'abnormally
long', but compared to what?                                                                                                 | We apologize for our confusing
expression. We meant that
Kaohsiung and Houbihu station
recorded the tsunami durations                                                                                                                                                                                                                                                                                                                                                   | Please see line 326-328
The calculated tsunami duration
at Dongkung was as much as 3.9
h, while the tsunami continued                                                                                                                                                                                                                                                                                                                                                                                                   |
| Line 254: You say 'abnormally
long', but compared to what?                                                                                                 | We apologize for our confusing
expression. We meant that
Kaohsiung and Houbihu station
recorded the tsunami durations
for more than 6 hours during the                                                                                                                                                                                                                                                                                                               | Please see line 326-328
The calculated tsunami duration
at Dongkung was as much as 3.9
h, while the tsunami continued
for more than 6 h in Kaohsiung                                                                                                                                                                                                                                                                                                                                                                 |
| Line 254: You say 'abnormally
long', but compared to what?                                                                                                 | We apologize for our confusing
expression. We meant that
Kaohsiung and Houbihu station
recorded the tsunami durations
for more than 6 hours during the
2006 tsunami. We have removed                                                                                                                                                                                                                                                                              | Please see line 326-328
The calculated tsunami duration
at Dongkung was as much as 3.9
h, while the tsunami continued
for more than 6 h in Kaohsiung
and Houbihu.                                                                                                                                                                                                                                                                                                                                                 |
| Line 254: You say 'abnormally
long', but compared to what?                                                                                                 | We apologize for our confusing
expression. We meant that
Kaohsiung and Houbihu station
recorded the tsunami durations
for more than 6 hours during the
2006 tsunami. We have removed
the word 'prolonged', and                                                                                                                                                                                                                                                 | Please see line 326-328
The calculated tsunami duration
at Dongkung was as much as 3.9
h, while the tsunami continued
for more than 6 h in Kaohsiung
and Houbihu.                                                                                                                                                                                                                                                                                                                                                 |
| Line 254: You say 'abnormally
long', but compared to what?                                                                                                 | We apologize for our confusing
expression. We meant that
Kaohsiung and Houbihu station
recorded the tsunami durations
for more than 6 hours during the
2006 tsunami. We have removed
the word 'prolonged', and
rephrased the sentence to                                                                                                                                                                                                                    | Please see line 326-328
The calculated tsunami duration
at Dongkung was as much as 3.9
h, while the tsunami continued
for more than 6 h in Kaohsiung
and Houbihu.                                                                                                                                                                                                                                                                                                                                                 |
| Line 254: You say 'abnormally
long', but compared to what?                                                                                                 | We apologize for our confusing
expression. We meant that
Kaohsiung and Houbihu station
recorded the tsunami durations
for more than 6 hours during the
2006 tsunami. We have removed
the word 'prolonged', and
rephrased the sentence to
improve the lack clarity.                                                                                                                                                                                       | Please see line 326-328
The calculated tsunami duration
at Dongkung was as much as 3.9
h, while the tsunami continued
for more than 6 h in Kaohsiung
and Houbihu.                                                                                                                                                                                                                                                                                                                                                 |
| Line 254: You say 'abnormally
long', but compared to what?
Line 271: What does the                                                                      | We apologize for our confusing
expression. We meant that
Kaohsiung and Houbihu station
recorded the tsunami durations
for more than 6 hours during the
2006 tsunami. We have removed
the word 'prolonged', and
rephrased the sentence to
improve the lack clarity.
We apologize for our lack                                                                                                                                                          | Please see line 326-328
The calculated tsunami duration
at Dongkung was as much as 3.9
h, while the tsunami continued
for more than 6 h in Kaohsiung
and Houbihu.
Please see line 346 to line 350                                                                                                                                                                                                                                                                                                              |
| Line 254: You say 'abnormally
long', but compared to what?
Line 271: What does the
background spectra contain? Are                                   | We apologize for our confusing
expression. We meant that
Kaohsiung and Houbihu station
recorded the tsunami durations
for more than 6 hours during the
2006 tsunami. We have removed
the word 'prolonged', and
rephrased the sentence to
improve the lack clarity.
We apologize for our lack
expression. The background                                                                                                                            | Please see line 326-328         The calculated tsunami duration         at Dongkung was as much as 3.9         h, while the tsunami continued         for more than 6 h in Kaohsiung         and Houbihu.         Please see line 346 to line 350         The background spectra are the                                                                                                                                                                                                                                         |
| Line 254: You say 'abnormally
long', but compared to what?
Line 271: What does the
background spectra contain? Are
they de-tided? Please clarify. | We apologize for our confusing
expression. We meant that
Kaohsiung and Houbihu station
recorded the tsunami durations
for more than 6 hours during the
2006 tsunami. We have removed
the word 'prolonged', and
rephrased the sentence to
improve the lack clarity.
We apologize for our lack
expression. The background
spectra are the spectral                                                                                                | Please see line 326-328         The calculated tsunami duration         at Dongkung was as much as 3.9         h, while the tsunami continued         for more than 6 h in Kaohsiung         and Houbihu.         Please see line 346 to line 350         The background spectra are the         spectral components calculated                                                                                                                                                                                                  |
| Line 254: You say 'abnormally
long', but compared to what?
Line 271: What does the
background spectra contain? Are
they de-tided? Please clarify. | We apologize for our confusing
expression. We meant that
Kaohsiung and Houbihu station
recorded the tsunami durations
for more than 6 hours during the
2006 tsunami. We have removed
the word 'prolonged', and
rephrased the sentence to
improve the lack clarity.
We apologize for our lack
expression. The background
spectra are the spectral
components calculated from de-                                                              | Please see line 326-328         The calculated tsunami duration         at Dongkung was as much as 3.9         h, while the tsunami continued         for more than 6 h in Kaohsiung         and Houbihu.         Please see line 346 to line 350         The background spectra are the         spectral components calculated         from de-tided observed data of 5                                                                                                                                                         |
| Line 254: You say 'abnormally
long', but compared to what?
Line 271: What does the
background spectra contain? Are
they de-tided? Please clarify. | We apologize for our confusing
expression. We meant that
Kaohsiung and Houbihu station
recorded the tsunami durations
for more than 6 hours during the
2006 tsunami. We have removed
the word 'prolonged', and
rephrased the sentence to
improve the lack clarity.
We apologize for our lack
expression. The background
spectra are the spectral
components calculated from de-
tided observed data of 5 h before                         | Please see line 326-328         The calculated tsunami duration         at Dongkung was as much as 3.9         h, while the tsunami continued         for more than 6 h in Kaohsiung         and Houbihu.         Please see line 346 to line 350         The background spectra are the         spectral components calculated         from de-tided observed data of 5         h before the tsunami arrival, and                                                                                                               |
| Line 254: You say 'abnormally
long', but compared to what?
Line 271: What does the
background spectra contain? Are
they de-tided? Please clarify. | We apologize for our confusing
expression. We meant that
Kaohsiung and Houbihu station
recorded the tsunami durations
for more than 6 hours during the
2006 tsunami. We have removed
the word 'prolonged', and
rephrased the sentence to
improve the lack clarity.
We apologize for our lack
expression. The background
spectra are the spectral
components calculated from de-
tided observed data of 5 h before
the tsunami arrival. | Please see line 326-328         The calculated tsunami duration         at Dongkung was as much as 3.9         h, while the tsunami continued         for more than 6 h in Kaohsiung         and Houbihu.         Please see line 346 to line 350         The background spectra are the         spectral components calculated         from de-tided observed data of 5         h before the tsunami arrival, and         the spectral components of the                                                                        |
| Line 254: You say 'abnormally
long', but compared to what?
Line 271: What does the
background spectra contain? Are
they de-tided? Please clarify. | We apologize for our confusing
expression. We meant that
Kaohsiung and Houbihu station
recorded the tsunami durations
for more than 6 hours during the
2006 tsunami. We have removed
the word 'prolonged', and
rephrased the sentence to
improve the lack clarity.
We apologize for our lack
expression. The background
spectra are the spectral
components calculated from de-
tided observed data of 5 h before
the tsunami arrival. | Please see line 326-328         The calculated tsunami duration         at Dongkung was as much as 3.9         h, while the tsunami continued         for more than 6 h in Kaohsiung         and Houbihu.         Please see line 346 to line 350         The background spectra are the         spectral components calculated         from de-tided observed data of 5         h before the tsunami arrival, and         the spectral components of the         observed tsunami waveform                                      |
| Line 254: You say 'abnormally
long', but compared to what?
Line 271: What does the
background spectra contain? Are
they de-tided? Please clarify. | We apologize for our confusing
expression. We meant that
Kaohsiung and Houbihu station
recorded the tsunami durations
for more than 6 hours during the
2006 tsunami. We have removed
the word 'prolonged', and
rephrased the sentence to
improve the lack clarity.
We apologize for our lack
expression. The background
spectra are the spectral
components calculated from de-
tided observed data of 5 h before
the tsunami arrival. | Please see line 326-328         The calculated tsunami duration         at Dongkung was as much as 3.9         h, while the tsunami continued         for more than 6 h in Kaohsiung         and Houbihu.         Please see line 346 to line 350         The background spectra are the         spectral components calculated         from de-tided observed data of 5         h before the tsunami arrival, and         the spectral components of the         observed tsunami waveform         were computed using 5 h data |

|                                   |                                   | tsunami wave arrived.             |
|-----------------------------------|-----------------------------------|-----------------------------------|
| Line 293: I think this is stating | Thank you very much for           |                                   |
| the obvious, and it could perhaps | pointing this out. We skipped     |                                   |
| be skipped?                       | this statement.                   |                                   |
| Line 329: 'determined' à          | Thank you very much for           | Please see line 388               |
| 'estimated'                       | pointing this out. The vocabulary | Assuming the mean sea depths      |
|                                   | was revised.                      | around tsunami source region is   |
|                                   |                                   | 300 m, the fault rupture          |
|                                   |                                   | dimensions for the two            |
|                                   |                                   | earthquakes could be estimated    |
|                                   |                                   | to 20- 40 km.                     |
| Line 372: I would say it is the   | Thank you very much for the       | Please see line 181-184           |
| opposite: The data can be used to | valuable comments. The            | Multiple forward tsunami          |
| validate the numerical            | sentence was rephrased.           | simulations were conducted        |
| simulations.                      |                                   | using single fault models with    |
|                                   |                                   | different fault depths and fault  |
|                                   |                                   | orientations. The main goal of    |
|                                   |                                   | the multiple forward tsunami      |
|                                   |                                   | simulations was to find a single  |
|                                   |                                   | fault model that could produce    |
|                                   |                                   | tsunami waveforms that were       |
|                                   |                                   | highly consistent with the tide   |
|                                   |                                   | gauge station observations in     |
|                                   |                                   | southern Taiwan.                  |
|                                   |                                   |                                   |
| Line 377: If there is             | We appreciate the reviewer for    | Please see section 2.4.2 (from    |
| undersampling, you would          | the valuable suggestions on this  | line 220 to 248) for the          |
| normally expect the numerical     | issue. We established and         | approach and section 5.2 (from    |
| simulations to overestimate the   | simulated the non-uniform slip    | line 464 to 478) for the results. |
| wave measurements, because the    | scenarios to examine whether      |                                   |
| measurements would miss out       | the measurements have missed      |                                   |
| on larger amplitude waves. Here   | out on larger amplitude waves.    |                                   |
| it seems to be the other way      |                                   |                                   |
| around, implying that the         |                                   |                                   |
| simulations are lower than you    |                                   |                                   |
| would expect from the             |                                   |                                   |

| measurements. The authors need
to elaborate on this. For instance,
why was not alternative
scenarios or random /
heterogeneous slip investigated
with several scenarios? |                                                                                                                                                           |                                                                                                                                                         |
|-----------------------------------------------------------------------------------------------------------------------------------------------------------------------------------------|-----------------------------------------------------------------------------------------------------------------------------------------------------------|---------------------------------------------------------------------------------------------------------------------------------------------------------|
| Line 388: Replace 'It is
commonly understood that' with
'The longest wave component'.
Then add an 'a' ahead of
'velocity'.                                                  | Thank you very much for the
valuable comments. The
vocabulary was revised, and
sentence was rephrased.                                           | Please see line 499
The longest wave component of
tsunami travel with a velocity
that is mainly governed by
seafloor depths.                |
| Line 390: Add 'through diffraction' after 'wave direction'.                                                                                                                             | We appreciate the reviewer for
the correction. The vocabulary
was added.                                                                            | Please see line 507 to 508
The significant change in
propagation speed allows the
tsunami to change its wave
direction through diffraction. |
| Line 391: 'of the' à 'using'                                                                                                                                                            | Thank you very much for the suggestion. The vocabulary was revised.                                                                                       | Please see line 511 to line 512
Simulated snapshots of tsunami
wave propagation using actual
(MS) bathymetry are shown in
Figure 21.        |
| Line 395: I found it difficult to
follow the authors in this
paragraph. I suggest that the
authors review the text and try to
rephrase it, at least the first 6-7
lines. | We apologize for the confusing
expression in this paragraph. The
paragraph was re-written.                                                          | Please see section 6.1 (from line 505 to 535).                                                                                                          |
| Line 422: I suggest to comment
on previous studies investigating
fits and misfits using open
source bathymetry and
topography data, e.g. Griffin et
al., (2015).         | Thank you very much for this
valuable suggestion. We
examined the misfits of modeled
results using open-accessible
bathymetry and topography. | Please see section 5.3 (from line
480 to 502)                                                                                                        |
| Line 426: The sentence starting
with 'These results further
confirmed' I found was                                                                                                | We appreciate the reviewer for
the valuable comments. To
strength the conclusion related to                                                         | Please see section 6.2 and 6.3 (from line 537 to 573)                                                                                                   |

| formulated too conclusive. The      | wave trapping, we applied         |                                  |
|-------------------------------------|-----------------------------------|----------------------------------|
| number of investigations are        | additional analysis including     |                                  |
| rather limited, and there should    | energy trapping ratio, and the    |                                  |
| be room for additional              | comparison of calculated          |                                  |
| investigations to strengthen the    | waveforms.                        |                                  |
| conclusion related to wave          |                                   |                                  |
| trapping.                           |                                   |                                  |
| Line 439-441: What the authors      | We apologize for the unclarity of | Pease see section 6.4 (from line |
| write here is not clear from the    | the figure. We replotted the      | 575 to 608) and Figure 27.       |
| figures. If there is additional not | figure and rephased the           |                                  |
| shown that back this up please      | statement in this paragraph.      |                                  |
| state this explicitly.              |                                   |                                  |
| Line 482: 'characterized' à         | Thank you very much for the       | Please see line 617              |
| 'analyzed'                          | suggestion. The vocabulary was    | The physical characteristics of  |
|                                     | revised.                          | tsunami waveforms in all three   |
|                                     |                                   | tide gauge stations in southern  |
|                                     |                                   | Taiwan during the December       |
|                                     |                                   | 2006 tsunami were analyzed.      |

**2.3 Numerical tsunami simulation**

Numerical simulation is a computer-based method that describes equations for the motion of tsunami wave propagation. Tsunami wave propagation can be numerically modeled based on various theories, including shallow water and dispersive wave theories. Among those theories, the shallow water equations are some of the most commonly used methods to model tsunami propagation from the source to nearshore areas. Various computational models have been developed to solve shallow water equations, and the TUNAMI (Tohoku University Numerical Analysis Model for Investigation of tsunamis) code is one of the widely used models to numerically simulate both far-field and near-field tsunamis (Suppasri et al., 2010; Suppasri et al., 2014). The second version of the TUNAMI code (TUNAMI-N2) was mainly developed to deal with near-field tsunamis by applying the nonlinear theory of shallow water equations, which is solved using a leap-frog scheme (Imamura, 1995). Since the 2006 tsunami presented as a near-field tsunami in Taiwan, the TUNAMI-N2 model was used in this study to simulate the 2006 tsunami with nonlinear shallow water equations. The nonlinear shallow water equations on the Cartesian coordinate system are presented in equations (2)-(4), and the nonlinear equations are solved by applying the finite difference method:

$$\frac{\partial \eta}{\partial t} + \frac{\partial M}{\partial x} + \frac{\partial N}{\partial y} = 0$$
(2)

$$\frac{\partial M}{\partial t} + \frac{\partial}{\partial \chi} \left(\frac{M^2}{D}\right) + \frac{\partial}{\partial y} \left(\frac{MN}{D}\right) + g D \frac{\partial \eta}{\partial \chi} + \frac{g n^2}{D^{\frac{7}{3}}} M \sqrt{M^2 + N^2} = 0$$
(3)

$$\frac{\partial N}{\partial t} + \frac{\partial}{\partial \chi} \left(\frac{MN}{D}\right) + \frac{\partial}{\partial y} \left(\frac{N^2}{D}\right) + g D \frac{\partial \eta}{\partial y} + \frac{g n^2}{D^{\frac{7}{3}}} N \sqrt{M^2 + N^2} = 0$$
(4)

[revised manuscript text omitted]